# Global retrieval of stratospheric and tropospheric BrO columns from OMPS-NM onboard the Suomi-NPP satellite

Heesung Chong[1], Gonzalo González Abad[1], Caroline R. Nowlan[1], Christopher Chan Miller[1,2], Alfonso Saiz-Lopez[3], Rafael P. Fernandez[4], Hyeong-Ahn Kwon[1,a], Zolal Ayazpour[1,5], Huiqun Wang[1], Amir H. Souri[1,b], Xiong Liu[1], Kelly Chance[1], Ewan O'Sullivan[1], Jhoon Kim[6], Ja-Ho Koo[6], William R. Simpson[7], François Hendrick[8], Richard Querel[9], Glen Jaross[10], Colin Seftor[10,11], and Raid M. Suleiman[1]

[1]Center for Astrophysics | Harvard & Smithsonian, Cambridge, MA, USA
[2]Harvard School of Engineering and Applied Sciences, Harvard University, Cambridge, MA, USA
[3]Department of Atmospheric Chemistry and Climate, Institute of Physical Chemistry Blas Cabrera, CSIC, Madrid, Spain
[4]Institute for Interdisciplinary Science (ICB), National Research Council (CONICET), FCEN-UNCuyo, Mendoza, Argentina
[5]Department of Civil, Structural and Environmental Engineering, University at Buffalo, Buffalo, NY, USA
[6]Department of Atmospheric Sciences, Yonsei University, Seoul, South Korea
[7]Geophysical Institute and Department of Chemistry and Biochemistry, University of Alaska Fairbanks, AK, USA
[8]Royal Belgian Institute for Space Aeronomy (BIRA-IASB), Brussels, Belgium
[9]National Institute of Water & Atmospheric Research Ltd (NIWA), Lauder, New Zealand
[10]NASA Goddard Space Flight Center, Greenbelt, MD, USA
[11]Science Systems and Applications, Inc., Lanham, MD, USA
[a]now at: Department of Environmental & Energy Engineering, University of Suwon, Hwaseong, Gyeonggi-do, South Korea
[b]now at: NASA Goddard Space Flight Center, Greenbelt, MD, USA

**Correspondence:** Heesung Chong (hee_sung.chong@cfa.harvard.edu)

**Abstract.** Quantifying the global bromine monoxide (BrO) budget is essential to understand ozone chemistry better. In particular, the tropospheric BrO budget has not been well characterized. Here, we retrieve nearly a decade (February 2012–July 2021) of stratospheric and tropospheric BrO vertical columns from the Ozone Mapping and Profiling Suite Nadir Mapper (OMPS-NM) onboard the Suomi National Polar-orbiting Partnership (Suomi-NPP) satellite. In quantifying tropospheric BrO
enhancements from total slant columns, the key aspects involve segregating them from stratospheric enhancements and applying appropriate air mass factors. To address this concern and improve upon the existing methods, our study proposes an approach that applies distinct BrO vertical profiles based on the presence or absence of tropospheric BrO enhancement at each pixel, identifying it dynamically using a satellite-derived stratospheric ozone-BrO relationship. We demonstrate good agreement for both stratosphere ($r$ = 0.81–0.83) and troposphere ($r$ = 0.50–0.70) by comparing monthly mean BrO vertical
columns from OMPS-NM with ground-based observations from three stations (Lauder, Utqiaġvik, and Harestua). Although algorithm performance is primarily assessed at high latitudes, the OMPS-NM BrO retrievals successfully capture tropospheric enhancements not only in polar regions but also in extrapolar areas, such as the Rann of Kutch and the Great Salt Lake. We also estimate random uncertainties in the retrievals pixel by pixel, which can assist in quantitative applications of the OMPS-NM BrO dataset. Our BrO retrieval algorithm is designed for cross-sensor applications and can be adapted to other space-borne
ultraviolet spectrometers, contributing to the creation of continuous long-term satellite BrO observation records.

# 1 Introduction

Inorganic bromine compounds ($Br_y$) contribute significantly to the loss of ozone ($O_3$) in the stratosphere through catalytic reaction cycles (Lary, 1996; Salawitch et al., 2005; Yung et al., 1980), especially exerting synergistic interactions with chlorine compounds in polar regions (Chipperfield and Pyle, 1998; Lee et al., 2002; McElroy et al., 1986; Sinnhuber et al., 2009). Stratospheric $Br_y$ compounds originate mainly from the photolysis or oxidation of organic brominated substances. The most abundant long-lived organic source gas is methyl bromide ($CH_3Br$), emitted primarily by natural oceanic processes (L. Hu et al., 2012) and by anthropogenic activities such as agriculture (Choi et al., 2022). Long-lived halons also contribute to the stratospheric $Br_y$ budget, transported from their anthropogenic emission sources (Fraser et al., 1999). Another contributor to stratospheric $Br_y$ amounts is the transport of very short-lived bromine source gases, such as bromoform ($CHBr_3$) (Pfeilsticker et al., 2000; Salawitch et al., 2005), released mainly from marine lifeforms (e.g., macroalgae and phytoplankton) (Butler et al., 2007; Raimund et al., 2011).

Bromine chemistry also affects $O_3$ concentrations and the oxidizing capacity in the troposphere (von Glasow et al., 2004; Saiz-Lopez and von Glasow, 2012; Simpson et al., 2015). $Br_y$ can be present in the free troposphere, associated with the decomposition of organic bromine compounds (Bognar et al., 2020; Dvortsov et al., 1999; Fitzenberger et al., 2000; Koenig et al., 2017; Schauffler et al., 1999; Sturges et al., 2000; Wamsley et al., 1998; Wang et al., 2015). Furthermore, ground- and aircraft-based measurements identified $Br_y$ even in the boundary layer, particularly in polar regions (Bognar et al., 2020; Hausmann and Platt, 1994; Hönninger and Platt, 2002; Peterson et al., 2015, 2017, 2018; Saiz-Lopez et al., 2007; Simpson et al., 2017), volcanic plumes (Bobrowski et al., 2003; Bobrowski and Platt, 2007; Bobrowski and Giuffrida, 2012; Boichu et al., 2011; Dinger et al., 2018; Kelly et al., 2013; Lübcke et al., 2019; Warnach et al., 2019), the marine boundary layer (Leser et al., 2003; Koenig et al., 2017; Saiz-Lopez et al., 2004), and over salt lakes (Hebestreit et al., 1999; Stutz et al., 2002). However, in-depth quantification of reactive bromine amounts in the global troposphere remains a challenge to address (Saiz-Lopez and von Glasow, 2012; Simpson et al., 2015).

Bromine monoxide (BrO) is a reactive radical accounting for a significant portion of the $Br_y$ amounts during daylight hours. Having strong absorption features in the ultraviolet (UV) spectral region, BrO is one of the earliest detected species in the history of air quality monitoring from satellite-based hyperspectral UV spectrometers (González Abad et al., 2019). The initial satellite observations were made from the Global Ozone Monitoring Experiment (GOME), suggesting the ubiquitous presence of BrO in the global free troposphere and enhanced columns mainly over polar regions (Chance, 1998; Hegels et al., 1998; Richter et al., 1998; Van Roozendael et al., 2002; Wagner and Platt, 1998).

Succeeding nadir-viewing spectrometers have continued satellite-based BrO retrievals, including the SCanning Imaging Absorption SpectroMeter for Atmospheric CHartographY (SCIAMACHY) (Van Roozendael et al., 2004), GOME-2 (Hörmann et al., 2013; Sihler et al., 2012; Theys et al., 2009a, b, 2011), the Ozone Monitoring Instrument (OMI) (Hörmann et al., 2016; Suleiman et al., 2019), and the TROPOspheric Monitoring Instrument (TROPOMI) (Herrmann et al., 2022; Seo et al., 2019). These retrievals have demonstrated the detectability of extrapolar BrO enhancements from satellites. For example, Hörmann et al. (2016) analyzed the seasonal variations of BrO columns over the Rann of Kutch, a salt marsh located on the border between

Pakistan and India, using OMI and GOME-2 observations. Retrievals from OMI also detected enhanced BrO columns over the Great Salt Lake in the USA (Chance, 2006; Suleiman et al., 2019) and the Dead Sea (Hörmann et al., 2016). Furthermore, GOME-2, OMI, and TROPOMI observed BrO emissions from volcanoes (Heue et al., 2011; Hörmann et al., 2013; Seo et al., 2019; Suleiman et al., 2019; Theys et al., 2009a).

In response to a lack of quantitative understanding of the tropospheric $Br_y$ budget (Saiz-Lopez and von Glasow, 2012; Simpson et al., 2015), the separation between stratospheric and tropospheric columns has been among the primary interests of satellite-based BrO studies. The common framework of the existing separation approaches involves deriving the tropospheric field by subtracting stratospheric columns from the total columns retrieved from a nadir-viewing satellite sensor (see Sect 2.5.1 for details). In this framework, an important aspect is avoiding the misattribution of stratosphere-driven variabilities in total columns to tropospheric enhancements (Salawitch et al., 2010). Another consideration is addressing the high variability of light paths in the troposphere, which impacts the accuracy of the retrieved tropospheric vertical columns. To enhance the global applications of satellite BrO data, we suggest a modified stratosphere-troposphere separation (STS) method, combining the benefits of the existing methods. The key feature of the proposed scheme is the dynamic identification of tropospheric enhancements, where distinct BrO vertical profiles are applied depending on the presence or absence of enhancements on a pixel-by-pixel basis.

In this study, we retrieve global stratospheric and tropospheric BrO vertical columns from the Ozone Mapping and Profiling Suite Nadir Mapper (OMPS-NM) onboard the Suomi National Polar-orbiting Partnership (Suomi-NPP) satellite launched in 2011. Starting with the one on the Suomi-NPP, two more OMPS-NM instruments have been deployed on the NOAA-20 and NOAA-21 satellites in 2017 and 2022, respectively. There are also plans for two additional launches scheduled in 2027 and 2032. Building a long-term time series of BrO using multiple OMPS-NM instruments can minimize the complicated intercalibration required when combining datasets from different sensors (Bougoudis et al., 2020). The OMPS-NM instruments are currently the only planned space-borne hyperspectral UV spectrometers to continuously be launched into afternoon orbit subsequent to the decommissioning of TROPOMI (Nowlan et al., 2023). Furthermore, the one onboard the Suomi-NPP can specifically provide daily global afternoon BrO data from 2012, which are currently missing in part due to the influence of an instrumental issue (the so-called "row anomaly") on the OMI BrO product (Suleiman et al., 2019).

In Sect. 2, we describe in detail the OMPS-NM instrument, our retrieval algorithm, and estimated uncertainties. Section 3 presents the intercomparison of stratospheric and tropospheric BrO columns from OMPS-NM and ground-based retrievals from February 2012 to July 2021. While the consistent algorithm configuration is applied globally, the retrieval examples discussed in Sects. 2–3 primarily center around high latitudes, considering the substantial variabilities and implications of BrO concentrations in those regions. In Sect. 4, we broaden our examination to a global perspective, analyzing tropospheric BrO columns retrieved from 8-year OMPS-NM measurements (January 2013–December 2020). Additionally, we explore extrapolar hotspots detected from February 2012 to July 2021. Section 5 provides a discussion and conclusions.

## 2 OMPS-NM BrO retrieval

Figure 1 shows the flow chart of the OMPS-NM BrO retrieval algorithm. In the framework of the two-step trace-gas retrieval method (González Abad et al., 2019), the algorithm first retrieves slant columns from earthshine radiance spectra stored in the OMPS-NM Level 1B product, described in Sect. 2.1. Here, the slant column refers to the amount of BrO integrated along contributing light paths. The ultimate algorithm outputs, stratospheric and tropospheric BrO vertical columns, are subsequently derived by accounting for these light paths. The retrieval algorithm consists of four main components: (i) slant column retrieval, (ii) air mass factor calculations, (iii) reference sector correction, and (iv) stratosphere-troposphere separation, which are framed in blue in Fig. 1. The four algorithm components are described in Sects. 2.2–2.5 in the order of execution. Uncertainties in the retrievals are described in Sect. 2.6.

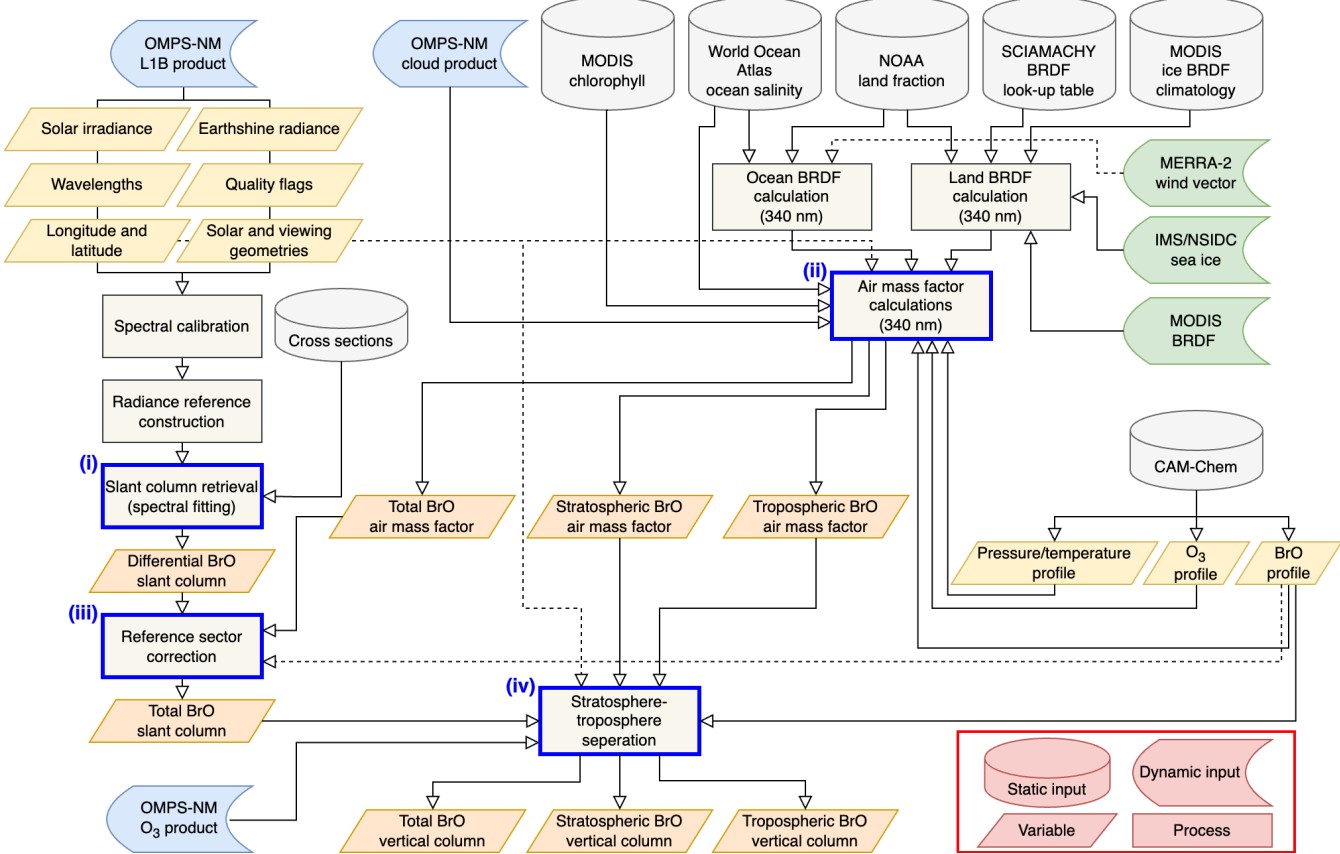

**Figure 1.** Flow chart of the OMPS-NM BrO retrieval algorithm. Different symbols are used for static input, dynamic input, variable, and process, as indicated in the lower-right corner. Four main algorithm components (i–iv) are highlighted with blue frames.

## 2.1 OMPS-NM instrument and Level 1B product

The OMPS-NM instrument, launched on 28 October 2011 onboard the Suomi-NPP spacecraft, measures backscattered earth-shine radiances from a low Earth orbit with an equatorial overpass of 13:30 local solar time (LST) (Dittman et al., 2002; Flynn et al., 2014; Seftor et al., 2014). The instrument uses a grating spectrometer with a 2-dimensional charge-coupled device (CCD) detector comprising 340 (spectral) × 740 (spatial) pixels, of which 196 × 708 are illuminated. In the spectral dimension, the illuminated pixels cover a wavelength range of 300–380 nm at 0.42 nm sampling and 1 nm resolution (full width at half maximum). Spatially, the CCD pixels are projected onto the Earth's surface with a 110° field of view, resulting in a 2800 km-wide swath and daily full global coverage at the equator.

For nominal operation, the spatial CCD pixels are rebinned into 36 cross-track positions to meet the noise and ground pixel size requirements. As a result, OMPS-NM has a spatial resolution of 50 km in the cross-track dimension at the nadir. A different rebinning approach is applied to the two central cross-track positions, providing 30 km × 50 km and 20 km × 50 km resolutions. The signal-to-noise ratios (SNRs) after rebinning are above 1000 at all wavelengths (Seftor et al., 2014). In the along-track dimension, the integration time of 7.6 s leads to a resolution of 50 km. Each OMPS-NM orbit typically has 400 swaths (along-track pixels), where a swath is a single set of 36 cross-track measurements.

For the BrO retrieval, we use solar irradiance and earthshine radiance data, along with corresponding geographic locations of ground pixels, observation geometries, wavelengths, and quality flags (see Fig. 1). These data are from the NASA OMPS Nadir Mapper Earth View (NMEV) Version 2.0 Level 1B product, accessible through the Goddard Earth Sciences Data and Information Services Center (GES DISC) (Johnson and Seftor, 2017). Unlike radiances measured at every pixel, solar irradiances in the Level 1B product are based on four measurements taken in March and April of 2012, adjusted to the Sun-Earth distance for the time of radiance measurements. The Level 1B data used in this study cover the time period from February 2012 to July 2021.

## 2.2 Slant column retrieval (spectral fitting)

The retrieval algorithm starts with spectral calibration of solar irradiance measured by the OMPS-NM instrument (see blue frame i and the preceding steps in Fig. 1). This calibration provides an on-orbit spectral response function (SRF) of OMPS-NM for each cross-track position, which is required to convolve high-resolution reference spectra in the following steps. We adopt the approach outlined by Beirle et al. (2017) and Nowlan et al. (2023) to model the SRF using a super-Gaussian. The optimal super-Gaussian parameters are derived for each cross-track position simultaneously with a spectral shift by iterative cross-correlation between the measured spectrum and a convolved high-resolution solar reference spectrum (Chance and Kurucz, 2010).

We retrieve a BrO slant column for each ground pixel of OMPS-NM, employing the Smithsonian Astrophysical Observatory (SAO) approach that performs direct least-squares fitting of a modeled radiance spectrum $\boldsymbol{F}(\boldsymbol{x}, \boldsymbol{b})$ to a measurement vector $\boldsymbol{y}$

(Chance, 1998):

$$\hat{\boldsymbol{x}} = \arg\min_{\boldsymbol{x}} \sum_{i=1}^{m} [y_i - F_i(\boldsymbol{x}, \boldsymbol{b})]^2. \tag{1}$$

Here, $\boldsymbol{y}$ consists of earthshine radiances measured at discretely sampled wavelengths ($\lambda$) in a fitting window, with the number of spectral points referred to as $m$; $\boldsymbol{x}$ represents a state vector composed of a set of geophysical and spectroscopic variables, including the slant column of BrO; $\boldsymbol{b}$ describes predetermined model parameters; and $\hat{\boldsymbol{x}}$ is the retrieved state vector. The retrieval is based on nonlinear regression, with a Jacobian matrix $\mathbf{K}_x = \partial \boldsymbol{F}/\partial \boldsymbol{x}$ updated in each iteration using the Gauss-Newton ELSUNC algorithm (Lindström and Wedin, 1987). Bad pixels determined by quality flags from Level 1B files are excluded from the spectral fitting.

Modeling $\boldsymbol{F}(\boldsymbol{x}, \boldsymbol{b})$ requires a source spectrum $I_0$ that is under minimal or no influence of the absorption by the trace gas of interest (BrO, in this study). Solar irradiance measured from the same sensor is a traditional option for $I_0$. However, we use a radiance reference to minimize cross-track striping in the retrieved slant columns, as in previous OMPS-NM retrieval studies (González Abad et al., 2016; Nowlan et al., 2023). Our algorithm constructs a radiance reference daily for each cross-track position by averaging the earthshine radiance spectra measured at 0–10°N from a reference orbit. Here, the reference orbit refers to the one overpassing 160°W at the equator (over the Pacific), selected for minimal spatial and seasonal variabilities in the total BrO columns. In this study, the latitude range is chosen as a compromise, narrowing to simplify BrO variabilities while ensuring simultaneously that there are sufficient radiance samples to achieve reliable averages with suitable SNRs for retrieval. This area is hereafter referred to as the "reference sector." The use of equatorial radiance references can also be found in other satellite-based BrO retrieval studies (Bougoudis et al., 2020; Herrmann et al., 2022; Seo et al., 2019).

Once $I_0$ is constructed, we perform spectral calibration using the predetermined SRFs to correct for spectral shifts. The spectrally calibrated $I_0$ is then input into the formula to derive $\boldsymbol{F}(\boldsymbol{x}, \boldsymbol{b})$:

$$F(\lambda) = \left[ (I_0(\lambda = \lambda' + x_s) + x_u b_u(\lambda) + x_r b_r(\lambda)) e^{-\sum_j x_j b_j(\lambda)} \right] \sum_{k=0}^{n_{SC}} x_k^{SC} (\lambda - \overline{\lambda})^k + \sum_{l=0}^{n_{BL}} x_l^{BL} (\lambda - \overline{\lambda})^l, \tag{2}$$

where each term represents either an atmospheric or instrumental process that a radiance spectrum undergoes until the sensor makes the measurement. The variable $x_s$ represents a spectral shift in the wavelength registration of $y(\lambda)$ versus $I_0(\lambda')$, mainly caused by thermal changes in the instrument. The states $x_u$ and $x_r$ in the first two additive terms account for the effects of the undersampling correction (Chance et al., 2005) and rotational Raman scattering (Chance and Spurr, 1997), whose spectra are represented by $b_u(\lambda)$ and $b_r(\lambda)$, respectively. The following multiplicative term $e^{-\sum_j x_j b_j(\lambda)}$ accounts for trace gas absorption based on the Beer-Lambert law, where $x_j$ and $b_j(\lambda)$ represent a slant column and the absorption cross section of a trace gas species $j$, respectively. The cross sections are convolved using the modeled SRF and corrected for the solar $I_0$ effect (Aliwell et al., 2002) before the spectral fitting. Since the radiance reference itself contains nonzero trace gas information, the retrieved states $\hat{x}_j$ here are referred to as "differential" slant column densities ($\Delta$SCDs). Lastly, the algorithm considers broadband features such as molecular scattering, aerosol attenuation, and surface reflection, using the variables $x_k^{SC}$ and $x_l^{BL}$ as coefficients of scaling ($n_{SC}$th degree) and baseline ($n_{BL}$th degree) polynomials that are symmetric with respect to the center of the fitting window ($\overline{\lambda}$).

**Table 1.** Parameter configuration for OMPS-NM BrO retrieval. The parameters are listed in their order of appearance in Eq. (2).

| Parameter | Detail |
|---|---|
| Spectral shift | |
| Undersampling correction spectrum | Chance et al. (2005) |
| Rotational Raman scattering spectrum | Chance and Spurr (1997) |
| BrO cross section (228 K) | Wilmouth et al. (1999) |
| $O_3$ cross sections (243 and 273 K) | Serdyuchenko et al. (2014) |
| The first-order Taylor series expansion for $O_3$ absorption (243 K) | Puķīte et al. (2010) |
| $O_2$–$O_2$ cross section | Finkenzeller and Volkamer (2022) |
| $NO_2$ cross section | Vandaele et al. (1998) |
| HCHO cross section | Chance and Orphal (2011) |
| Baseline polynomial | Zeroth order |
| Scaling polynomial | Second order |
| Fitting window | 331.5–358 nm |

Table 1 presents the details of the parameters used for the spectral fitting, including the cross sections of the trace gases and the degrees of polynomials. The selection of trace gases for fitting is based on their impacts on fitting root-mean-square error (RMSE) and BrO $\Delta$SCD uncertainty values, as well as the spatial distribution of each species resulting from the fit. To account for the wavelength dependence of the $O_3$ slant columns, we include two additional parameters derived from the first-order Taylor series expansion as suggested by Puķīte et al. (2010). For numerical stability, we normalize all spectra close to unity, including the irradiance, radiance, and cross sections. We use the fitting window of 331.5–358 nm, optimized by assessing fitting RMSEs, BrO $\Delta$SCD uncertainties, and interferences between Jacobians of BrO and other trace gas $\Delta$SCDs. Details of the fitting window optimization are described in Appendix A.

Based on the spectral fitting results, we assign quality flags to OMPS-NM BrO retrievals. If a certain pixel meets the following three requirements, we define it as a "good" pixel: (a) the fitting converges above the noise level (determined by the ELSUNC algorithm); (b) the retrieved $\Delta$SCD is smaller than $1.0 \times 10^{19}$ molecules cm$^{-2}$; (c) $\Delta$SCD is greater than negative two times its random uncertainty (the random $\Delta$SCD uncertainty estimation is described in Sect. 2.6.1). It is considered "bad" if the fitting fails to converge within 10 iterations or the sum of $\Delta$SCD and three times its random uncertainty is smaller than zero. Lastly, all remaining cases are considered "suspect." Among the 47280 OMPS-NM orbits processed through the last stage of the algorithm (from February 2012 to July 2021), the proportions of good, suspect, and bad pixels are 98.7%, 1.1%, and 0.2%, respectively.

Figure 2 shows an example of slant optical depths of the fitted gases on 12 April 2018 for a single OMPS-NM pixel in Hudson Bay. For $O_3$ optical depths, we combine the two Taylor series parameters and the cross sections at the two temperatures (Puķīte et al., 2010). Despite the dominating optical depths of $O_3$, the BrO signal is clearly detected with a $\Delta$SCD of $2.11 \times 10^{14}$ molecules cm$^{-2}$. The fitting RMSE in this example is low at $4.24 \times 10^{-4}$.

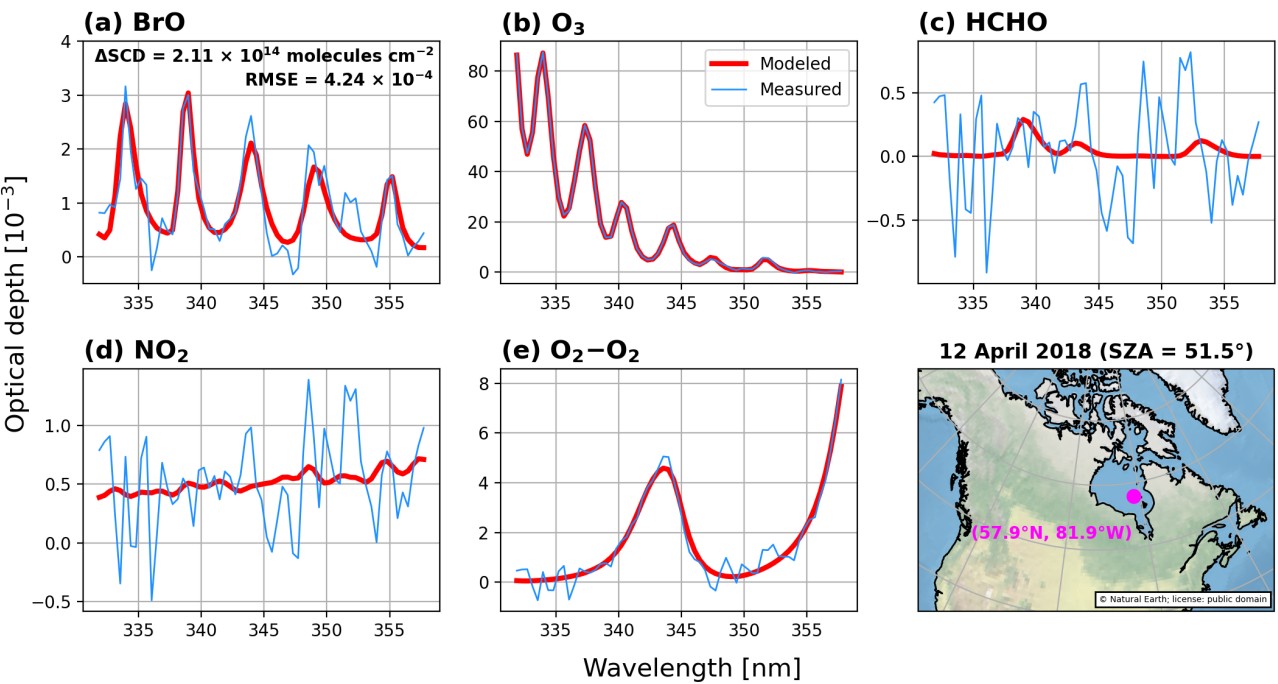

**Figure 2.** Slant optical depths of fitted gases in Hudson Bay. The date, latitude, longitude, and solar zenith angle (SZA) of the observation are presented in the lower-right panel. Optical depths of (a) BrO, (b) $O_3$, (c) formaldehyde (HCHO), (d) nitrogen dioxide ($NO_2$), and (e) the oxygen collision-induced absorption ($O_2$–$O_2$) are shown in the respective panels. The red and blue curves represent the modeled and measured optical depths, respectively. The measured optical depths are defined as the sum of modeled optical depths and the residuals.

## 2.3 Air mass factor calculations

In the two-step retrieval method, converting a trace-gas slant column to a vertical column requires an air mass factor (AMF), a dimensionless quantity that accounts for the sum over possible light paths. By definition, the AMF is equivalent to the ratio of the slant to vertical columns of the trace gas. Assuming optically thin absorption and neglecting the temperature dependence of the cross sections, we calculate the AMF ($A$) following the formula of Palmer et al. (2001):

$$A = \frac{\sum_{p=n_l}^{n_u} W_p C_p}{\sum_{p=n_l}^{n_u} C_p}. \tag{3}$$

For computational purposes, the continuous atmosphere is divided into discrete vertical layers. Here, $n_l$ and $n_u$ are the indices of the lower and upper limits of the atmospheric layers used for the AMF calculation. The variable $W_p$ represents a scattering weight, the sensitivity of the total slant optical depth of the atmosphere to a partial vertical optical depth of the $p$th layer. The variable $C_p$ represents a partial vertical column of the trace gas at the $p$th layer. We define the proportion $C_p / \sum_{p=n_l}^{n_u} C_p$ from Eq. (3) as the "shape factor."

Separate determination of stratospheric and tropospheric vertical columns in this study requires total ($A_\text{total}$), stratospheric ($A_\text{strat}$), and tropospheric ($A_\text{trop}$) AMFs. All three quantities are calculated following Eq. (3), and the only differences are in the setting of $n_l$ and $n_u$. The total AMF is calculated using $W_p$ and $C_p$ values from the ground to the top of the atmosphere, while the stratospheric and tropospheric AMFs cover only layers above and below the tropopause, respectively.

We determine the AMFs by online radiative transfer calculations using the Vector LInearized Discrete Ordinate Radiative Transfer (VLIDORT) model Version 2.8 (Spurr, 2006, 2008; Spurr and Christi, 2019). Calculations are carried out on 26 vertical layers defined by the Community Atmosphere Model with Chemistry (CAM-Chem) climatology (Fernandez et al., 2019), from which we obtain atmospheric profiles including partial vertical columns of BrO (i.e., $C_p$ in Eq. 3). Details of the CAM-Chem climatology are presented below.

The spectroscopic and geophysical variables determining the scattering weights include observation geometries, cloud properties, surface reflectance, and optical depth profiles of $O_3$ absorption, Rayleigh scattering, and aerosol attenuation. In the UV spectral region, $O_3$ absorption and Rayleigh scattering vary with wavelength, resulting in the spectral dependence of AMFs. However, as the variability of BrO AMFs is relatively small in the fitting window of the present study (331.5–358 nm) (Suleiman et al., 2019), we use a single-wavelength AMF at 340 nm for computational efficiency. Table 2 summarizes the

variables input to the AMF calculations and the corresponding datasets used to quantify them (see also blue frame ii in Fig. 1). Detailed descriptions are presented in the following.

**Atmospheric profiles**

We employ a monthly diurnal climatology derived from the CAM-Chem model to obtain vertical profiles of $O_3$, BrO, pressure (including the tropopause pressure), and temperature. This climatology was produced with an interactive polar module, which

considered the ground-level photochemical production of full gas-phase and heterogeneous inorganic halogen species from sea ice and snowpack (Fernandez et al., 2019). The model provides global coverage with a horizontal resolution of 1.9° latitude × 2.5° longitude. Vertically, it covers from the surface up to ~3 hPa (~40 km) with 26 layers.

     For each OMPS-NM pixel, we sample profiles for the month and hour of the measurement from the horizontally nearest model grid. The profiles are used to calculate partial BrO vertical columns ($C_p$) in Eq. (3) and optical depths of $O_3$ absorption

and Rayleigh scattering. In addition, the tropopause pressure is used for STS, whose details are described in Sect. 2.5.

**Surface properties**

We derive surface reflectance with different approaches depending on the surface type (i.e., land, water, and sea ice). On land, we use a bidirectional reflectance distribution function (BRDF) product from the MODerate Resolution Imaging Spectroradiometer (MODIS) (MCD43C1 Version 6.1), which has a 0.05° × 0.05° resolution (Schaaf and Wang, 2021). The shortest

wavelength covered by the MODIS bands is 469 nm. To extend the BRDF kernels to 340 nm, we fit empirical orthogonal functions (EOFs) to the MODIS retrievals from the four shortest wavelength bands (469–859 nm). These spectral EOFs are derived from a high-spectral-resolution surface reflectance database, which has been acquired by merging the visible surface

**Table 2.** Inputs to air mass factor calculations.

| Input | Detail |
|---|---|
| Wavelength | 340 nm |
| Geographic locations and observation geometries | OMPS-NM Level 1B product (Jaross, 2017b) |
| $O_3$ and BrO profiles | CAM-Chem monthly diurnal climatology (Fernandez et al., 2019) |
| Pressure and temperature profiles | CAM-Chem monthly diurnal climatology (Fernandez et al., 2019) |
| Tropopause pressure | CAM-Chem monthly diurnal climatology (Fernandez et al., 2019) |
| Surface reflectance (land) | MODIS BRDF product MCD43C1 (Schaaf and Wang, 2021) extended to UV using EOFs |
| Surface reflectance (water) | Cox-Munk slope distribution (Cox and Munk, 1954) |
| Surface reflectance (sea ice) | Climatology derived using MODIS BRDF product MCD43C1 (Schaaf and Wang, 2021) |
| Wind vectors at 2 m above ground level | MERRA-2 (GMAO, 2015a) |
| Ocean salinity | Monthly climatology from World Ocean Atlas 2009 (Antonov et al., 2010) |
| Chlorophyll | MODIS Terra monthly climatology (C. Hu et al., 2012) |
| Land fraction | NOAA GLOBE (Hastings and Dunbar, 1999) |
| Sea ice fraction (Northern Hemisphere) | IMS Daily Northern Hemisphere Snow and Ice Analysis (U.S. National Ice Center, 2008) |
| Sea ice fraction (Southern Hemisphere) | NSIDC Sea Ice Index (Fetterer et al., 2017) |
| Cloud fraction | OMPS-NM cloud product (Joiner, 2020) with additional snow/ice pixel treatment |
| Cloud pressure | OMPS-NM cloud product (Joiner, 2020) |
| Aerosols | not included explicitly |

reflectance libraries produced by Zoogman et al. (2016) with the SCIAMACHY surface reflectance climatology (Tilstra et al., 2017). The same BRDF extension approach has also been employed for OMPS-NM HCHO retrievals (Nowlan et al., 2023).

The MODIS BRDF kernels, developed to describe the BRDF of the land surface, are unavailable in moderate or deep water regions (Schaaf et al., 2002) and are less reliable in shallow water regions (Fasnacht et al., 2019). Therefore, we determine the surface reflectances of all water bodies using the Cox-Munk slope distribution derived by wind speed/direction and salinity (Cox and Munk, 1954). We obtain the wind vectors at 2 m above ground level from an hourly time-averaged 2-dimensional data collection in the Modern-Era Retrospective Analysis for Research and Applications Version 2 (MERRA-2) with a spatial resolution of $0.5°$ latitude $\times$ $0.625°$ longitude (GMAO, 2015a). The ocean salinity data are acquired from a monthly climatology from the World Ocean Atlas 2009 at $1° \times 1°$ resolution (Antonov et al., 2010). The VBRDF supplement in the VLIDORT model is used for reflectance calculations (Spurr and Christi, 2019).

In addition to reflected sunlight, we consider surface-leaving radiance from water bodies using the VSLEAVE supplement in VLIDORT (Spurr and Christi, 2019). Calculating the water-leaving radiance requires chlorophyll concentration, observation

geometries, and wind speed. For chlorophyll concentrations, we use the MODIS Terra 18-year monthly climatology (2000–2018), which has a resolution of 9.28 km (C. Hu et al., 2012).

To account for the reflectance of sea ice, we produce a 19-year ice BRDF climatology using the MCD43C1 product. Since the MODIS kernels are available over shallow water regions, albeit with lower accuracy, we use them to describe BRDFs of ice on waters. First, we derive monthly mean BRDF kernels for ice on inland waters globally at $0.05° \times 0.05°$ resolution for 15 December 2000–15 January 2020 (19 years). In this step, we sample only pixels with 100% snow fractions, 0% land fractions, and quality flags $\leq 2$ ("relatively good" to "best" qualities). Second, we calculate a 19-year global median for each kernel (i.e., isotropic, volumetric, and geometric) by aggregating the monthly gridded mean data across all locations and months. As a result, a single representative value for each BRDF kernel is acquired to account for the global ice reflectance. This procedure is applied to each of the four shortest wavelength bands of MODIS. The climatological ice BRDF kernels thus obtained are then extended to 340 nm during BrO retrieval, using the same method as that applied to the land BRDF kernels.

The above-mentioned approaches to determine surface reflectances apply to pure land, water, and sea ice pixels. In practice, OMPS-NM pixels can be inhomogeneous (i.e., mixtures of land, water, and sea ice). We account for the surface reflectances of inhomogeneous pixels by

$$k_{q\in\{\text{iso, geo, vol}\}} = f_{\text{land}} \cdot k_q^{\text{land}} + f_{\text{ice}} \cdot k_q^{\text{ice}} + (1 - f_{\text{land}} - f_{\text{ice}}) \cdot a_{\text{water}}. \tag{4}$$

Here, $k_{q\in\{\text{iso, geo, vol}\}}$ represents either an isotropic, geometric, or volumetric kernel at 340 nm for a given inhomogeneous pixel. The parameters $f_{\text{land}}$ and $f_{\text{ice}}$ represent the fractions of land and sea ice, whose 340-nm BRDF kernels are denoted by $k_q^{\text{land}}$ and $k_q^{\text{ice}}$, respectively. The variable $a_{\text{water}}$ accounts for the surface reflectance for pure water, determined by the Cox-Munk slope distribution. The land fractions are derived using the NOAA Global Land One-kilometer Base Elevation (GLOBE) data (Hastings and Dunbar, 1999). For the Northern Hemisphere, the sea ice fraction of each OMPS-NM pixel is calculated using the 4-km Interactive Multisensor Snow and Ice Mapping System (IMS) product (U.S. National Ice Center, 2008). Sea ice fractions for the Southern Hemisphere are determined using the Sea Ice Index from the National Snow and Ice Data Center (NSIDC) (Fetterer et al., 2017). Both the IMS and NSIDC products are updated daily. The joint use of multiple data sources in Eq. (4) may encounter differences in surface-type definitions. If the IMS or NSIDC data indicate snow or land over water as determined by GLOBE, we update $k_q^{\text{land}}$ accordingly to employ MODIS BRDF kernels for a larger fraction. Since these occurrences are typically noted on bright surfaces (e.g., ice shelves around Antarctica), this step prevents significant underestimations of surface albedos.

**Clouds and aerosols**

We account for the influence of clouds on the scattering weights with the independent pixel approximation (Martin et al., 2002):

$$W_{p\in\{n_l, n_l+1, ..., n_u\}} = (1 - c_{\text{rad}}) \cdot W_p^{\text{clear}} + c_{\text{rad}} \cdot W_p^{\text{cloud}}, \tag{5}$$

where $c_{\text{rad}}$ represents a radiative cloud fraction, and the variables $W_p^{\text{clear}}$ and $W_p^{\text{cloud}}$ denote the scattering weights for completely clear and cloudy scenes, respectively. The radiative cloud fraction is calculated by

$$c_{\text{rad}} = \frac{c_{\text{eff}} I_{\text{cloud}}}{(1 - c_{\text{eff}}) I_{\text{clear}} + c_{\text{eff}} I_{\text{cloud}}}, \tag{6}$$

where $c_{\text{eff}}$ represents an effective cloud fraction (ECF), and $I_{\text{clear}}$ and $I_{\text{cloud}}$ are the VLIDORT-simulated radiances of completely
clear and cloudy scenes, respectively. We use a Lambertian cloud model with a fixed albedo of 0.8, which also applies to calculating $W_p^{\text{cloud}}$ in Eq. (5). Determining $W_p^{\text{cloud}}$ and $I_{\text{cloud}}$ requires cloud pressure as input. We obtain the cloud pressure from the OMPS-NM cloud product (OMPS-NPP_NMCLDRR-L2 Version 2.0), along with the ECF (Joiner, 2020; Vasilkov et al., 2014).

Since snow/ice surfaces play important roles in bromine activation (Simpson et al., 2015), it is essential to enhance the
accuracy of AMF calculations over snow/ice pixels. However, due to the inherent difficulty in discriminating snow/ice from clouds, the NMCLDRR-L2 algorithm assigns constant ECFs of 100% to snow/ice pixels. This decision was made to identify the existence of thick clouds (Johnson et al., 2020).

Meanwhile, the effective scene (cloud) pressure is derived using the same rotational Raman scattering (RRS) approach regardless of the surface type. Vasilkov et al. (2010) segregated clouds over snow/ice pixels from the OMI RRS cloud product
by assessing the differences between scene and surface pressure values. Adapting this approach, we determine whether to treat a given snow/ice scene from the NMCLDRR-L2 product as a cloud or surface based on the difference between the scene and surface altitudes. If the scene-surface altitude difference is smaller than 100 m, we replace the ECF with 0% to secure clear-sky scenes. The scene-surface altitude differences are calculated based on the barometric formula with nonzero standard temperature lapse rate (COESA, 1976):

$$z_c - z_s = \frac{T_s}{\Gamma} \left( 1 - \left( \frac{P_c}{P_s} \right)^{\frac{R\Gamma}{g}} \right), \tag{7}$$

where $z_c$ and $z_s$ represent scene (cloud) and surface altitudes above sea level, respectively; $\Gamma$ denotes the lapse rate (0.0065 K m$^{-1}$); $T_s$ is the surface temperature from CAM-Chem; $P_c$ and $P_s$ represent the scene (cloud) and surface pressure, respectively; $R$ is the ideal gas constant (287 J kg$^{-1}$ K$^{-1}$); and $g$ denotes the acceleration of gravity (9.8 m s$^{-2}$).

The presence of aerosols can increase or decrease the number of photons absorbed by trace gases, depending on their vertical
profiles and optical properties (Leitão et al., 2010). Scattering aerosols increase the light path length within/above their layer and shield photons from penetrating below it. Absorbing aerosols reduce the sensitivity of radiance measurements to trace gas amounts within and below their layer. Therefore, including aerosols in the radiative transfer calculations changes scattering weights (Hong et al., 2017; Jung et al., 2019; Kwon et al., 2017; Leitão et al., 2010). However, we calculate AMFs without aerosol inputs as the RRS cloud algorithm implicitly considers some of the radiative effects of aerosols. The mixed Lambertian-
equivalent reflectivity (MLER) approach used in the RRS algorithm simultaneously accounts for the scattering of aerosols and clouds (Joiner and Vasilkov, 2006). If absorbing aerosols are present in or above clouds, the RRS algorithm provides lower cloud fraction and pressure values (Johnson et al., 2020; Vasilkov et al., 2008).

## 2.4 Reference sector correction

Since we use radiance reference in the spectral fitting, the retrieved BrO $\Delta$SCD ($\Delta S$) represents the difference between the total SCD at a given OMPS-NM pixel and the background SCD ($S_R$) in the reference sector. Therefore, to determine the total BrO SCDs, it is necessary to add the background SCD estimates to $\Delta$SCDs. The resultant total SCDs, however, have systematic biases that smoothly vary in the along-track dimension, mainly induced by errors in radiance measurements or during the spectral fitting at high latitudes and high solar zenith angles (SZAs) (Nowlan et al., 2023). Accordingly, we correct this bias for each pixel by adding a correction term $S_B$. In brief, we determine the final total BrO SCD for each OMPS-NM pixel ($S_{\text{total}}$) by

$$S_{\text{total}} = \Delta S + S_R + S_B. \tag{8}$$

The combined procedure of applying $S_R$ and $S_B$ to determine the total SCD is referred to as the reference sector correction (see blue frame iii in Fig. 1).

To estimate the background SCD ($S_R$) for each OMPS-NM orbit, we first multiply the modeled total vertical column densities (VCDs) of BrO from the CAM-Chem climatology (i.e. $\sum_{p=1}^{26} C_p$) by the co-located total AMFs within the reference sector[1]. This step provides a modeled total SCD for every pixel in the reference sector. Then we determine $S_R$ for each cross-track position by calculating the median of the modeled SCDs in the sector. A single $S_R$ value is constantly applied to every along-track pixel in each cross-track position separately, as a fixed radiance reference is used for each cross-track position in the spectral fitting procedure.

Then we derive the bias correction terms ($S_B$) by comparing the baseline of the background-corrected SCDs (i.e., $\Delta S + S_R$) and the baseline of the modeled total SCDs for each cross-track position of the reference orbit. Here, the baseline refers to a smooth trend in SCDs in the along-track dimension, which is determined through a third-degree polynomial fit. This approach assumes that without biases, the background-corrected SCDs would have the same baseline as modeled, attributed only to physical changes in local background BrO amounts that vary with latitudes and SZAs. Unlike $S_R$, the $S_B$ values are determined using all along-track pixels from the reference orbit. To avoid the potential contamination from enhanced BrO SCDs in the baseline extraction, the polynomial fitting excludes pixels where the absolute differences between the background-corrected and modeled SCDs exceed $1.0 \times 10^{14}$ molecules cm$^{-2}$. Once the baselines of the background-corrected and modeled total SCDs are extracted for a given cross-track position, their difference is allocated to each along-track pixel as $S_B$.

Figure 3 shows examples of the intermediate variables and the resulting $S_{\text{total}}$ field from the reference sector correction. Presented here is orbit number 7594 (o7594), with $S_R$ and $S_B$ values derived from the reference orbit o7585. An important aspect of the reference sector correction is to preserve detailed spatial structures in the retrieved $\Delta S$ field, simultaneously addressing

---

[1]The modeled total VCDs within the reference sector (0–10°N) are typically $2.0$–$2.2 \times 10^{13}$ molecules cm$^{-2}$, smaller than the constant VCD employed by Seo et al. (2019) and Richter et al. (2002) ($3.5 \times 10^{13}$ molecules cm$^{-2}$). However, within the reference sector of the previous studies (30°S–30°N), the modeled VCDs range within $2.0$–$5.9 \times 10^{13}$ molecules cm$^{-2}$. In particular, for February, the mean VCD within 30°S–30°N is $2.9 \times 10^{13}$ molecules cm$^{-2}$, which is closer to the constant used in the previous studies.

offsets that smoothly vary in the along-track direction. The comparison between Fig. 3a and d illustrates the consistent spatial patterns in the $\Delta S$ and $S_{\text{total}}$ fields, further supported by Fig. 3e that shows the along-track variations of $\Delta S$ and $S_{\text{total}}$ overlaid.

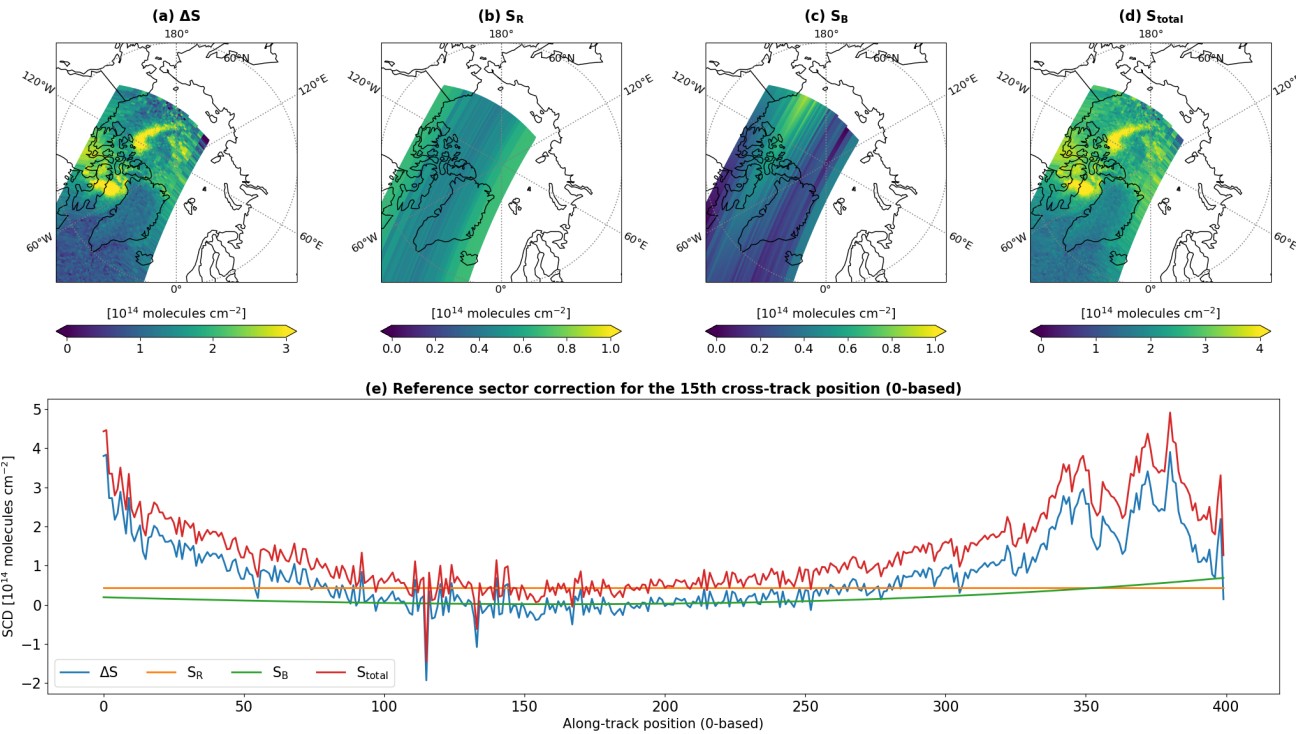

**Figure 3.** Description of the reference sector correction. Intermediate quantities are presented for o7594 from 15 April 2013. Panels (a–d) show the fields of $\Delta$SCD ($\Delta S$), background SCD ($S_R$), bias correction term ($S_B$), and total SCD ($S_{\text{total}}$), respectively. Panel (e) depicts the along-track variabilities of the four quantities for the 15th cross-track position (0-based).

## 2.5 Stratosphere-troposphere separation

The last stage of the OMPS-NM BrO retrieval algorithm is the STS, which provides stratospheric, tropospheric, and total BrO VCDs by combining the SCDs and AMFs determined in the previous stages (see blue frame iv in Fig. 1). In Sect. 2.5.1, we provide an overview of the existing methods, and in Sect. 2.5.2, we describe the method proposed in this study.

### 2.5.1 Overview of existing methods

To our knowledge, the separation approaches employed so far can be roughly categorized into four groups, hereafter referred to as M1 to M4. Typically, these methods derive the tropospheric BrO field by subtracting stratospheric columns from the total columns retrieved from a nadir-viewing satellite sensor. The primary differences among the methods lie in the estimation of stratospheric columns.

In the first method (M1), stratospheric columns are constructed using BrO vertical profiles from limb-viewing satellite observations. This observation-based method showed reliable performance (Koo et al., 2012). For consistent long-term applications, however, it requires new limb-viewing BrO datasets after the decommissioning of SCIAMACHY in 2012.

The second method (M2) estimates stratospheric columns using the background values of total columns collected within geophysically adjacent areas, assuming a small stratospheric BrO variability therein (Wagner et al., 2001; Hörmann et al., 2013, 2016). It conducts the separation efficiently without requiring auxiliary data, usually targeting a narrow domain to hold the assumption valid (Hörmann et al., 2016). Accurate representation of the background BrO columns in this approach requires preceding discrimination between areas with and without tropospheric enhancements (Hörmann et al., 2016). To facilitate global applications, this approach may need to be combined with a scheme for identifying tropospheric BrO enhancements.

In the third method (M3), the spatial distribution of stratospheric BrO columns is simulated using a chemical transport model (Begoin et al., 2010; Bougoudis et al., 2020; Choi et al., 2012, 2018; Theys et al., 2011; Toyota et al., 2011). Simulations with a detailed bromine chemistry scheme effectively reproduce the stratospheric BrO distribution. On the other hand, Sihler et al. (2012) pointed out that modeled data are potentially biased due to incomplete mechanisms and parameterizations. To remove the dependency on a model, the fourth method (M4) estimates stratospheric BrO columns using $O_3$ and nitrogen dioxide ($NO_2$) columns concurrently derived from the same satellite instrument (Herrmann et al., 2022; Sihler et al., 2012). This method robustly retrieves dynamic fields of stratospheric BrO columns using only observations without propagating errors from the auxiliary data. However, designed for retrievals over bright surfaces (e.g., the Arctic), this method assumes that tropospheric BrO molecules are uniformly distributed within a specific altitude range above the ground (e.g., 0–500 m) without relying on modeled profiles (Sihler et al., 2012). Global applications of the method may benefit from region-dependent variations.

### 2.5.2 Proposed method

To perform the STS, we adopt a scheme suggested by Bucsela et al. (2013) as a reference and apply adjustments, aggregating the physical bases behind M2, M3, and M4 described in Sect. 2.5.1. The reference scheme, developed for $NO_2$ retrievals from nadir-viewing satellite instruments, has been used to derive the OMI $NO_2$ standard product up to the most recent version (4.0) (Lamsal et al., 2021).

Bucsela et al. (2013) employed modeled $NO_2$ concentrations for the STS, similar to the M3 method designed for BrO (Begoin et al., 2010; Choi et al., 2012, 2018; Salawitch et al., 2010; Theys et al., 2009b, 2011; Toyota et al., 2011). The difference is that Bucsela et al. (2013) used the model data to construct an initial estimate of the tropospheric VCD field rather than the stratospheric. In other words, the reference scheme derived the stratospheric field from satellite retrievals, attributing the magnitudes of the retrieved total SCDs primarily to the stratospheric contribution. This approach is based on the fact that, for most of the Earth, the satellite-derived total $NO_2$ SCDs are almost entirely stratospheric (Bucsela et al., 2013). Since the same holds true for BrO, we apply this framework to the STS in this study.

The basic premise that the total SCDs are predominantly influenced by the stratosphere may not be applicable in areas where tropospheric contamination occurs. Accordingly, the reference scheme employed a masking technique to exclude satellite pixels potentially affected by high $NO_2$ pollution from the estimated stratospheric field, utilizing climatological tropospheric

NO$_2$ columns. The masked pixels accounted for up to 35% in the Northern Hemisphere when this technique was applied to OMI (Bucsela et al., 2013). Their stratospheric NO$_2$ columns were then estimated by spatial interpolation using values from neighboring unmasked areas. In this study, we suggest a different masking approach for BrO to effectively minimize the extent of the masked areas, leveraging the correlation between stratospheric BrO and O$_3$ concentrations.

The spatial correlation between stratospheric BrO and O$_3$ VCDs has been demonstrated by previous studies (Salawitch et al., 2010; Sihler et al., 2012; Theys et al., 2009b, 2011). This correlation suggests that positive anomalies in total BrO columns found within a consistent stratospheric O$_3$ VCD range can be attributed to tropospheric BrO enhancements. To be precise, stratospheric O$_3$ concentrations are correlated with those of stratospheric Br$_y$, and the proportions of BrO in the Br$_y$ group (i.e., the BrO/Br$_y$ ratios) are determined primarily by the stratospheric NO$_2$ chemistry (Lary, 1996; Choi et al., 2018;

Salawitch et al., 2010; Sihler et al., 2012; Theys et al., 2009b). On this basis, Sihler et al. (2012) identified tropospheric BrO enhancements using the ratio between total BrO and O$_3$ SCDs as a function of NO$_2$ VCD, SZA, and the viewing zenith angle (VZA). This approach is the M4 method described in Sect. 2.5.1 (Sihler et al., 2012; Herrmann et al., 2022).

   In this study, we pinpoint OMPS-NM pixels with tropospheric BrO contamination, i.e., "hotspots," by comparing the spatial distributions of the initial stratospheric BrO VCDs and the total O$_3$ VCDs. Removing only those hotspots from the stratospheric

BrO field enables minimizing the extent of masked areas. Here, the initial estimate of the stratospheric BrO field is derived by subtracting the model-based initial tropospheric SCDs from the total SCDs. To prevent the underestimation of stratospheric VCDs and ensure that all BrO hotspots appear in the initial stratospheric field, the initial tropospheric BrO SCDs must not exhibit enhancements ahead of the subtraction. For this purpose, we generate a second set of BrO vertical profiles devoid of tropospheric enhancements. Without additional modeling, we achieve this by simply smoothing out the vertical gradients of

tropospheric profiles from the CAM-Chem climatology. This empirical treatment of profiles is added to the STS scheme in this study, taking advantage of the fact that BrO has a lower probability of tropospheric enhancement than NO$_2$. This procedure is referred to as "flattening" hereafter.

   In short, the STS scheme for OMPS-NM BrO retrievals is conducted on an orbit-by-orbit basis in six steps:

   i.  flatten tropospheric BrO profiles from the CAM-Chem climatology and determine initial tropospheric SCDs;

ii.  subtract the initial tropospheric BrO SCDs from the total SCDs to derive an initial estimate of the stratospheric field;

   iii.  detect and mask BrO hotspots by comparing the spatial distributions of the initial stratospheric BrO VCDs and total O$_3$ VCDs;

   iv.  complete the stratospheric BrO field construction by filling the masked pixels and by horizontal smoothing;

   v.  derive the final tropospheric BrO field by subtracting the stratospheric SCDs from the total SCDs;

vi.  calculate the total BrO VCDs by summing the final stratospheric and tropospheric fields.

Detailed descriptions of the respective steps are presented in the following.

We perform the empirical flattening of the tropospheric profile for each OMPS-NM pixel, using co-located BrO volume mixing ratios (VMRs) obtained from the CAM-Chem climatology (step i). The flattening aims to generate a vertical profile exhibiting gradually decreasing (or constant) BrO VMRs from the tropopause toward the ground, representing background BrO conditions in the troposphere. For a given pixel, we first extract BrO VMR values below the tropopause determined by CAM-Chem. Then, in descending order of altitude, we recursively compare two adjacent VMRs and replace the larger value with the smaller one. The output of the flattening procedure is a boxcar-shaped tropospheric-background BrO profile. The flattening step is applied globally to each BrO profile allocated to every ground pixel, resulting in the initial estimates of tropospheric BrO VCDs. More details of the flattening, including the rationale behind it, can be found in Appendix B.

Figure 4 depicts examples of tropospheric BrO profiles before and after flattening. The two maps in Fig. 4a and d show BrO $\Delta$SCDs retrieved from orbits number 7594 (o7594) and 9756 (o9756) over Northern and Southern sea ice locations, respectively. The two orbits successfully captured the bromine explosions on 15 April 2013 (Northern sea ice locations) and 15 September 2013 (Southern sea ice locations). On visual inspection, pixels marked with red circles are suspected to be influenced by tropospheric enhancements (i.e., hotspots), while those with blue (cyan) circles are not. (These are confirmed by our hotspot detection scheme.) However, regardless of whether the given pixel is a hotspot or not, the modeled profile co-located with each of the four selected pixels exhibits tropospheric enhancement before flattening (Fig. 4b and e). It is not uncommon to encounter such a mismatch between (dynamic) satellite retrievals and (static) climatological profiles, especially when they possess different spatial resolutions. For the non-hotspots in Fig. 4a and d, subtracting tropospheric BrO columns based on the pre-flattening profiles (panels b and e) can lead to underestimation of the initial stratospheric columns. After flattening, on the other hand, all the resultant profiles are devoid of tropospheric enhancements as intended (Fig. 4c and f). The use of flattened profiles leads to the overestimation of the stratospheric columns at the hotspots, but these pixels are ultimately removed from the stratospheric field by masking (in step iii).

Another benefit of the flattening is that selective allocation becomes possible between the two sets of BrO vertical profiles for each OMPS-NM pixel to mitigate the mismatch between the satellite retrievals and the modeled profiles in the AMF calculations. For this purpose, our algorithm stores both pre- and post-flattening profiles for every pixel. If certain pixels are found to have tropospheric enhancements (in step iii), we apply the pre-flattening profiles for their AMF calculations. Ultimately, pre- and post-flattening profiles are used for hotspot and non-hotspot AMF calculations, respectively (in step v). For example, in Fig. 4, the red profiles in the middle panels (b and e) and the blue profiles in the right panels (c and f) are assigned to the hotspots and non-hotspots in the left panels (a and d), respectively.

Once the flattening step is complete for the given orbit, the initial stratospheric BrO VCD ($V_{strat}^0$) for each OMPS-NM pixel is derived using the flattened tropospheric profile (step ii):

$$V_{strat}^0 = \frac{S_{total} - V_{trop}^{flat} A_{trop}^{flat}}{A_{strat}},$$

(9)

where $V_{trop}^{flat}$ and $A_{trop}^{flat}$ represent the tropospheric VCD and AMF calculated with the flattened profile, respectively. Figure 5a presents a $V_{strat}^0$ field encompassing all 14 orbits on 13 March 2016 (o22667–o22680). Subtracting the post-flattening tropo-

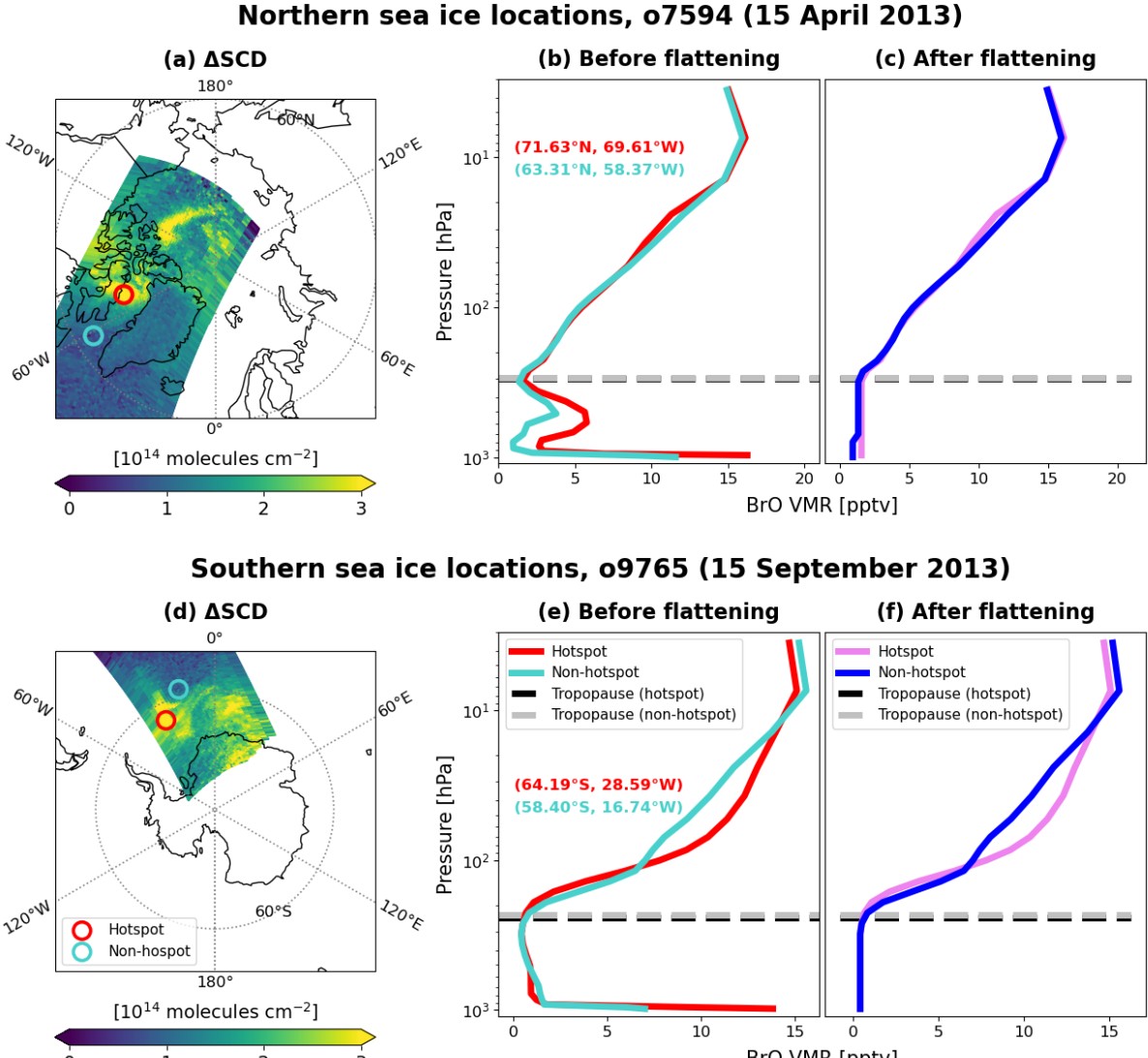

**Figure 4.** BrO profiles before and after flattening. Four OMPS-NM pixels are selected from sea ice locations in the Northern Hemisphere (o7594, 15 April 2013) and Southern Hemisphere (o9765, 15 September 2013). Panels (a) and (d) show the locations of these pixels overlaid on BrO ΔSCDs retrieved from the two orbits. Red and blue (cyan) circles on the maps represent hotspots and non-hotspots, respectively. Their BrO vertical profiles before flattening are presented in panels (b) and (e), using the same color code as in panels (a) and (d). The profiles after flattening are shown in panels (c) and (f). Tropopause pressures are indicated in black and gray dashed lines. The description of each line is shown in the legend. Latitudes and longitudes of the selected pixels are also indicated.

spheric columns allows for the propagation of the stratosphere-driven variabilities in total BrO columns to the initial stratospheric field with minimal spatial distortion. In other words, this method can capture the daily variations in the stratosphere.

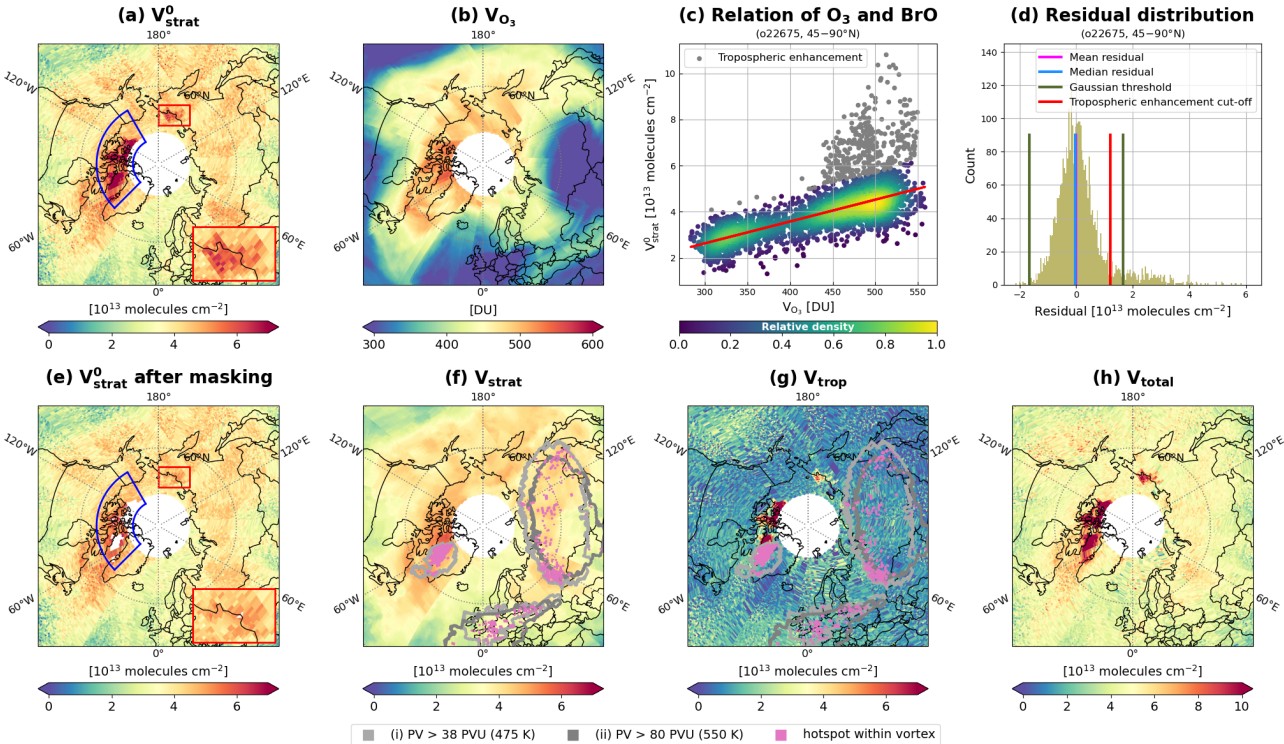

**Figure 5.** Description of the stratosphere-troposphere separation (STS) scheme in the OMPS-NM BrO retrieval algorithm. Intermediate quantities are presented for 13 March 2016. The panels represent (a) initial stratospheric BrO VCDs ($V_{strat}^0$); (b) total $O_3$ VCDs ($V_{O_3}$); (c) scatter plot of $V_{O_3}$ versus $V_{strat}^0$ for 45–90°N latitudes from o22675; (d) distribution of residuals from the linear regression shown in panel (c) (the description of the colored lines is shown in the legend); (e) $V_{strat}^0$ after hotspot masking (some masked pixels appear as if they are filled due to overlapping swaths); (f) final stratospheric BrO VCDs ($V_{strat}$); (g) tropospheric BrO VCDs ($V_{trop}$); and (h) total BrO VCDs ($V_{total}$). Note that a different color-bar range is used for $V_{total}$. In panels (f) and (g), gray curves represent areas within the polar vortex, while magenta pixels indicate hotspots within the vortex (see legend for details).

However, as expected, subtracting the post-flattening tropospheric columns results in tropospheric contamination of the initial stratospheric field. For example, the areas marked with the blue fan shape and red rectangles on the map in Fig. 5a have the potential for this type of contamination. Accordingly, the following step of STS is to mask the hotspots from the initial stratospheric BrO field (step iii). Masking should be carried out with caution because not all enhanced BrO VCDs are attributable to tropospheric contribution, as demonstrated by Salawitch et al. (2010). To differentiate actual hotspots from stratospheric BrO enhancements, we use total VCDs of $O_3$ derived for the same orbits, provided by the NASA OMPS-NM total $O_3$ product (OMPS-NPP_NMTO3-L2 Version 2.1) (Jaross, 2017a).

The total $O_3$ VCDs observed from OMPS-NM on 13 March 2016 are presented in Fig. 5b. The spatial distribution consistently corresponds with the initial stratospheric BrO VCDs (Fig. 5a). A quantitative analysis of their relationship is presented in Fig. 5c for the latitude range of 45–90°N from a single orbit (o22675). The scatter plot indicates two noticeable features simultaneously: (a) a strong linear relationship between $O_3$ and BrO VCDs, driven by stratospheric dynamics, and (b) pixels with large positive BrO anomalies contributed by tropospheric enhancements. Based on this finding, we define BrO hotspots as pixels with significant positive residuals from the linear regression between the total $O_3$ and the initial stratospheric BrO VCDs.

We derive the $O_3$-BrO relationship using an iterative approach, adopted from the M4 method (Sihler et al., 2012). In brief, we iteratively perform the linear regression under a consistent $BrO/Br_y$ condition, removing pixels with significant residuals. In collecting pixels with consistent $BrO/Br_y$ ratios, we constrain the latitude range. For each orbit, we derive the $O_3$-BrO relationship for every 45°-wide latitude bins (i.e., [90°S, 45°S], [45°S, 0°], [0°, 45°N], and [45°N, 90°N]).

In the presence of tropospheric BrO enhancements, the residual distribution from linear regression appears to be Gaussian but has a heavy tail in the positive direction (see the histogram in Fig. 5d). We assume that the linear regression would result in symmetric Gaussian residuals if derived using only the pixels free of tropospheric influence. This condition can be achieved by cropping the tails on both sides of the residual distribution while iteratively performing the regression. Here, we aim to extract this condition from each residual distribution for every 45°-wide latitude bin in each orbit to determine the stratosphere-driven $O_3$-BrO relationship. Once the Gaussian distribution is determined, pixels with residuals larger than the mean plus twice the standard deviation (outside the 95% confidence interval) are defined to have tropospheric BrO enhancements.

To crop the tails of a residual distribution, we use a threshold for the deviations from the mean value. The threshold is initially set to be the maximum deviation and is decreased by 10% iteratively until the cropped distribution becomes Gaussian. The linear regression is re-performed in each iteration, excluding the pixels outside the thresholds on both sides of the distribution. We determine whether the distribution is Gaussian using the asymmetry parameter $a_b$ (Sihler et al., 2012), defined for each latitude bin $b$ from each orbit:

$$a_b = |\frac{\bar{r}_b - \tilde{r}_b}{\sigma_b}|, \tag{10}$$

where $\bar{r}_b$, $\tilde{r}_b$, and $\sigma_b$ represent the mean, median, and standard deviation of the residuals, all of which are re-calculated in every iteration.

The iteration stops when either $a_b \leq 0.05$ or the maximum number of iterations (30 times) is reached. We find that 78.8% of the residual distributions from the entire study period already meet the condition of $a_b \leq 0.05$ even without cropping, while 10.4% (10.8%) of them require fewer than 10 iterations (10 iterations or more). Only 0.2% require 30 iterations or more. The red line in Fig. 5c indicates the result of the final linear regression. The histogram in Fig. 5d shows the distribution of the residuals from the final linear regression for the pixels shown in Fig. 5c. The two vertical green lines in Fig. 5d represent the final cropping thresholds. Once the iteration is terminated, we mask pixels with residuals larger than $\bar{r}_b + 2\sigma_b$ (the red vertical line in Fig. 5d). The gray dots in Fig. 5c show the masked pixels.

Figure 5e presents the $V_{\text{strat}}^0$ field on 13 March 2016 after the hotspot masking (outputs from step iii). As a result of masking, the areas within the blue fan shape and the red rectangle have missing pixels compared to Fig. 5a. It should be noted that some masked pixels appear as if they are filled in the figure due to overlapping swaths (as in the red rectangle). The relatively large stratospheric BrO VCDs remaining even after the masking in the blue fan shape supports that BrO enhancements occur not only in the troposphere but also in the stratosphere. It is worth noting that the total $O_3$ VCDs also appear to be enhanced in that area (Fig. 5b).

To complete the stratospheric BrO field construction (step iv), we first fill the masked pixels with the k-nearest neighbor (KNN) imputation (k=5) (Troyanskaya et al., 2001) using distances to neighbors as weighting factors. This gap-filling approach assumes that the stratospheric field is consistent within proximity, similar to the assumption made in the M2 method (Wagner et al., 2001; Hörmann et al., 2013, 2016). After filling in the masked pixels, we smooth the stratospheric field using the median filter. The final stratospheric BrO VCDs on 13 March 2016 are presented in Fig. 5f.

Once the final stratospheric VCD is derived for each pixel, it is used to determine the tropospheric VCD (step v):

$$V_{\text{trop}} = \frac{S_{\text{total}} - V_{\text{strat}} A_{\text{strat}}}{A_{\text{trop}}^{\text{select}}}, \tag{11}$$

where $V_{\text{strat}}$ and $V_{\text{trop}}$ represent the stratospheric and tropospheric VCDs, respectively. The variable $A_{\text{trop}}^{\text{select}}$ denotes the selected AMF. If the given pixel is a hotspot, we use the AMF calculated using the pre-flattening profile ($A_{\text{trop}}$); otherwise, we use the AMF calculated with the flattened profile ($A_{\text{trop}}^{\text{flat}}$). The $V_{\text{trop}}$ field on 13 March 2016 is shown in Fig. 5g. The pixels defined as hotspots in the stratospheric field show particularly high values.

For the two latitude bins that cover the northern and southern polar regions ([90°S, 45°S] and [45°N, 90°N]), the $O_3$-BrO relationships can be altered inside the polar vortex and under ozone hole conditions (Sihler et al., 2012). In these cases, our scheme may lead to an overdetection (or underdetection) of hotspots while still preserving the overall spatial pattern of the stratospheric field determined in step ii. Given the lower reliability of hotspot detection in the polar vortex, we introduce quality flags specifically designed for STS, represented by three-digit binary values. The first digit indicates whether a hotspot is detected, while the second and third digits denote whether the potential vorticity exceeds a threshold at potential temperatures of 475 and 550 K, respectively. The thresholds are 38 potential vorticity units (PVU) (475 K) and 80 PVU (550 K) in the Northern Hemisphere, while those are –55 PVU (475 K) and –90 PVU (550 K) in the Southern Hemisphere. For this purpose, we use potential vorticity data from MERRA-2 at 0.5° latitude × 0.625° longitude resolution (GMAO, 2015b). Gray curves in Fig. 5f and g indicate areas within the polar vortex at 475 and 550 K potential temperatures. Pink pixels represent hotspots detected within the vortex. OMPS-NM BrO data users can filter out polar vortex hotspots using the STS quality flags based on their specific analyses and requirements. However, for the purposes of our analyses in this study, we do not apply the STS quality flags to present the general retrieval performance.

Lastly, the total VCD at each pixel is calculated by the sum of the stratospheric and the tropospheric VCDs (step vi):

$$V_{\text{total}} = V_{\text{strat}} + V_{\text{trop}}. \tag{12}$$

Figure 5h presents the total BrO VCD ($V_{\text{total}}$) field on 13 March 2016. Around the North Pole (at latitudes $> 60°$N), the total BrO field shows stronger spatial variations than the total $O_3$ field (Fig. 5b), due to tropospheric enhancements. More consistent spatial patterns are found between the two species at lower latitudes, mainly due to stratospheric dynamics.

## 2.6 Uncertainty estimation

BrO VCDs retrieved from OMPS-NM have both random and systematic errors. Here, we define the term "error" as the absolute deviation of a retrieved value from the (unknown) truth. Errors are assumed to have Gaussian distributions. We use the term "uncertainty" to refer to the Gaussian error distributions; specifically, standard deviations and mean values (i.e. biases) are referred to as random and systematic uncertainties, respectively (von Clarmann et al., 2020).

To estimate the random uncertainties, we conduct a Gaussian error propagation, assuming that random errors in different
parameters are uncorrelated and independent of one another. The median absolute deviation (MAD) is used instead of the standard deviation when representing the uncertainty of a median value. For each OMPS-NM pixel, we estimate random uncertainties in SCDs, AMFs, and VCDs following the approaches described separately in Sects. 2.6.1–2.6.3. Specific statistics of uncertainties are presented for January, April, July, and October 2018, even though uncertainties are estimated for the entire study period.

Estimation of the systematic uncertainties is hindered by the limited knowledge of the input parameter biases. We discuss systematic uncertainties and possible contributing factors in Sect. 3 while describing the intercomparison between OMPS-NM and ground-based BrO retrievals.

### 2.6.1 Slant columns

The random uncertainty in a total BrO SCD at each OMPS-NM pixel ($\varepsilon_S$) can be estimated by

$$20 \quad \varepsilon_S^2 = \varepsilon_\Delta^2 + \varepsilon_R^2 + \varepsilon_B^2, \quad (13)$$

where $\varepsilon_\Delta$, $\varepsilon_R$, and $\varepsilon_B$ represent the random uncertainty in $\Delta$SCD ($\Delta S$), background SCD ($S_R$), and bias correction term ($S_B$), respectively. Each uncertainty term is estimated as described in the following.

To calculate $\varepsilon_\Delta$, we assume that the fitting residuals are dominated by the spectrally uncorrelated measurement noise (Chan Miller et al., 2014; González Abad et al., 2015, 2016). The random error covariance of $\hat{\boldsymbol{x}}$ in Eq. (1) can then be es-
25 timated by

$$\boldsymbol{S}_x^\epsilon = \epsilon_{\text{rms}}^2 \left( \frac{m}{m-n} \right) (\mathbf{K}_x^{\mathrm{T}} \mathbf{K}_x)^{-1}, \quad (14)$$

where $\epsilon_{\text{rms}}$ denotes the fitting RMSE, $m$ is the number of spectral points in the fitting window, and $n$ is the number of parameters fitted in the BrO retrieval. The diagonal elements of $\boldsymbol{S}_x^\epsilon$ represent squared random uncertainties of the retrieved states. Therefore, the square root of the diagonal element in the BrO row corresponds to the random uncertainty of BrO $\Delta$SCD (i.e.,
$\varepsilon_\Delta = \sqrt{S_{\text{BrO}}^\epsilon}$). Figure 6a–d shows the distributions of the $\varepsilon_\Delta$ values for every OMPS-NM orbit in January, April, July, and October 2018, respectively. The median absolute uncertainty is $\sim 1.8 \times 10^{13}$ molecules cm$^{-2}$ for each month.

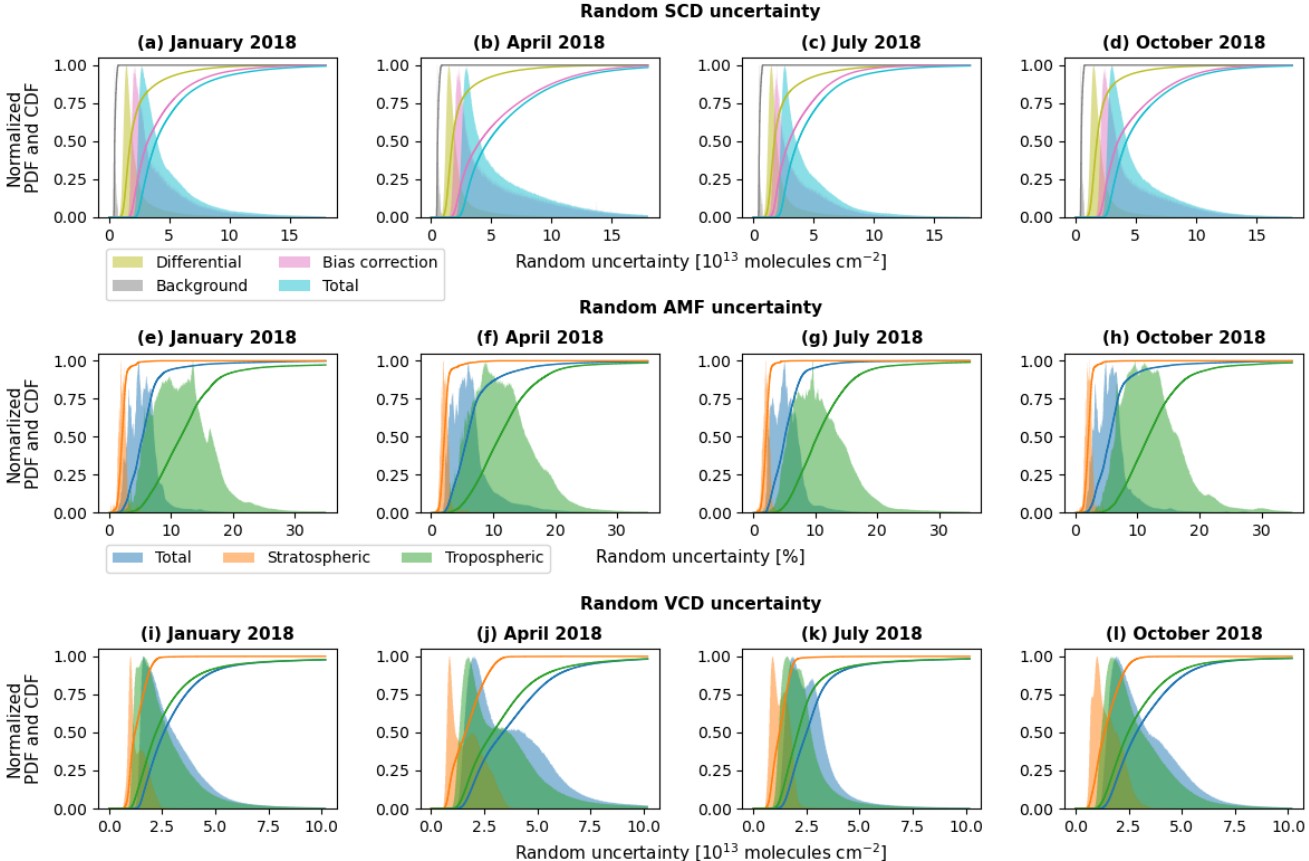

**Figure 6.** Normalized probability density functions (PDFs, shades) and cumulative distribution functions (CDFs, curves) of random uncertainties in (a–d) BrO SCDs, (e–h) AMFs, and (i–l) BrO VCDs. Columns from the left to right are for January, April, July, and October 2018. The colors indicated in the legends denote different error source terms. Absolute uncertainties are presented for BrO SCDs and VCDs, while relative uncertainties are presented for AMFs.

As described in Sect. 2.4, the background SCD is determined from the median of the modeled total SCDs in the reference sector for each cross-track position. Therefore, its random uncertainty ($\varepsilon_R$) has a component associated with the natural variability in the modeled total SCDs within the sector and is represented by the MAD. Another component of $\varepsilon_R$ is the random uncertainties of the total AMFs in the reference sector, as the modeled total SCDs are determined by the products of the total AMFs and the modeled total VCDs. The estimation of random AMF uncertainties is described in Sect. 2.6.2. Combining these contributing factors, we estimate $\varepsilon_R$ for each cross-track position. The estimated uncertainties for the OMPS-NM orbits in January, April, July, and October 2018 are presented in Fig. 6a–d. Notably, the background SCDs have the smallest absolute uncertainties among the three SCD components ($\Delta S$, $S_R$, and $S_B$) with the medians of 0.5, 0.6, 0.5, and $0.5\times10^{13}$ molecules cm$^{-2}$ for January, April, July, and October 2018, respectively.

The bias correction term is calculated by comparing two polynomials fitted to the background-corrected SCDs and the modeled SCDs (Sect. 2.4). Therefore, its random uncertainty ($\varepsilon_B$) is introduced by random uncertainties in the polynomial coefficients, which are associated with natural variabilities in SCDs. Additionally, $\varepsilon_B$ is contributed by the random uncertainties in the total AMFs, which are used to determine the modeled SCDs. Lastly, the random $\Delta$SCD uncertainty also contributes to $\varepsilon_B$, since the calculation of the background-corrected SCD involves $\Delta$SCD (Sect. 2.4). By propagating these uncertainties, we estimate $\varepsilon_B$ pixel by pixel. Figure 6a–d presents the $\varepsilon_B$ values from the OMPS-NM pixels in January, April, July, and October 2018. The figure shows that $\varepsilon_B$ contributes most to the total SCD uncertainty. The median uncertainties for January, April, July, and October 2018 are 3.2, 4.1, 3.0, and $3.6 \times 10^{13}$ molecules cm$^{-2}$, respectively.

The random uncertainties in total SCDs ($\varepsilon_S$), estimated according to Eq. (13), are presented in Fig. 6a–d for January, April, July, and October 2018. The median absolute uncertainties are 3.9, 4.8, 3.7, and $4.3 \times 10^{13}$ molecules cm$^{-2}$, respectively. Dividing the random uncertainty by the total SCD pixel by pixel, we estimate that the median percentage errors are 49.3%, 53.2%, 52.9%, and 57.0% for January, April, July, and October 2018, respectively.

### 2.6.2 Air mass factors

Assuming that the components do not correlate, we estimate the random AMF uncertainty for each OMPS-NM pixel by

$$\varepsilon_{A,z\in\{\text{total, strat, trop, flat}\}}^2 = (\varepsilon_{A,z}^{\text{SF}})^2 + \left(\frac{\partial A_z}{\partial r}\right)^2 \varepsilon_r^2 + \left(\frac{\partial A_z}{\partial c_{\text{eff}}}\right)^2 \varepsilon_c^2 + \left(\frac{\partial A_z}{\partial P_c}\right)^2 \varepsilon_P^2, \tag{15}$$

where $\varepsilon_{A,z\in\{\text{total, strat, trop, flat}\}}$ represents the random uncertainty in either the total, stratospheric, pre-flattening tropospheric, or post-flattening tropospheric AMF. The variables $r$, $c_{\text{eff}}$, and $P_c$ denote the surface reflectance, ECF, and cloud pressure, whose uncertainties correspond to $\varepsilon_r$, $\varepsilon_c$, and $\varepsilon_P$, respectively. The term $\varepsilon_{A,z}^{\text{SF}}$ represents the random uncertainty introduced by errors in the BrO shape factor. Estimation of random AMF uncertainties involves look-up tables (LUTs) for variables $\varepsilon_{A,z}^{\text{SF}}$, $\frac{\partial A_z}{\partial r}$, $\frac{\partial A_z}{\partial c_{\text{eff}}}$, and $\frac{\partial A_z}{\partial P_c}$ in Eq. (15), constructed separately for the four different types of AMFs (total, stratospheric, pre-flattening tropospheric, and post-flattening tropospheric). Detailed descriptions of the approach are provided below.

We determine the $\varepsilon_{A,z}^{\text{SF}}$ term by devising a method that employs the k-means clustering (Lloyd, 1982), instead of applying a partial derivative by parameterizing the vertical profiles (e.g., De Smedt et al., 2018). In brief, we classify OMPS-NM pixels into several clusters based on the shapes of co-located BrO profiles, and then we estimate $\varepsilon_{A,z}^{\text{SF}}$ by the standard deviation of AMFs for each cluster. The objective is to evaluate how AMFs respond to variations in input profiles within a defined range. This approach is devised as a simple and empirical alternative to an ideal method, which involves the execution of ensemble model simulations with various initialization/realization settings, aiming to explore the magnitude of the resulting changes in AMFs.

The k-means clustering in this study operates on the monthly global CAM-Chem BrO profiles sampled for the OMPS-NM overpass times. Clustering is performed using the pre-flattening (original) profiles from the 26 CAM-Chem layers that cover the vertical range from the surface up to $\sim$3 hPa. The main output of the k-means algorithm is a set of profile centroids, one for each cluster. Here, the centroid refers to a single vertical profile that represents the shapes of all the profiles in the cluster. Another algorithm output is the distortion, defined as the sum of the squared distances between each sample and its dominating

**Table 3.** Distinctive features of the four vertical profile clusters.

| Cluster index | Tropospheric BrO enhancement | BrO gradient toward the stratosphere | Tropopause height | Typical regions of occurrence |
| --- | --- | --- | --- | --- |
| 1 | Yes | Moderate | Moderate | Polar regions (60–90°S or 60–90°N) |
| 2 | No | Moderate | Moderate | Midlatitudes (30–60°S or 30–60°N) |
| 3 | No | Low | High | Tropics (30°S–30°N) |
| 4 | No | High | Moderate | Polar regions (60–90°S or 60–90°N) |

centroid. We use four clusters to classify all the CAM-Chem BrO profiles (i.e., k=4), as they result in sufficiently low distortion. Figure 7a shows the four vertical profile centroids resulting from the clustering. The four centroids are distinguishable in terms of (a) whether it has a tropospheric BrO enhancement, (b) the steepness of BrO gradient toward the stratosphere, (c) tropopause height, and (d) typical regions of occurrence. The distinctive features of each cluster are summarized in Table 3.

Based on the clustering results, we assign a cluster index of 1 to 4 to each OMPS-NM pixel by finding the centroid closest to its profile. Figure 7b–c presents the results of assigning the cluster indices to the pixels on 15 April and 16 September 2018. These examples demonstrate that profile shapes are strongly dependent on latitudes, as summarized in Table 3. It is noticeable that green pixels (with cluster index 1) are concentrated around the North Pole and the South Pole in Fig. 7b and c, respectively. Given that the corresponding profile centroid has a tropospheric enhancement (Fig. 7a), the spatial distributions of these pixels

reflect the ground-level BrO production in the Arctic and Antarctic in the respective spring seasons. The green pixels over the tropical North Atlantic Ocean in Fig. 7b correspond to the areas where ground- and ship-based observations have detected high surface BrO concentrations (Leser et al., 2003; Mahajan et al., 2010; Martin et al., 2009; Read et al., 2008; Sander et al., 2003). These elevated concentrations are linked to the rapid debromination of sea salt aerosols contributed by the outflow of nitric acid and sulfur dioxide from the nearby continent (Wang et al., 2021). Overall, the four vertical profile clusters are able

to represent the sub-hemispherical-scale variabilities in the global monthly BrO profiles.

      Using one year of AMF data produced for 2015, we construct an LUT of $\varepsilon_{A,z}^{\mathrm{SF}}$. The AMFs are first binned according to the following six parameters: (a) BrO profile cluster index, (b) geometric AMF, (c) surface type, (d) surface reflectance, (e) cloud fraction, and (f) cloud pressure. Here, the geometric AMF is defined as the sum of the secant of solar and viewing zenith angles. For a simpler parameterization of surface reflectance, we convert the BRDF parameters to geometry-dependent surface

Lambertian equivalent reflectivity (GLER) by matching the radiances simulated by VLIDORT with the BRDF and LER options (Fasnacht et al., 2019; Qin et al., 2017; Vasilkov et al., 2017). The surface types include land, water, and glint (the incident angle for specular reflection < 30°). The center and width of each bin, which are ultimately used as the node and interval of the LUT, are presented in Table 4. After binning, the standard deviation of the AMFs (i.e. $\varepsilon_{A,z}^{\mathrm{SF}}$) is calculated for each bin.

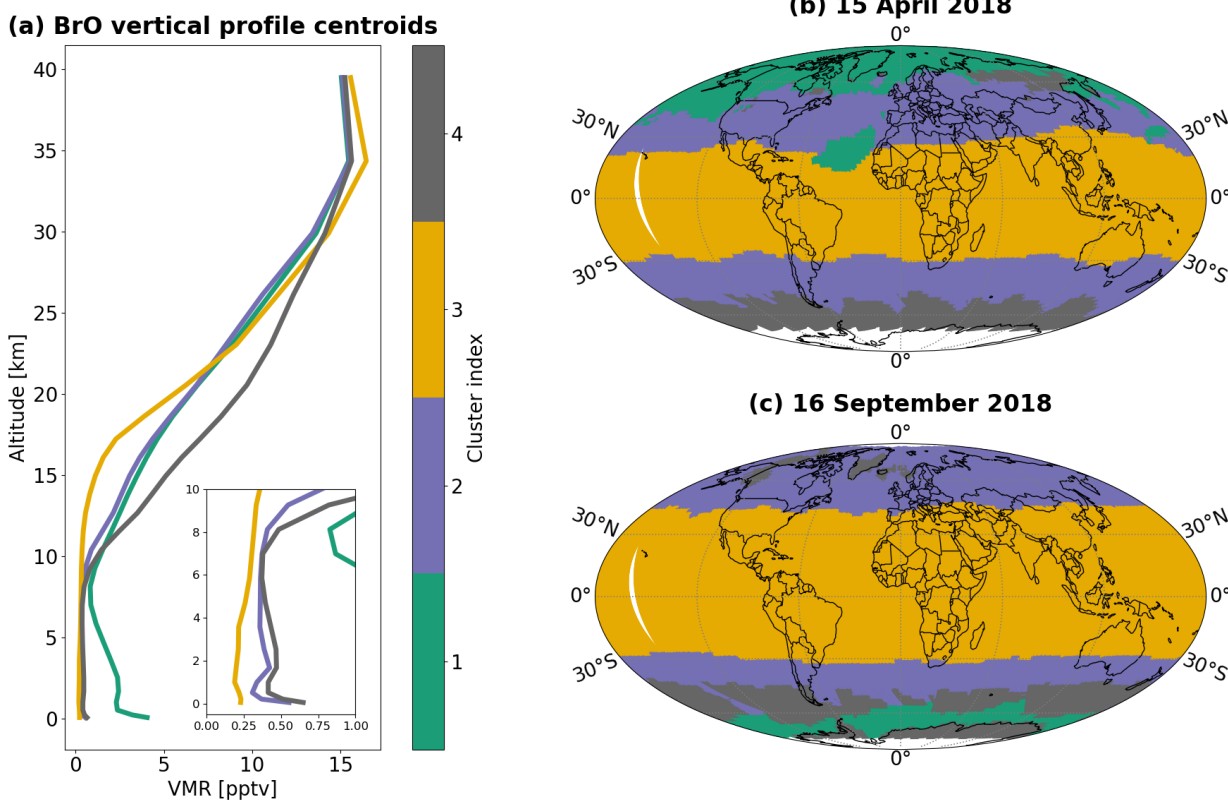

**Figure 7.** Results of the k-means clustering for CAM-Chem BrO vertical profiles. Panel (a) shows four vertical profile centroids obtained from the clustering. Each cluster is indexed and colored (see the color bar). Panels (b–c) show the results of assigning the cluster indices to the OMPS-NM pixels on 15 April and 16 September 2018.

**Table 4.** Nodes and intervals of the look-up tables for $\varepsilon_{A,z}^{\mathrm{SF}}$ (random AMF uncertainty introduced by errors in BrO shape factor), $\frac{\partial A_z}{\partial r}$ (partial derivative of AMF with respect to surface reflectance), $\frac{\partial A_z}{\partial c}$ (partial derivative of AMF with respect to cloud fraction), and $\frac{\partial A_z}{\partial P}$ (partial derivative of AMF with respect to cloud pressure).

| Parameter | Number of nodes | Nodes |
|---|---|---|
| Surface type | 3 | Land, water, glint |
| Geometric AMF | 41 | 2.1–10.1 with 0.2 interval |
| Vertical profile cluster | 4 | 1, 2, 3, 4 |
| Surface reflectance (GLER) | 20 | 0.025–0.975 with 0.05 interval |
| Cloud fraction | 10 | 0.05–0.95 with 0.1 interval |
| Cloud pressure | 7 | 175–1075 hPa with 150 hPa interval |

The AMF bins are used to construct not only the LUT for $\varepsilon_{A,z}^{\mathrm{SF}}$ but also the LUTs for the partial derivatives in Eq. (15). To construct the partial derivative LUTs, we first calculate the AMF averages for the respective bins. Then, by calculating the gradients of the average AMFs between adjacent bins for each parameter, we derive the partial derivatives with respect to surface reflectance ($\frac{\partial A_z}{\partial r}$), cloud fraction ($\frac{\partial A_z}{\partial c_{\mathrm{eff}}}$), and cloud pressure ($\frac{\partial A_z}{\partial P_c}$). The results are assigned to the nodes in Table 4. As mentioned earlier, this approach is applied to each of the four types of AMFs (i.e., $A_{\mathrm{total}}$, $A_{\mathrm{strat}}$, $A_{\mathrm{trop}}$, and $A_{\mathrm{trop}}^{\mathrm{flat}}$). In this process, the cluster indices derived using the pre-flattening profiles from the 26 CAM-Chem layers are fixed regardless of the AMF type. Consequently, a total of four LUTs ($\varepsilon_{A,z}^{\mathrm{SF}}$, $\frac{\partial A_z}{\partial r}$, $\frac{\partial A_z}{\partial c_{\mathrm{eff}}}$, and $\frac{\partial A_z}{\partial P_c}$) are constructed for each type of AMF.

To determine the terms $\varepsilon_r$, $\varepsilon_c$, and $\varepsilon_P$ in Eq. (15), we employ estimates from previous studies. For the random uncertainty in surface reflectance ($\varepsilon_r$), which varies depending on the surface type, we assume that the random errors in the GLERs derived in this study are equivalent to those in the albedos from the MCD43 product (Wang et al., 2018; Wu et al., 2018). However, the uncertainties in these albedo values, retrieved with the BRDF parameters from MODIS, cover only land pixels. Over water bodies, we employ the RMSE values from the comparison between OMI-derived GLERs and LERs over the deep ocean (Fasnacht et al., 2019). The estimates of $\varepsilon_r$ values used in this study are further described in Appendix C. Random uncertainties in cloud fraction ($\varepsilon_c$) and cloud pressure ($\varepsilon_P$) are adopted as 0.084 and 46.2 hPa, based on previous assessments of the RRS cloud retrievals (Stammes et al., 2008; Vasilkov et al., 2014).

The constructed LUTs are applied to each OMPS-NM pixel to estimate $\varepsilon_{A,\mathrm{total}}$, $\varepsilon_{A,\mathrm{strat}}$, $\varepsilon_{A,\mathrm{trop}}$, and $\varepsilon_{A,\mathrm{flat}}$ according to Eq. (15). Then, the random uncertainty in $A_{\mathrm{trop}}^{\mathrm{select}}$ ($\varepsilon_{A,\mathrm{select}}$) is determined by assigning either $\varepsilon_{A,\mathrm{trop}}$ or $\varepsilon_{A,\mathrm{flat}}$, depending on whether the pixel in question has a tropospheric BrO enhancement. Figure 6e–h shows $\varepsilon_{A,\mathrm{total}}$, $\varepsilon_{A,\mathrm{strat}}$, and $\varepsilon_{A,\mathrm{select}}$ from every OMPS-NM orbit in January, April, July, and October 2018. Unlike the SCD uncertainties, the percentage values for the AMF uncertainties are presented in Fig. 6. The stratospheric AMFs typically show the smallest percentage uncertainties, with medians of 2.2%, 2.2%, 2.1%, and 2.1%, respectively. The tropospheric AMF uncertainties have the largest medians and the widest distribution. The medians for the respective months are 11.6%, 11.1%, 10.4%, and 11.8%. The median values of the total AMF uncertainties for the respective months are 5.5%, 5.9%, 5.2%, and 5.6%.

### 2.6.3 Vertical columns

The random uncertainties in stratospheric, tropospheric, and total BrO VCDs are estimated by applying the Gaussian error propagation to Eqs. (9), (11), and (12), respectively. We assume that $V_{\mathrm{strat}}$ has the same random uncertainty as $V_{\mathrm{strat}}^0$, determined by

$$\varepsilon_{V,\mathrm{strat}}^2 = \left(\frac{1}{A_{\mathrm{strat}}}\right)^2 \varepsilon_S^2 + \left(\frac{A_{\mathrm{trop}}^{\mathrm{flat}}}{A_{\mathrm{strat}}}\right)^2 \varepsilon_{V,\mathrm{flat}}^2 + \left(\frac{V_{\mathrm{trop}}^{\mathrm{flat}}}{A_{\mathrm{strat}}}\right)^2 \varepsilon_{A,\mathrm{flat}}^2 + \left(\frac{S_{\mathrm{total}} - V_{\mathrm{trop}}^{\mathrm{flat}} A_{\mathrm{trop}}^{\mathrm{flat}}}{A_{\mathrm{strat}}^2}\right)^2 \varepsilon_{A,\mathrm{strat}}^2, \tag{16}$$

where $\varepsilon_{V,\mathrm{strat}}$, $\varepsilon_{V,\mathrm{flat}}$, $\varepsilon_{A,\mathrm{flat}}$, and $\varepsilon_{A,\mathrm{strat}}$ represent the random uncertainties in $V_{\mathrm{strat}}$, $V_{\mathrm{trop}}^{\mathrm{flat}}$, $A_{\mathrm{trop}}^{\mathrm{flat}}$, and $A_{\mathrm{strat}}$, respectively. The term $\varepsilon_{V,\mathrm{flat}}$ is estimated by calculating the standard deviation of $V_{\mathrm{trop}}^{\mathrm{flat}}$ values for each profile cluster. Once $\varepsilon_{V,\mathrm{strat}}$ is determined for a given OMPS-NM pixel, we estimate the random uncertainty in $V_{\mathrm{trop}}$ by

$$\varepsilon_{V,\mathrm{trop}}^2 = \left(\frac{1}{A_{\mathrm{trop}}^{\mathrm{select}}}\right)^2 \varepsilon_S^2 + \left(\frac{A_{\mathrm{strat}}}{A_{\mathrm{trop}}^{\mathrm{select}}}\right)^2 \varepsilon_{V,\mathrm{strat}}^2 + \left(\frac{V_{\mathrm{strat}}}{A_{\mathrm{trop}}^{\mathrm{select}}}\right)^2 \varepsilon_{A,\mathrm{strat}}^2 + \left(\frac{S_{\mathrm{total}} - V_{\mathrm{strat}} A_{\mathrm{strat}}}{(A_{\mathrm{trop}}^{\mathrm{select}})^2}\right)^2 \varepsilon_{A,\mathrm{select}}^2, \tag{17}$$

where $\varepsilon_{V,\text{trop}}$ and $\varepsilon_{A,\text{select}}$ denote the random uncertainties in $V_{\text{trop}}$ and $A_{\text{trop}}^{\text{select}}$, respectively. Lastly, the random uncertainty in $V_{\text{total}}$ is determined by

$$\varepsilon_{V,\text{total}}^2 = \varepsilon_{V,\text{strat}}^2 + \varepsilon_{V,\text{trop}}^2. \tag{18}$$

Figure 6i–l shows the random uncertainties in stratospheric, tropospheric, and total BrO VCDs in January, April, July, and October 2018. The stratospheric uncertainties have the medians of 1.2, 1.6, 1.3, and $1.4 \times 10^{13}$ molecules cm$^{-2}$ in the respective months. The distributions of the tropospheric uncertainties have heavier tails in the positive direction than the stratospheric uncertainties, and their medians are 2.2, 2.9, 2.1, and $2.5 \times 10^{13}$ molecules cm$^{-2}$, respectively. The total VCD uncertainties for the respective months have medians of 2.6, 3.4, 2.5, and $2.9 \times 10^{13}$ molecules cm$^{-2}$.

## 3 Intercomparison with ground-based observations

We assess stratospheric and tropospheric BrO VCDs retrieved from OMPS-NM by intercomparison with ground-based retrievals. Reference ground stations are Lauder, New Zealand (Querel et al., 2021), Utqiaġvik (Barrow), Alaska (Simpson, 2018), and Harestua, Norway (Hendrick et al., 2007), covering both the Northern and Southern Hemispheres. Lauder provides stratospheric VCD, Utqiaġvik provides tropospheric VCD, and Harestua provides both.

The intercomparison is performed using daily and monthly mean data. The monthly averages are calculated only for months with more than three data points. For spatial co-location, we average OMPS-NM retrievals within a 0.5° radius from each ground station. Here, we use only OMPS-NM retrievals with cloud fractions ≤ 0.5, SZAs ≤ 80°, "good" quality flags, and cross-track positions from 1 to 34 (0-based). Temporal co-location is carried out with different criteria depending on the station due to the different sampling approaches (Table 5). Since data from Lauder are unavailable at the OMPS-NM overpass times, we use ground-based BrO VCDs observed at 80° SZA in the morning, neglecting any diurnal variation. This choice is based on our calculation that the difference between the nominal OMPS-NM overpass time (13:30 LST) and the average ground-based observation time for 80° SZA in Lauder is slightly smaller in the morning (~5.1 hours) than in the evening (~5.5 hours). For the Utqiaġvik and Harestua stations, we average the ground-based observations within 100 min before and after each OMPS-NM observation.

Located at 71.3°N latitude, the instrument at the Utqiaġvik station can observe Arctic tropospheric BrO enhancements in spring (Simpson et al., 2017). Figure 8 presents the intercomparison between the daily tropospheric BrO VCDs from OMPS-NM and the ground-based instrument at the Utqiaġvik station in 2012 and 2013. The time series shows that the OMPS-NM BrO VCDs vary consistently with the ground-based observations. We present the OMPS-NM retrievals on the maps for four selected dates when both OMPS-NM and the ground-based instrument observed large VCDs (see red circles in the time series). The OMPS-NM retrievals reveal a large BrO plume stretching over the Utqiaġvik station on each occasion. These examples demonstrate that the OMPS-NM retrievals can provide a broad perspective for the interpretation of ground-based BrO observations.

Figure 9 shows scatter plots of ground-based versus OMPS-NM BrO retrievals from all stations with the regression lines derived by the least squares linear fit. As expected, the monthly mean VCDs show higher correlation coefficients ($r$) than the

**Table 5.** Specifications of ground-based BrO retrievals used for the intercomparison with OMPS-NM retrievals.

| Station | Location | Instrument | BrO columns observed | Temporal coverage | Temporal sampling | Note | Reference |
|---------|----------|------------|----------------------|-------------------|-------------------|------|-----------|
| Lauder | 45.0°S, 169.7°E | Zenith-sky DOAS | Stratospheric | February 2012– July 2021 | 1° SZA | Provided for SZA $\geq 75°$ | Querel et al. (2021) |
| Utqiaġvik | 71.3°N, 156.7°W | MAX-DOAS | Lower-tropospheric (< 4 km) | March 2012– June 2016 | 1 hour | Provided for February–June every year | Simpson (2018) |
| Harestua | 60.2°N, 10.8°E | Zenith-sky DOAS | Stratospheric and tropospheric | February 2013– July 2021 | 1 day | Photochemically converted to 13:30 LST | Hendrick et al. (2007) |

daily VCDs at every station due to the reduced random errors led by increased numbers of averaged samples. At the Lauder and Harestua stations, the monthly stratospheric BrO VCDs show high $r$ values exceeding 0.8. Meanwhile, the monthly tropospheric VCDs show $r = 0.70$ in Utqiaġvik and $r = 0.50$ in Harestua. Overall, the stratospheric and tropospheric BrO retrievals from OMPS-NM demonstrate reliable performance, and the stratospheric VCDs show better agreements with ground-based observations. It should be noted, however, that accounting for the BrO diurnal variation in Lauder, e.g., using photochemical conversion as in Harestua (Hendrick et al., 2007), may change the intercomparison result.

Although ground-based observations also have errors, the scatter plots in Fig. 9 enable the estimation of systematic uncertainties in OMPS-NM BrO retrievals. We use the mean bias error (MBE) as an indicator. The stratospheric BrO VCDs at the Lauder and Harestua stations both show negative MBEs, albeit with different magnitudes. Daily stratospheric VCDs from OMPS-NM are more biased in Harestua (MBE = $-0.75\times10^{13}$ molecules cm$^{-2}$) than in Lauder (MBE = $-0.05\times10^{13}$ molecules cm$^{-2}$). In comparison, the biases in the tropospheric BrO VCDs show different patterns (signs) in Utqiaġvik and Harestua. Daily OMPS-NM retrievals are higher than ground-based retrievals in Utqiaġvik by $0.09\times10^{13}$ molecules cm$^{-2}$ on average. In addition to the systematic uncertainties in the OMPS-NM BrO retrievals, the difference in the vertical coverages can contribute to the discrepancies in Utqiaġvik (the entire troposphere versus $\sim$4 km; see Table 5). In Harestua, the daily tropospheric BrO VCDs from OMPS-NM are lower by $0.55\times10^{13}$ molecules cm$^{-2}$ on average.

The time series in Fig. 10 show temporal changes in the monthly averages of the OMPS-NM and ground-based BrO retrievals. We examine the monthly dependence of systematic errors in the OMPS-NM retrievals by comparing the time series. In Utqiaġvik, the OMPS-NM retrievals typically show peaks in April, which are not always supported by the ground-based observations (Fig. 10b). At the Harestua station, the agreements between OMPS-NM and ground-based retrievals are better in spring than in other seasons (Fig. 10c and d). This seasonal dependence in Harestua is especially prominent for the tropospheric VCDs (Fig. 10d). Indeed, the monthly retrievals in spring appear in the scatter plot as a cluster of data points close to the identity line, distinct from others (Fig. 9d). Biases in the OMPS-NM retrievals at the Lauder station show relatively weak seasonal dependence (Fig. 10a).

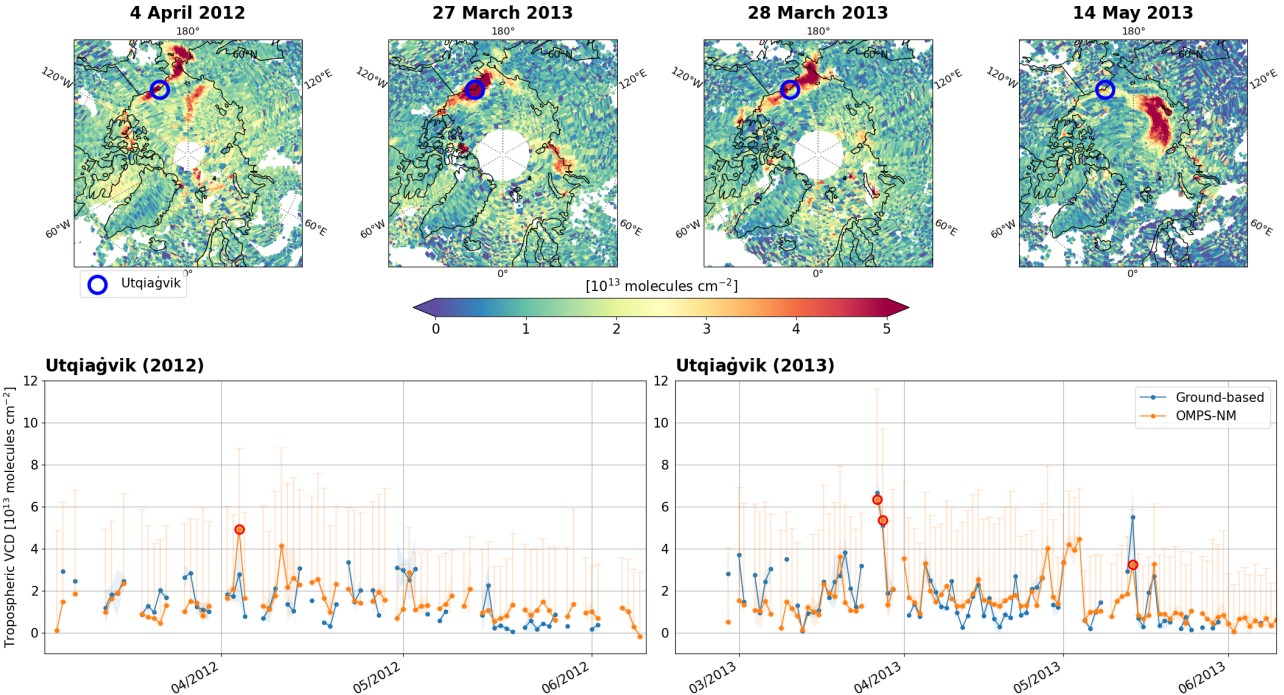

**Figure 8.** OMPS-NM and ground-based BrO retrievals at the Utqiaġvik station. The OMPS-NM retrievals target the entire troposphere, while the ground-based target the lower troposphere (< 4 km). The time series show daily retrievals in February–June 2012 and 2013. The shades represent standard deviations of the data averaged for the spatial and temporal co-location. The error bars indicate estimated uncertainties in the OMPS-NM tropospheric BrO VCDs. The lower error bars are omitted for display purposes. The red circles indicate four dates selected for large BrO VCDs from both OMPS-NM and ground-based retrievals. OMPS-NM retrievals for the selected dates are presented on the maps with the location of the Utqiaġvik station indicated with blue circles.

Systematic errors in OMPS-NM retrievals result from both SCD and AMF uncertainties. Contributors to systematic SCD uncertainties include errors in reference spectra, SRF characterization, instrument calibration, and fitting window and fitting parameters configuration. All of these combine to appear as SCD biases, which are analyzed for different fitting windows in Appendix A. Systematic AMF uncertainties are contributed by errors in radiative transfer calculations and input parameters, as

5   well as BrO profile shapes. The impacts of BrO profile shape uncertainties on tropospheric AMF calculations differ between non-hotspots and hotspots due to the use of flattened profiles for non-hotspots and climatology profiles for hotspots. This selective profile allocation assumes that climatology can adequately represent profile shapes for tropospheric enhancement events; however, systematic errors may occur when this assumption is not satisfied. In addition, systematic errors in BrO VCD retrievals result from uncertainties in the initial estimates of tropospheric VCDs ($V_{\mathrm{trop}}^{\mathrm{flat}}$) in the STS scheme. The $V_{\mathrm{trop}}^{\mathrm{flat}}$ fields are

10  intended to represent background conditions, and this study proposes a flattening technique to achieve this representation. Nev-

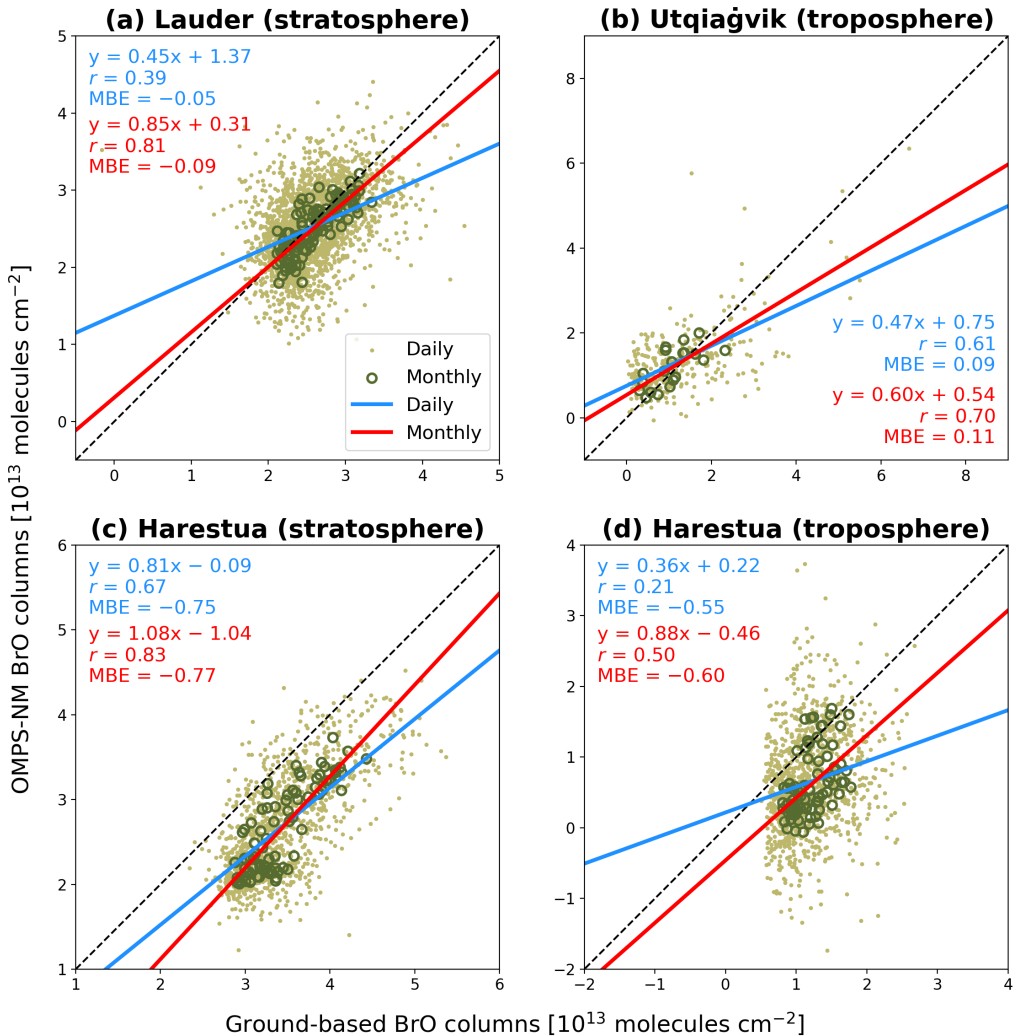

**Figure 9.** Scatter plots of ground-based versus OMPS-NM BrO retrievals. The station name and the targeted column (stratosphere or troposphere) are indicated above each panel. Monthly and daily data and their regression lines are overlaid (see the legend for description). Slopes and intercepts of regression lines, correlation coefficients ($r$), and mean bias errors (MBEs) are also indicated.

ertheless, biases in the flattened profiles can propagate to the $V_{\text{trop}}^{\text{flat}}$ estimates and, consequently, influence the final tropospheric retrievals.

Unlike random uncertainties, systematic uncertainties in parameters have signs and thus can either reinforce or cancel one another. To establish systematic uncertainty budgets, it is essential to perform further intercomparisons of OMPS-NM BrO retrievals with more independent observation data at various locations in the future.

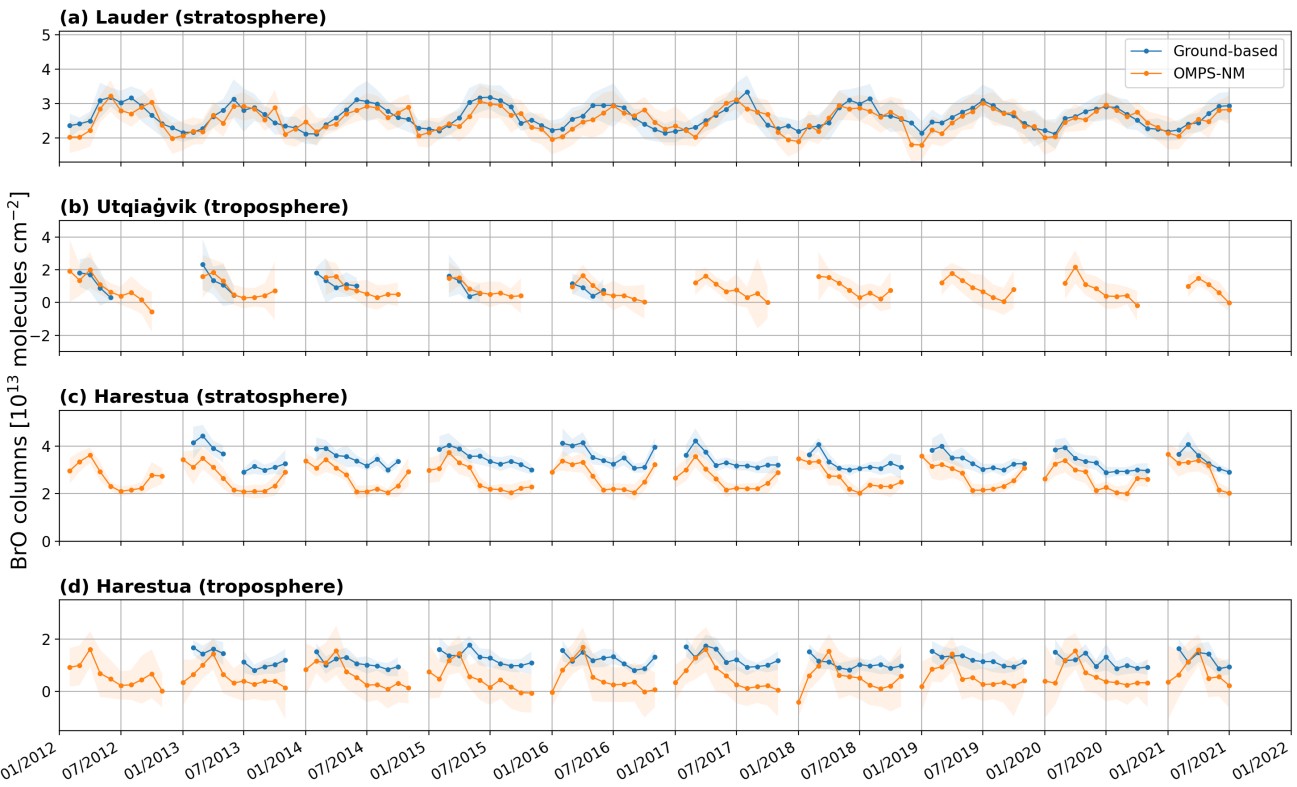

**Figure 10.** Time series of monthly OMPS-NM and ground-based BrO VCDs. The station name and the targeted column (stratosphere or troposphere) are presented above each panel. The shades represent standard deviations of data averaged each month.

## 4 Global tropospheric BrO column retrieval results

Here, we discuss global distributions of tropospheric BrO VCDs retrieved from OMPS-NM by producing gridded monthly mean data with a cell size of $0.1° \times 0.1°$. The physical oversampling method is used to grid the pixel-by-pixel BrO retrievals (Sun et al., 2018). As in the intercomparison above, we use only OMPS-NM retrievals with cloud fractions $\leq 0.5$, SZAs $\leq$
5    80°, "good" quality flags, and cross-track positions from 1 to 34 (0-based).

We divide the globe into six 30°-wide latitude bands and derive the monthly variations of tropospheric BrO VCDs for each band. By aggregating 8-year gridded data from January 2013 to December 2020 for each latitude band and each month, we calculate the 10th, 50th (median), and 90th percentiles.

Figure 11c shows the results for the northern high latitudes (NH, 60–90°N). The highest monthly median value in NH
10    is found in April ($1.61 \times 10^{13}$ molecules cm$^{-2}$), primarily contributed by Arctic springtime BrO enhancements. The 90th percentile is up to $2.01 \times 10^{13}$ molecules cm$^{-2}$. Figure 11a shows the spatial distribution of the 8-year April mean tropospheric BrO VCDs in NH. Large VCDs are typically found over the ocean and in coastal areas, mainly contributed by BrO production

over sea ice and snowpack (see Figs. D1a and D2a). The corresponding 8-year April mean tropospheric AMFs show high values in similar areas (Fig. 11b), demonstrating that high albedos over snow/ice are adequately considered during AMF calculations. Tropospheric BrO VCDs are especially large in the Baffin Bay (Fig. 11a), with the 50th and 90th percentiles in April of $2.17 \times 10^{13}$ and $2.59 \times 10^{13}$ molecules cm$^{-2}$, respectively (Fig. 11c). It should be noted that these values are based on monthly averages, and VCDs from individual bromine explosion episodes are higher (e.g., see the upper panels in Fig. 8).

Monthly tropospheric BrO VCD variations in the southern high latitudes (SH, 60–90°S) are presented in Fig. 11f. The highest monthly median value is found in October ($1.13 \times 10^{13}$ molecules cm$^{-2}$). This value is lower than the April median from NH, but it is worth noting that the medians in SH include grid cells in Antarctica, where tropospheric BrO enhancements are hardly found (Fig. 11d). The Southern Ocean is where enhanced tropospheric BrO VCDs are detected, leading to the 90th percentile in SH up to $2.58 \times 10^{13}$ molecules cm$^{-2}$ in August (Fig. 11f). Moreover, narrowing the domain down to the Ross Sea (Fig. 11d), the highest 90th percentile from September is up to $3.33 \times 10^{13}$ molecules cm$^{-2}$ (Fig. 11f). The highest median is found in September as well, which is $2.22 \times 10^{13}$ molecules cm$^{-2}$ (Fig. 11f). As in NH, the spatial distribution of tropospheric AMFs in SH reflects the snow/ice surface effects (Fig. 11e).

On the sub-hemispherical scale, monthly variabilities in tropospheric BrO VCDs found in the mid- and low-latitude regions are typically weaker than those at the high latitudes (Fig. 12c). The highest monthly median values in 30–60°N, 0–30°N, 0–30°S, and 30–60°S are found in March, February, July, and September, respectively, and the maximum among the four values is $1.11 \times 10^{13}$ molecules cm$^{-2}$ from March in 30–60°N. As an example, the monthly tropospheric VCD field in March 2018 is presented in Fig. 12a. Large VCDs contributing to the March peak in 30–60°N are typically found over the ocean. Due to the intrinsic design characteristics of the STS method, the relatively high values in the retrieved tropospheric BrO VCD fields ($V_{\text{trop}}$) can be attributed to either the initial estimates (post-flattening VCDs, $V_{\text{trop}}^{\text{flat}}$), predominantly contributed by the free troposphere, or detected tropospheric enhancements. The contribution of $V_{\text{trop}}^{\text{flat}}$ constantly appears for a given location and month, regardless of the day or year, while that of tropospheric enhancements dynamically varies. Separation of these two impacts can be achieved by calculating the difference between $V_{\text{trop}}$ and $V_{\text{trop}}^{\text{flat}}$, denoted as the contribution of enhancement $V_{\text{trop}}^{\text{enh}} = V_{\text{trop}} - V_{\text{trop}}^{\text{flat}}$. If elevated values in the $V_{\text{trop}}$ field are due to tropospheric enhancements, these pixels should appear in the $V_{\text{trop}}^{\text{enh}}$ field. The $V_{\text{trop}}^{\text{enh}}$ values can be calculated either at the pixel level, assisting in the interpretation of daily retrievals, or by averaging the corresponding $V_{\text{trop}}$ and $V_{\text{trop}}^{\text{flat}}$ data. Figure 12b displays the spatial distribution of $V_{\text{trop}}^{\text{enh}}$ for March 2018, calculated by subtracting the monthly average of $V_{\text{trop}}^{\text{flat}}$ from that of $V_{\text{trop}}$, providing information on where enhancements were detected.

On urban/regional scales, hotspots are detected on land in the mid- and low-latitude regions. Magenta markers in Fig. 12a and b indicate two noticeable hotspots: the Rann of Kutch and the Great Salt Lake. Figures 13–14 show enlarged views of tropospheric BrO VCDs for the two hotspots in March 2018. The figures also present intermediate variables to verify whether the retrieved high VCD values are actual BrO signals rather than artifacts.

The Rann of Kutch is a well-known BrO hotspot (Fig. 13a), previously detected by OMI, GOME-2, and TROPOMI (Hörmann et al., 2016; Seo et al., 2019). Hörmann et al. (2016) found that the tropospheric BrO VCDs around this salt marsh had significant correlations with surface UV radiation, suggesting that the BrO molecules were generated by photochemistry. The large ΔSCDs in Fig. 13c demonstrate that OMPS-NM can also detect enhanced BrO signals over the Rann of Kutch. The

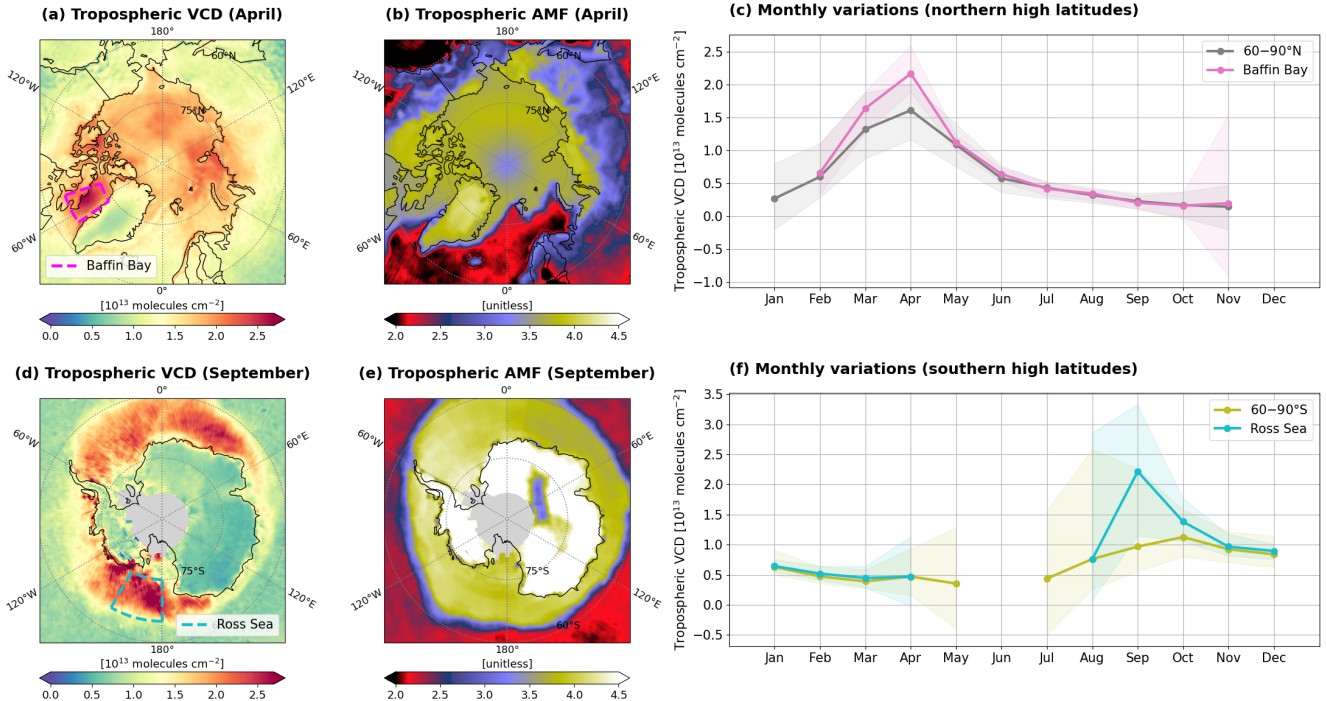

**Figure 11.** Monthly averages and variations of tropospheric BrO VCDs in the northern (60–90°N) and southern (60–90°S) high latitudes. The upper panels show the northern high-latitude (a) mean tropospheric BrO VCDs (April 2013–2020), (b) mean tropospheric AMFs (April 2013–2020), and (c) monthly variations of tropospheric BrO VCDs (2013–2020). The lower panels (d–f) show the corresponding results for the southern high latitudes. In the time series, the dots and shades represent the medians and the 10th–90th percentile ranges, respectively, of all monthly gridded data within the selected areas from the respective months in 2013–2020. Monthly variations in the Baffin Bay and the Ross Sea are also presented. The locations of these two areas are indicated in panels (a) and (d), respectively.

enhanced $\Delta$SCD values are above the random uncertainty levels (Fig. 13d). The GLER distribution reveals bright surfaces over the Rann of Kutch (Fig. 13b), as described by Hörmann et al. (2016), which can lead to positive biases in cloud fraction (Fig. 13e) and cloud pressure (Fig. 13f) retrievals. However, the $\Delta$SCD and GLER fields show no strong spatial correlations. Furthermore, the enhancements in $\Delta$SCDs due to the high surface reflectances are compensated by tropospheric AMFs, which

5 also show higher values in the area (Fig. 13g). As such, we attribute the enhanced tropospheric VCDs from OMPS-NM over the Rann of Kutch to physical/chemical variabilities in BrO concentrations (Fig. 13h).

Figure 14a shows the geographic features of the Great Salt Lake in the USA. Based on ground-based observations, Stutz et al. (2002) suggested that high-molality salt solutions or crystalline salt concentrated around this lake could host heterogeneous reactions that generate BrO molecules in the atmosphere. OMI later demonstrated space-borne detection of enhanced BrO

10 VCDs around the lake (Chance, 2006; Suleiman et al., 2019). Here, we find that elevated BrO columns around the Great Salt Lake can also be detected from OMPS-NM. Figure 14c shows BrO $\Delta$SCDs from OMPS-NM averaged for March 2018. The

enhanced BrO signals are above the noise level (Fig. 14d) and are led neither by surface reflectance (Fig. 14b) nor clouds (Fig. 14e–f), judged by the different spatial distributions. By applying tropospheric AMFs that account for the surface and cloud effects (Fig. 14g), we confirm that enhanced BrO signals also appear in the tropospheric VCD field.

Figure 15a presents persistent tropospheric BrO enhancements over the Rann of Kutch (within the magenta box in Fig. 13a) above the background represented by the values over the area outside the magenta box in Fig. 13a. Consistent with the findings of Hörmann et al. (2016), the enhancements appear especially strong during March–May. The maximum difference between the Rann of Kutch and the background monthly median is $0.66\times10^{13}$ molecules cm$^{-2}$ (111%), found in May 2019. This month, the difference for the 90th percentiles between the Rann of Kutch and surrounding areas was also the largest, corresponding to $1.66\times10^{13}$ molecules cm$^{-2}$. On average, the Rann of Kutch and the background medians (90th percentiles) differ by $0.29\times10^{13}$ ($0.78\times10^{13}$) molecules cm$^{-2}$ in March–May.

The tropospheric BrO enhancements over the Great Salt Lake have a weak seasonal dependence (Fig. 15b). The maximum difference between the monthly medians of tropospheric BrO VCDs from the Great Salt Lake (within the magenta box in Fig. 14a) and the background (outside the magenta box in Fig. 14a) is $0.57\times10^{13}$ molecules cm$^{-2}$ (72%), found in January 2013. The difference in the 90th percentiles this month was $0.56\times10^{13}$ molecules cm$^{-2}$. Throughout the period shown in Fig. 15, the mean difference in the median values is $0.12\times10^{13}$ molecules cm$^{-2}$.

As mentioned earlier, the OMPS-NM BrO retrievals offer a significant advantage through the continuation of the OMI data. In this context, we verify the consistency between OMPS-NM and OMI BrO VCDs in two areas with prominent tropospheric enhancements: (a) the northern high latitudes (60–90°N) and (b) the Rann of Kutch. Given that the OMI BrO retrievals offer only total VCDs, the comparisons made here are limited to total VCDs. Figure 16 shows the monthly total BrO VCDs from OMPS-NM and OMI overlaid from February 2012 to July 2021. The monthly variations agree well, demonstrating that the OMPS-NM BrO retrievals are capable of extending the afternoon BrO time series. The OMPS-NM BrO retrievals are typically lower than those from OMI, likely due to differences in the retrieval algorithms. For example, the OMI retrieval algorithm uses different configurations for the source spectrum (solar irradiance) and the fitting window (319–347.5 nm) (Suleiman et al., 2019). Figure 17 presents the comparison between the OMPS-NM and OMI total BrO VCDs over the Rann of Kutch for March 2018. The spatial distributions align consistently between the two datasets, supporting the earlier discussions based on the OMPS-NM retrievals.

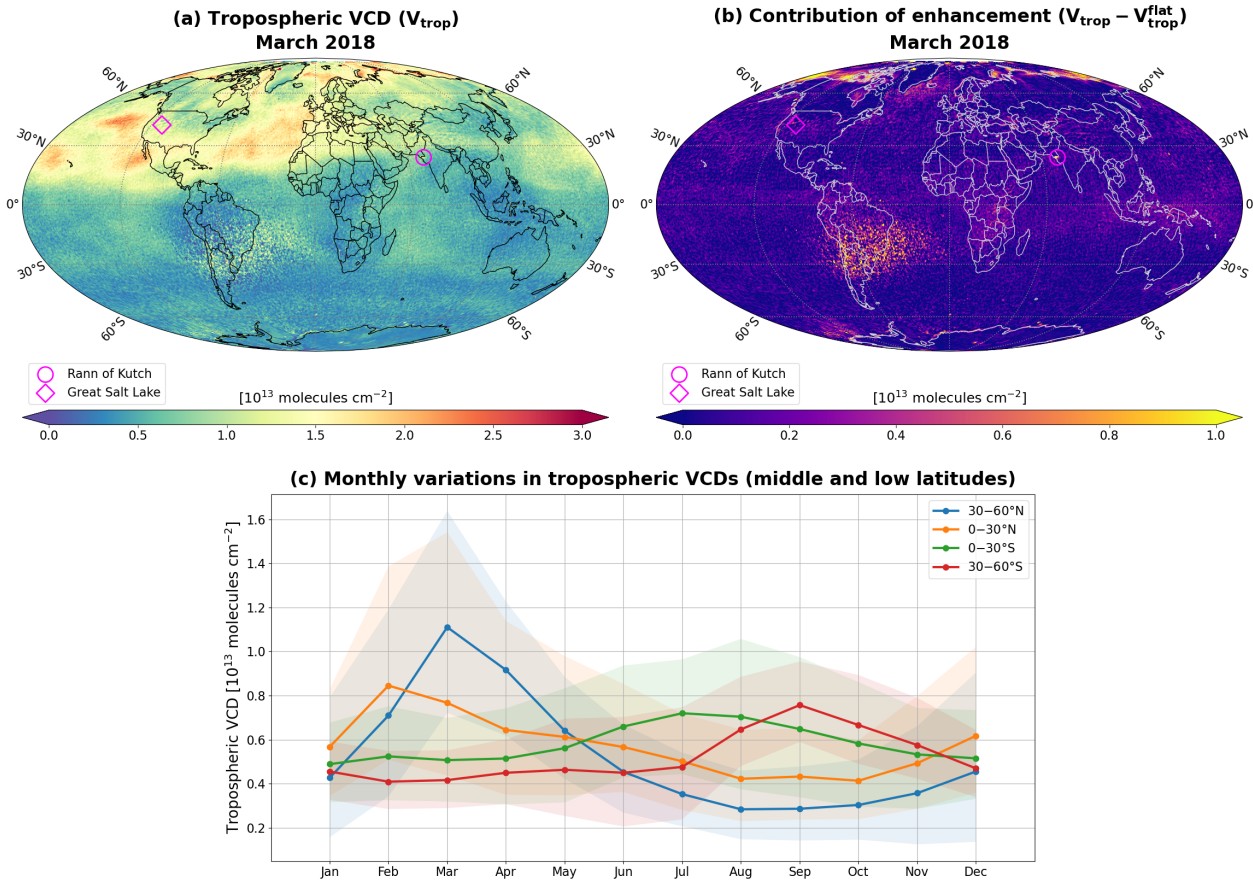

**Figure 12.** Monthly averages and variations of tropospheric BrO VCDs in the mid- and low-latitude regions. (a) Global monthly mean tropospheric VCDs ($V_{trop}$) in March 2018; (b) quantified contributions of enhancements ($V_{trop} - V_{trop}^{flat}$) in the tropospheric VCDs in March 2018; (c) monthly variations of tropospheric BrO VCDs (2013–2020) for every 30°-wide latitude band from 60°N to 60°S. The locations of two selected hotspots are indicated in the maps (the Rann of Kutch and the Great Salt Lake). In panel (b), the enhancements detected in the South Atlantic Anomaly (SAA) are likely associated with spikes and transient events in the radiance data. In panel (c), the dots and shades represent the medians and the 10th–90th percentile ranges, respectively, derived from all monthly gridded data during 2013–2020.

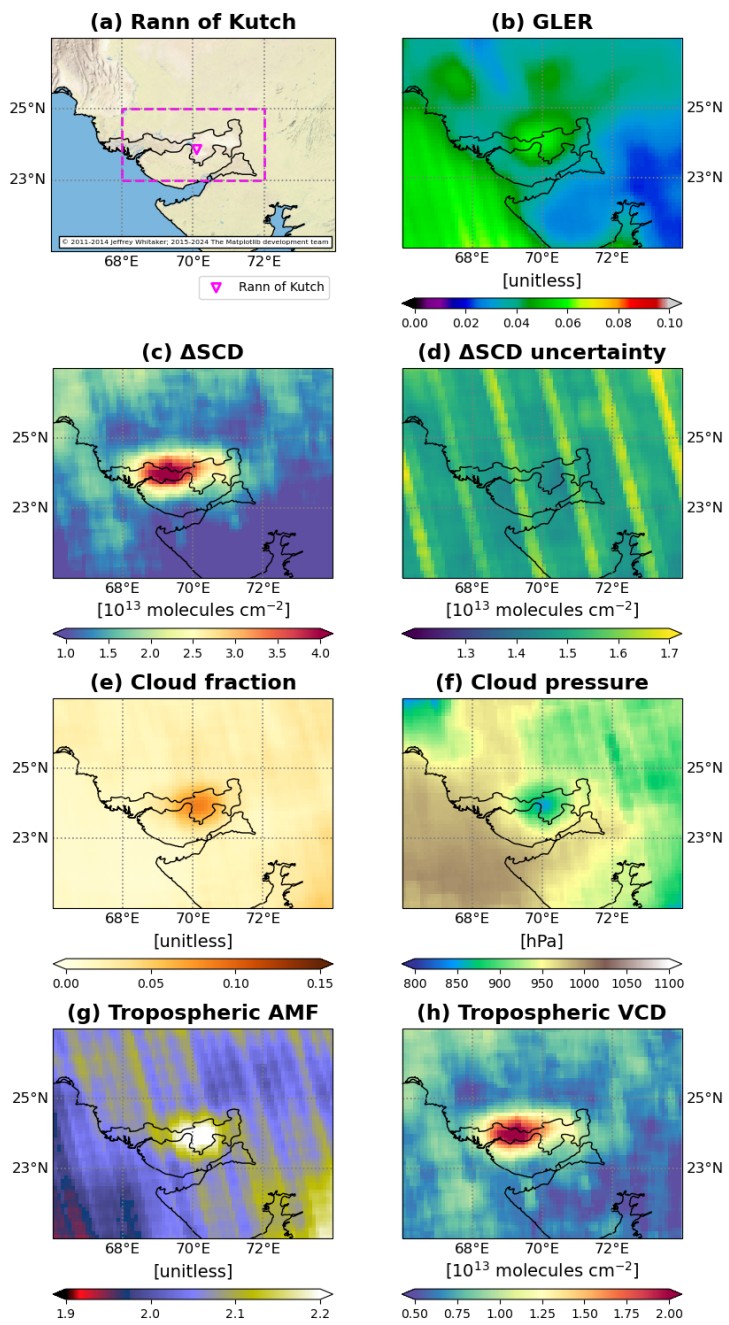

**Figure 13.** Monthly averages of tropospheric BrO VCDs and intermediate variables in March 2018 over the Rann of Kutch. Panel (a) shows the geographic features around the Rann of Kutch (highlighted by the magenta box), and the other panels show monthly averages of (b) GLERs; (c) ΔSCDs; (d) random ΔSCD uncertainties; (e) effective cloud fractions; (f) cloud pressures; (g) tropospheric AMFs; and (h) tropospheric BrO VCDs. It should be noted that in panel (d), the displayed values represent monthly averages of individual pixel uncertainties rather than uncertainties of monthly ΔSCD averages.

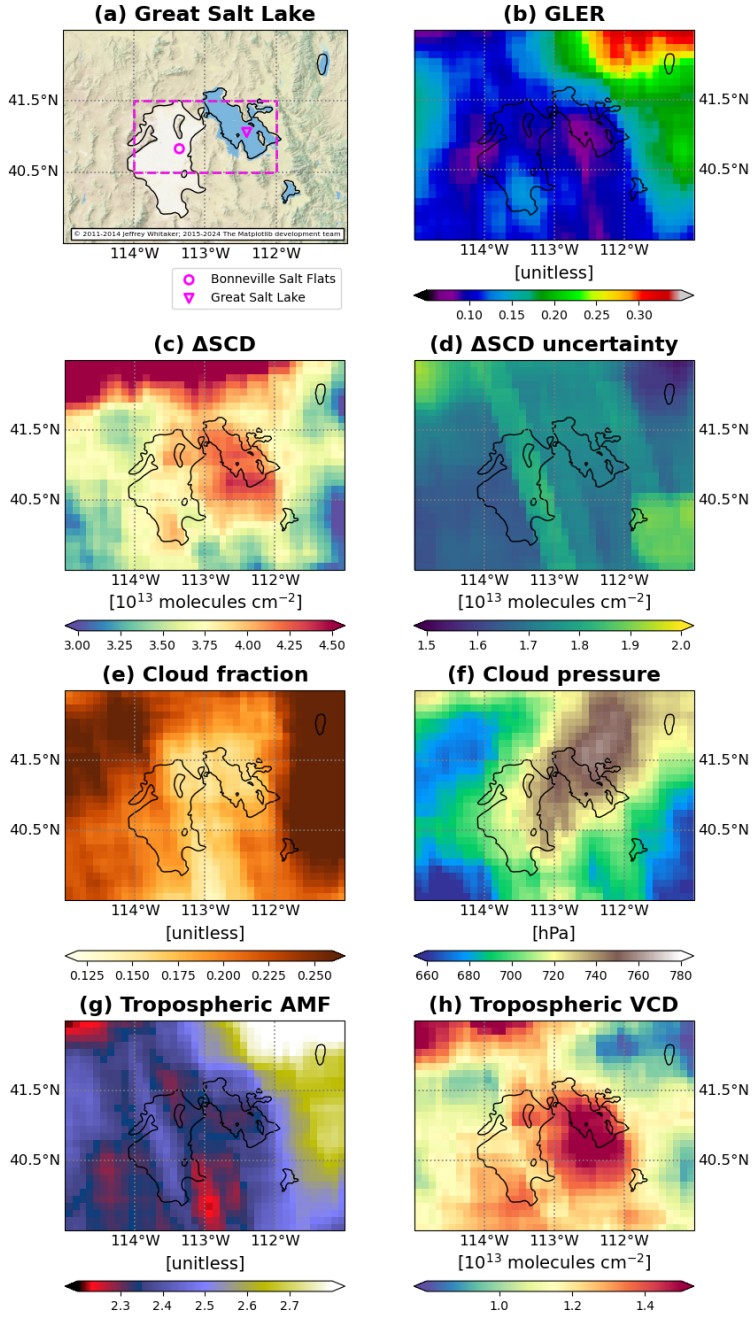

**Figure 14.** Same as Fig. 13 but for the Great Salt Lake.

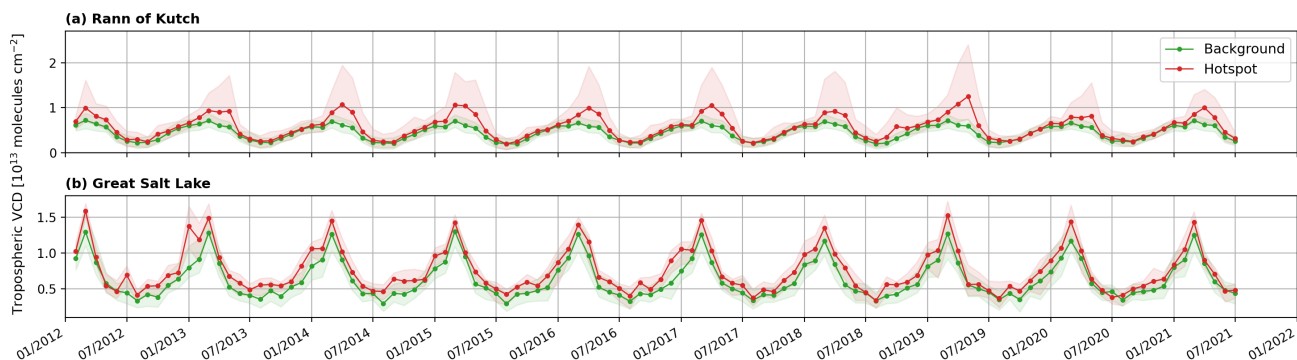

**Figure 15.** Time series of monthly tropospheric BrO VCDs in (a) the Rann of Kutch and (b) the Great Salt Lake. The hotspot in the legend represents the areas within the magenta boxes with dashed lines in Figs. 13a and 14a. The background in the legend represents the areas between the boxes and the map frames in Figs. 13a and 14a. In the time series, the dots and shades represent the medians and the 10th–90th percentile ranges within the selected areas, respectively.

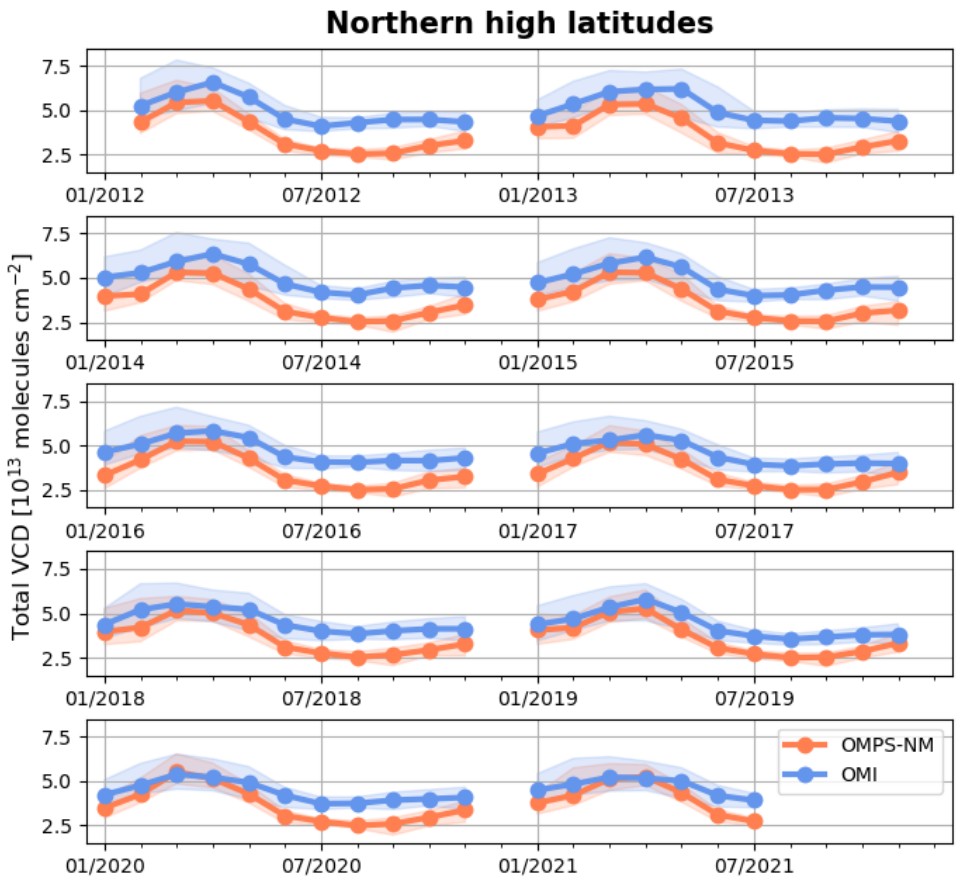

**Figure 16.** Time series of monthly total BrO VCDs from OMPS-NM and OMI in the northern high latitudes (60–90°N). The dots and shades represent the medians and the 10th–90th percentile ranges in the region, respectively.

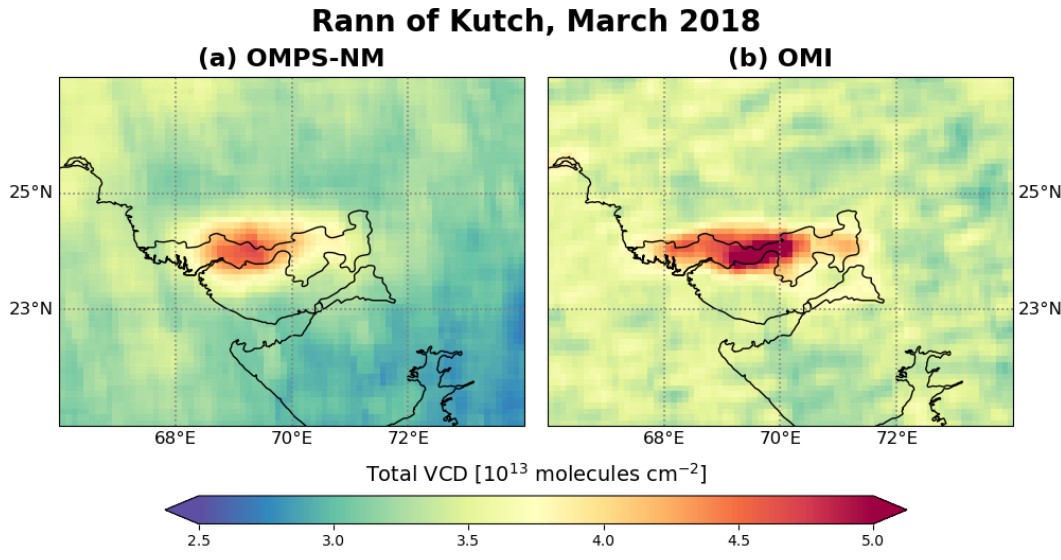

**Figure 17.** Monthly averages of total BrO VCDs from OMPS-NM and OMI over the Rann of Kutch in March 2018.

## 5 Discussion and conclusions

Stratospheric and tropospheric BrO columns retrieved from OMPS-NM are stored in a Level 2 file for each orbit, along with their uncertainties and supporting data. A primary benefit of the OMPS-NM BrO dataset is its ability to provide continued space-based BrO retrievals in the afternoon on a global scale each day. This dataset is particularly beneficial for completing daily global coverage from 2012, which is partially missing due to the loss of useful OMI data caused by a systematic instrumental issue (the row anomaly). In addition to the instrumental benefit, the OMPS-NM BrO dataset offers a significant advantage by providing empirically separated stratospheric and tropospheric columns for nearly a decade, a feature that is currently rare in publicly available datasets.

The STS scheme designed in this study combines the benefits of the existing methods to address two main aspects of separating stratospheric and tropospheric BrO VCDs: (a) segregation of tropospheric enhancements from stratospheric and (b) reliable calculations of tropospheric AMFs. Specifically, for the first aspect, our scheme dynamically detects tropospheric BrO enhancements (hotspots) by deriving stratospheric $O_3$-BrO relationships. For the second aspect, we employ modeled BrO profiles, creating a second set of BrO profiles by smoothing out the vertical gradients of the original tropospheric profiles (i.e., by flattening). This second profile set serves two different purposes: (a) estimation of the initial tropospheric BrO field and (b) AMF calculations for non-hotspots. Ultimately, we assess whether each OMPS-NM pixel is a hotspot or not and apply the appropriate BrO profile for the AMF calculation, choosing between the pre-flattening (hotspot) and post-flattening (non-hotspot) profiles. In addition to the selective use of BrO profiles, the performance of tropospheric AMFs is enhanced particularly by online calculations with dynamic inputs of surface BRDF (for land), surface wind speed (for ocean), and sea ice extent.

The STS method proposed in this study is not without limitations. First, hotspot detection exhibits lower reliability within the polar vortex. Users can choose to filter out hotspots detected within the vortex based on their specific analyses and requirements, using the STS quality flags in the product. Second, the retrieved tropospheric BrO VCDs depend on the initial estimates determined using the flattened profiles. Therefore, biases in the flattened profiles propagate to the final tropospheric retrievals. Specifically, relatively high values in the retrieved tropospheric BrO VCD fields are due to either the initial estimates or detected tropospheric enhancements. To facilitate the separation of the two contributors, the OMPS-NM BrO product provides information on (a) the initial estimates of tropospheric BrO VCDs and (b) the specific pixel locations with tropospheric enhancements (STS quality flags). Additionally, it is worth noting that the contribution of the initial estimates constantly appears for a given month, regardless of the day or year.

Despite the limitations, OMPS-NM retrievals demonstrate their ability to capture daily tropospheric enhancements, particularly those observed over the northern high latitudes. Meanwhile, as expected, the monthly averages exhibit lower discrepancies from ground-based retrievals, likely attributed to smaller random errors. The intercomparison of monthly averages with ground-based retrievals in Lauder, Utqiaġvik, and Harestua shows good agreement for both the stratosphere ($r$ = 0.81–0.83) and the troposphere ($r$ = 0.50–0.70). The MBEs of the monthly stratospheric VCDs from OMPS-NM against the ground-based retrievals are $-0.09 \times 10^{13}$ and $-0.77 \times 10^{13}$ molecules cm$^{-2}$ in Lauder and Harestua, respectively. The monthly tropospheric

BrO VCDs from OMPS-NM have MBEs of 0.11 $\times 10^{13}$ and $-0.60 \times 10^{13}$ molecules cm$^{-2}$ in Utqiaġvik and Harestua, respectively. To enhance quantitative applications of the OMPS-NM BrO dataset, random uncertainties are explicitly estimated pixel by pixel. On the other hand, systematic uncertainties need further characterization through intercomparisons with more ground-based data from various areas. More thorough intercomparisons will aid in a better understanding of biases arising

from the initial estimates of tropospheric VCDs and AMF uncertainties.

Eight-year (2013–2020) monthly mean OMPS-NM BrO retrievals gridded at $0.1° \times 0.1°$ resolution reveal climatological monthly variations in tropospheric BrO VCDs over the northern (60–90°N) and southern (60–90°S) high latitudes. Within the respective regions, the Baffin Bay and the Ross Sea exhibit relatively large monthly tropospheric VCDs with April and September medians of $2.17 \times 10^{13}$ and $2.22 \times 10^{13}$ molecules cm$^{-2}$. It should be noted that significantly larger tropospheric BrO

VCDs are found for individual bromine explosion episodes. The OMPS-NM dataset identifies tropospheric BrO enhancements not only in the polar but also in the extrapolar regions. In particular, the $0.1° \times 0.1°$ monthly mean tropospheric BrO VCDs within the Rann of Kutch and the Great Salt Lake show the spatial medians larger than the background medians by up to 111% (May 2019) and 72% (January 2013), respectively.

Our BrO retrieval algorithm is designed for cross-sensor applications. Specifically, its application to instruments with higher

spatial resolutions will enable a more detailed investigation of tropospheric BrO distributions. Ultimately, applying the proposed algorithm to multiple UV spectrometers will enable continuous long-term records of satellite BrO data record.

*Data availability.* The OMPS-NM BrO data will be available online through NASA GES DISC as a Community Product (https://doi.org/10.5067/PSPSYHVDNSJE). Before release, the data are available from the corresponding author upon request. The other OMPS-NM datasets used for BrO retrieval are available from the following NASA GES DISC links: https://doi.org/10.5067/DL081SQY7C89

(Level 1B), https://doi.org/10.5067/CJAALTQUCLO2 (cloud), and https://doi.org/10.5067/0WF4HAAZ0VHK (total O$_3$). The MODIS BRDF product (MCD43C1) used for AMF calculations is available from the NASA LP DAAC (https://doi.org/10.5067/MODIS/MCD43C1.061). The MERRA-2 data are available from NASA GES DISC (https://doi.org/10.5067/VJAFPLI1CSIV and https://doi.org/10.5067/QBZ6MG944HW0). The ocean salinity product (World Ocean Atlas 2009) is available from the NOAA NCEI (https://www.nodc.noaa.gov/OC5/WOA09/pr_woa09.html). The MODIS Terra chlorophyll product is available at https://oceancolor.gsfc.nasa.gov/atbd/chlor_a. The sea ice data for the Northern and

Southern Hemispheres are available from NOAA NSIDC at https://doi.org/10.7265/N52R3PMC and https://doi.org/10.7265/N5K072F8, respectively.

## Appendix A: Fitting window optimization

The BrO $\Delta$SCDs derived from Eq. (2) and the corresponding errors vary with the fitting window. To find the optimal one, we evaluate candidate fitting windows based on the following four variables: (a) fitting RMSE, (b) BrO $\Delta$SCD random uncertainty, (c) BrO $\Delta$SCD absolute bias, and (d) absolute correlation coefficients between Jacobians of BrO and other trace gas $\Delta$SCDs. We consider spectral ranges of 320–343 nm and 346–369 nm for the lower and upper limits of fitting windows, with a 0.25 nm sampling. To select a fitting window exhibiting good performance for all seasons, we employ four OMPS-NM orbits for the optimization: o32229 (15 January 2018), o33506 (16 April 2018), o34797 (15 July 2018), and o36102 (15 October 2018). These are all reference orbits, i.e., the radiances within 0–10°N latitudes from each orbit are averaged and used as radiance references to retrieve the BrO $\Delta$SCDs.

We aim to find a fitting window that provides low values of all four above-mentioned variables. The fitting RMSE and BrO $\Delta$SCD random uncertainty are calculated as described in Sect. 2.6.1. The biases in BrO $\Delta$SCDs are calculated against modeled $\Delta$SCDs, which are determined using BrO vertical columns from the Community Atmosphere Model with Chemistry (CAM-Chem) (Fernandez et al., 2019). For the fitting window evaluation, we use only stratospheric $\Delta$SCDs from CAM-Chem to avoid a possible mismatch between the observed and modeled tropospheric BrO columns. After co-locating the modeled stratospheric vertical columns onto OMPS-NM pixels, we determine the modeled stratospheric SCDs by applying the geometric air mass factors (AMFs) (the sum of the secant of solar and viewing zenith angles)[2]. Lastly, for each cross-track position in each orbit, we calculate the modeled stratospheric $\Delta$SCDs by subtracting the median of the modeled stratospheric SCDs within 0–10°N latitudes from the modeled stratospheric SCD at every along-track pixel.

Here, we define the $\Delta$SCD bias as the modeled stratospheric BrO $\Delta$SCD minus the retrieved total $\Delta$SCD. This definition assumes that the total BrO SCD is dominated by the stratospheric portion, which is valid in most cases. These bias values are used only in a relative sense to compare performances among different fitting windows. We take the absolute values of the biases before comparison.

The correlation coefficients between BrO and other $\Delta$SCD Jacobians ($r_{\mathrm{BrO},j}$) are calculated from the Jacobian covariance matrix $\boldsymbol{C} = (\mathbf{K}_x^{\mathrm{T}}\mathbf{K}_x)^{-1}$:

$$r_{\mathrm{BrO},j} = \frac{C_{\mathrm{BrO},j}}{\sqrt{C_{\mathrm{BrO,BrO}}C_{j,j}}}, \tag{A1}$$

where $C_{\mathrm{BrO},j}$, $C_{\mathrm{BrO,BrO}}$, and $C_{j,j}$ represent the elements of the Jacobian covariance matrix with the corresponding rows and columns indicated in subscripts. For interfering species to take into account, we select $O_3$ and formaldehyde (HCHO), whose Jacobians are potentially correlated with that of BrO (González Abad et al., 2016; Seo et al., 2019). We consider $O_3$ $\Delta$SCD Jacobians at both temperatures because the BrO cross section interferes with each individual $O_3$ absorption spectrum during the spectral fitting. Since we fit three parameters to account for the absorption of $O_3$ at 243 K, we calculate a combined correlation

---

[2]The configurations described in this appendix is used for the purpose of fitting window optimization only. For other applications, please refer to Sects. 2.3 and 2.4 for details.

coefficient $R_{\mathrm{BrO},O_3}$ for this temperature to capture their effects simultaneously:

$$R_{\mathrm{BrO},O_3} = \left( \begin{bmatrix} r_{\mathrm{BrO},O_3} & r_{\mathrm{BrO},T_1} & r_{\mathrm{BrO},T_2} \end{bmatrix} \begin{bmatrix} r_{O_3,O_3} & r_{O_3,T_1} & r_{O_3,T_2} \\ r_{T_1,O_3} & r_{T_1,T_1} & r_{T_1,T_2} \\ r_{T_2,O_3} & r_{T_2,T_1} & r_{T_2,T_2} \end{bmatrix} \begin{bmatrix} r_{\mathrm{BrO},O_3} \\ r_{\mathrm{BrO},T_1} \\ r_{\mathrm{BrO},T_2} \end{bmatrix} \right)^{1/2}, \tag{A2}$$

where $T_1$ and $T_2$ represent the two parameters from the first-order Taylor series expansion (Pukīte et al., 2010). As a result, the absolute correlation coefficients for $O_3$ at 243 K are typically higher than those at 273 K (see Figs. A1–A3).

To find a fitting window that shows the best performance in all latitude ranges, we perform the assessment separately for three latitude bins: (a) high latitudes (60–90°S and 60–90°N combined), (b) middle latitudes (30–60°S and 30–60°N combined), and (c) low latitudes (30°S–30°N). In the assessment, we exclude pixels with cloud fractions > 0.2 and those with snow or ice to constrain conditions for stratospheric bias analysis. Additionally, pixels with solar zenith angles (SZAs) > 80° are filtered out.

Figures A1–A3 display the assessment results for o33506 (16 April 2018) at high, middle, and low latitudes, respectively. The same assessments are also conducted for the other three orbits (o32229, o34797, and o36102), although the detailed results for these orbits are not presented. The focus here is on highlighting the assessment methodology and presenting the most relevant findings for o33506 on April 16, 2018. The first assessment step involves extracting windows that fall below the medians for all six variables (fitting RMSE, BrO $\Delta$SCD random uncertainty, BrO $\Delta$SCD absolute bias, and three absolute correlation

coefficients). This assessment is performed for each orbit and latitude bin. The next step involves sorting out windows that satisfy the below-median conditions for all orbits and latitude bins. Among the sorted windows, we identify 331.5–358 nm as the most optimal fitting window, marked with red circles in Figs. A1–A3. This specific window simultaneously exhibits low values of fitting RMSE, random $\Delta$SCD uncertainty, absolute $\Delta$SCD bias, and correlation coefficients with $O_3$ and HCHO.

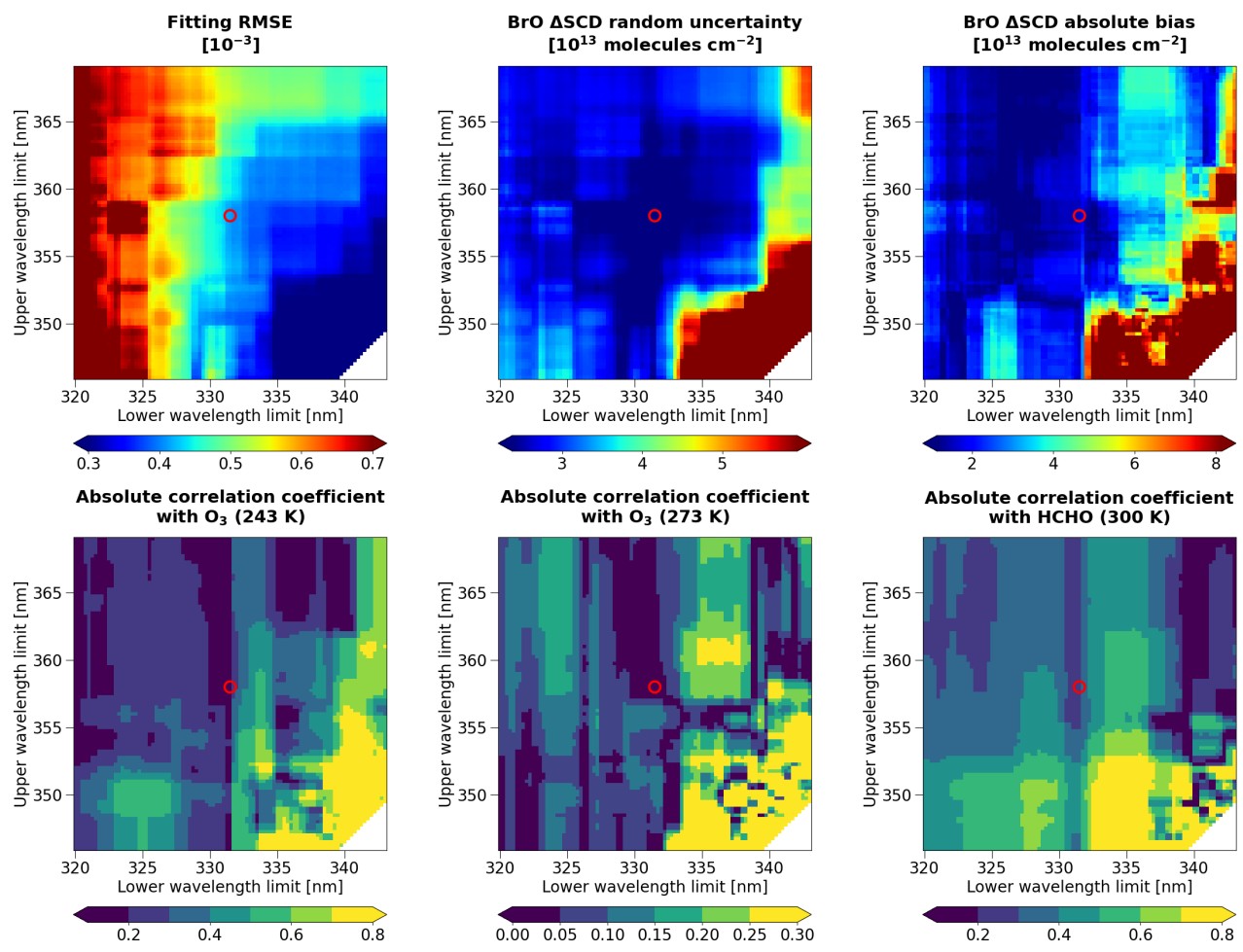

**Figure A1.** Fitting window assessment for high latitudes using o33506 (16 April 2018). Spectral ranges of 320–343 nm and 346–369 nm are used for the lower and upper limits, respectively, with 0.25 nm sampling. The name of the variable is noted above each panel. The optimal fitting window is indicated with red circles.

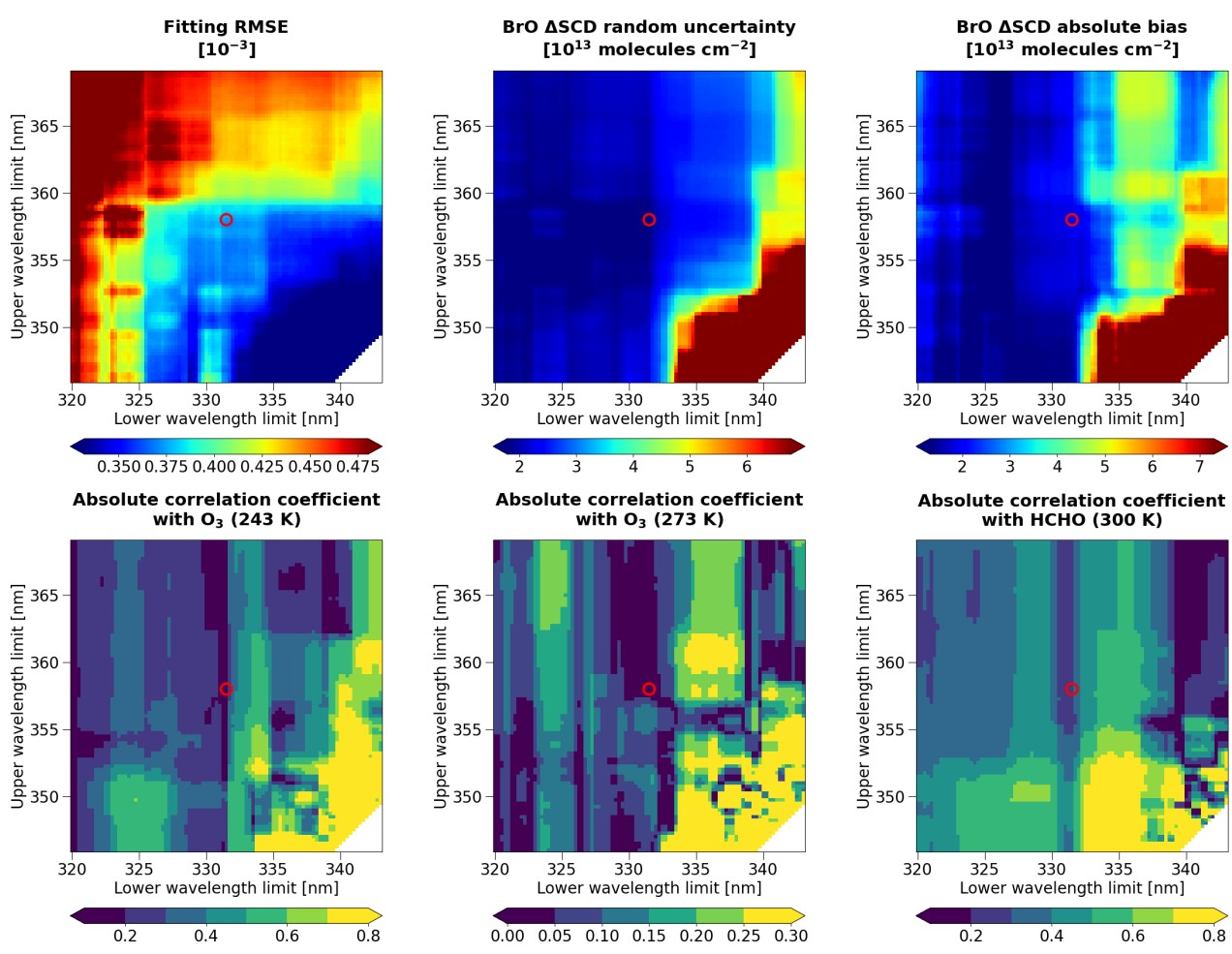

**Figure A2.** Same as Fig. A1 but for middle latitudes.

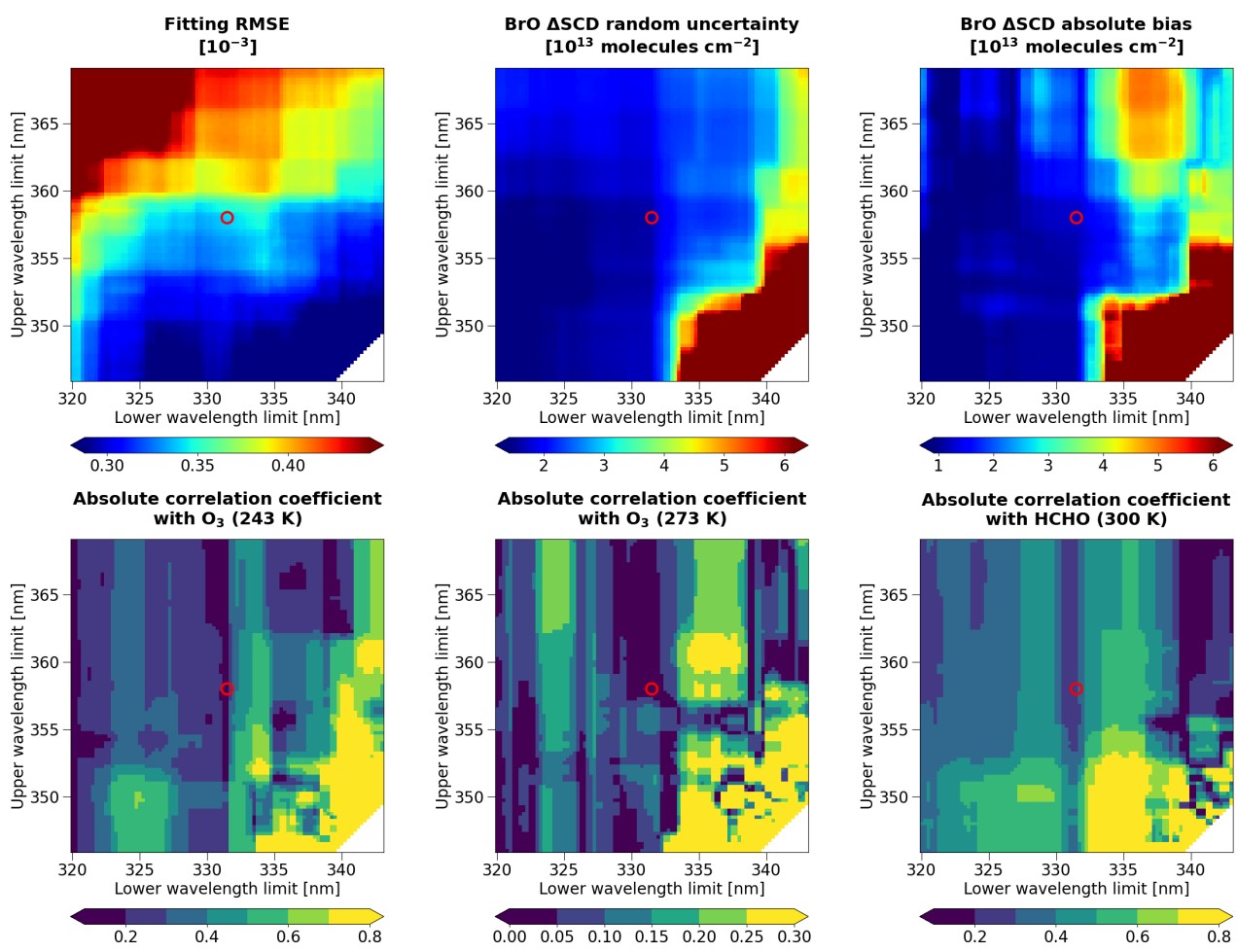

**Figure A3.** Same as Fig. A1 but for low latitudes.

## Appendix B: Profile flattening

The profile flattening technique aims to simulate tropospheric BrO vertical profiles representing localized background conditions based solely on climatology. To achieve this, we first investigate the dependence of modeled tropospheric profile shapes on the corresponding tropospheric VCD values. Ideally, this analysis should be performed for a fixed location, involving various model simulations with different initializations. To replicate this analysis using predetermined climatology, we gather BrO profiles from different locations using the CAM-Chem model, constraining surface type, time, latitude, month, and tropopause pressure.

In Fig. B1, the vertical profiles of BrO sampled on land for 13:30 LST are grouped to examine the variations in each 45°-wide latitude band for January, April, July, and October. Each panel exclusively displays profiles with tropopause pressures within $\pm 20$ hPa of the mean value for the corresponding latitude band and month. Figure B2 depicts the profiles sampled with the same criteria but for ocean and sea ice. These two figures demonstrate that higher tropospheric VCDs tend to have more complex vertical structures, while the lowest (background) VCDs typically have flat profiles with a decreasing pattern of BrO VMRs from the tropopause toward the ground. This characteristic is also supported by Fig. 7a.

Given the relatively high stability of the free troposphere and the typical occurrence of BrO VMR enhancements in the lower troposphere, we devise the flattening technique that smooths out tropospheric profiles beginning at the tropopause and proceeding downward in altitude. As described in Sect. 2.5.2, for each profile, we recursively compare two consecutive VMRs and replace the larger value with the smaller one, moving from the tropopause to the surface. The outputs of the flattening procedure are model-based boxcar-shaped profiles. Each of these BrO profiles is composed of background VMRs that are never below the minimum VMR value within the tropospheric layers of the co-located CAM-Chem profile. Figure B3 presents flattening results for four pixels selected from o7594 (15 April 2013). Panel (c) illustrates a prominent background profile, demonstrating our motivation through minimal differences in VMR values before and after flattening. The other panels show more significant differences between the pre- and post-flattening profiles.

The profile flattening outputs serve two purposes: (a) estimating initial (background) tropospheric BrO VCDs and (b) calculating AMFs for non-hotspots. Figure B4 presents examples of the CAM-Chem tropospheric BrO VCDs before ($V_{\text{trop}}^{\text{CTM}}$) and after ($V_{\text{trop}}^{\text{flat}}$) flattening, along with the final tropospheric VCD retrievals ($V_{\text{trop}}$). Without flattening, the $V_{\text{trop}}^{\text{CTM}}$ fields show homogeneously elevated values around the polar regions (panels b and f), whereas the $V_{\text{trop}}^{\text{flat}}$ fields exhibit reduced values (panels c and g). Having $V_{\text{trop}}^{\text{flat}}$ as initial estimates, the $V_{\text{trop}}$ fields capture individual dynamic tropospheric enhancements (panels d and h) appearing in the $\Delta S$ fields (panels a and e).

# Land

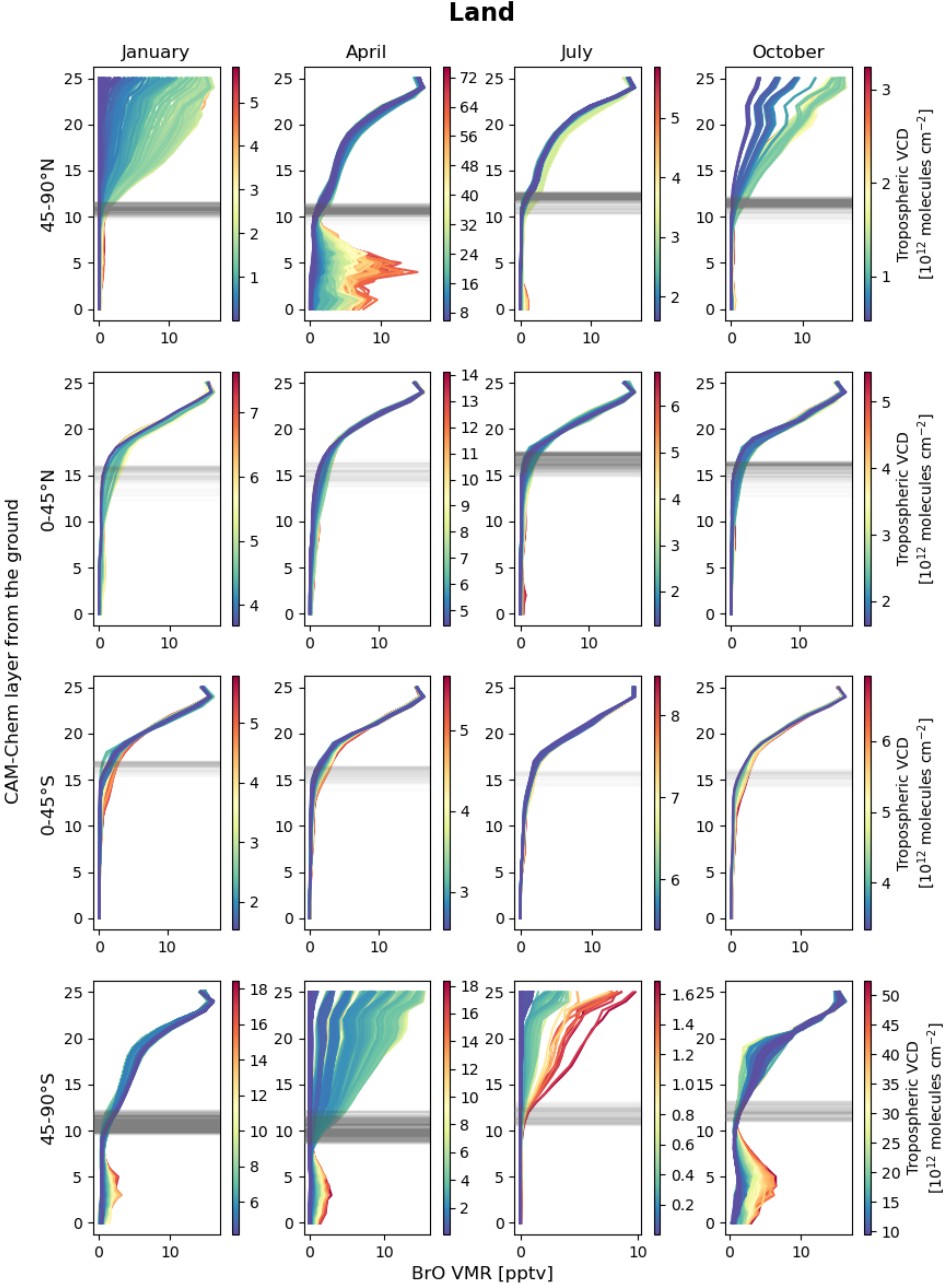

**Figure B1.** Variations in BrO vertical profile shapes over land for different latitudes and months, depending on tropospheric VCD values. Each row corresponds to a latitude band, and each column corresponds to a month. The curves in each panel are color-coded to represent tropospheric BrO VCDs, with the corresponding color bar presented on the right side. The y-axes indicate the 26 CAM-Chem layers; gray horizontal lines mark the tropopause layers.

# Ocean and sea ice

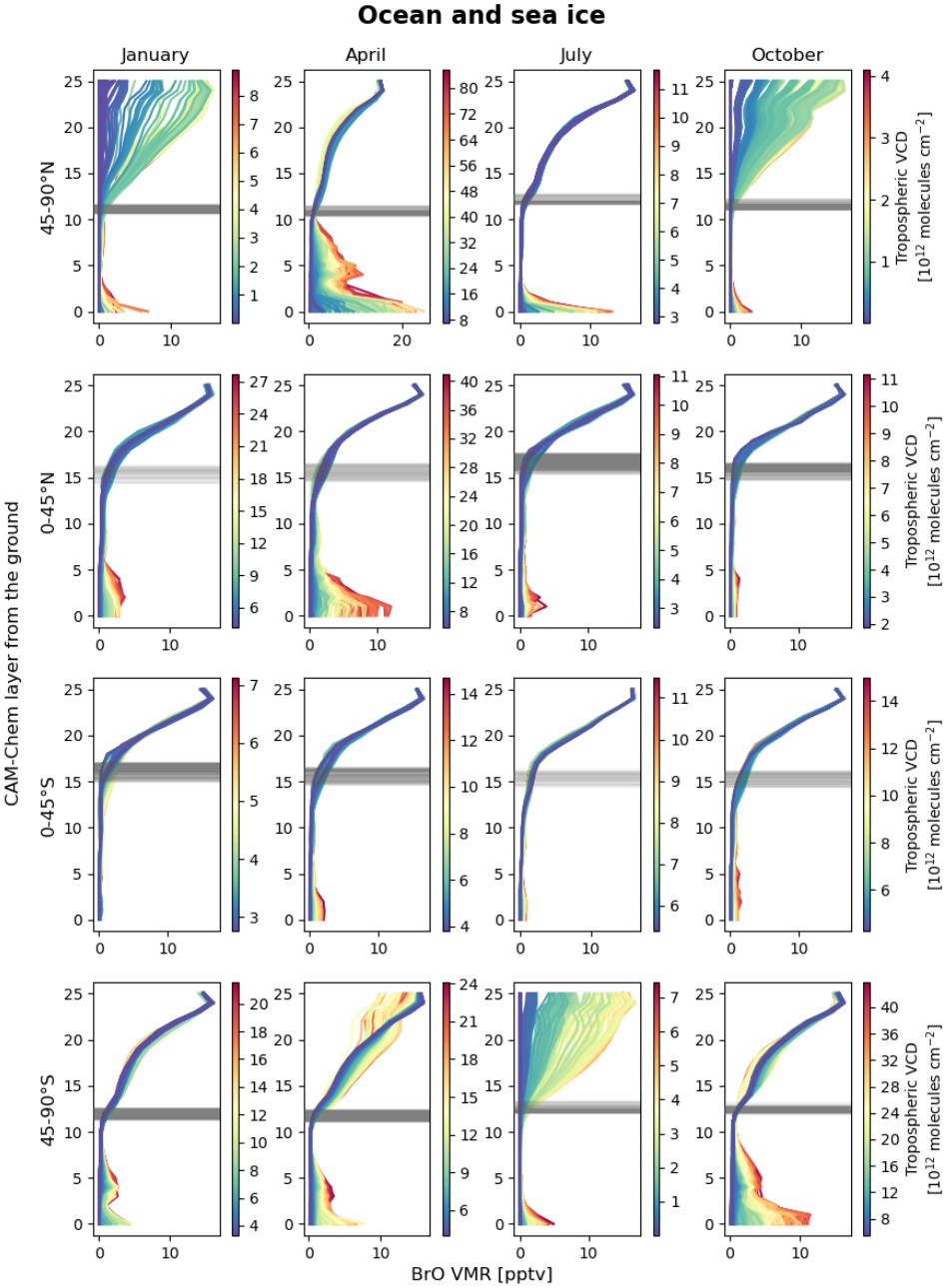

**Figure B2.** Same as Fig. B1 but for ocean and sea ice.

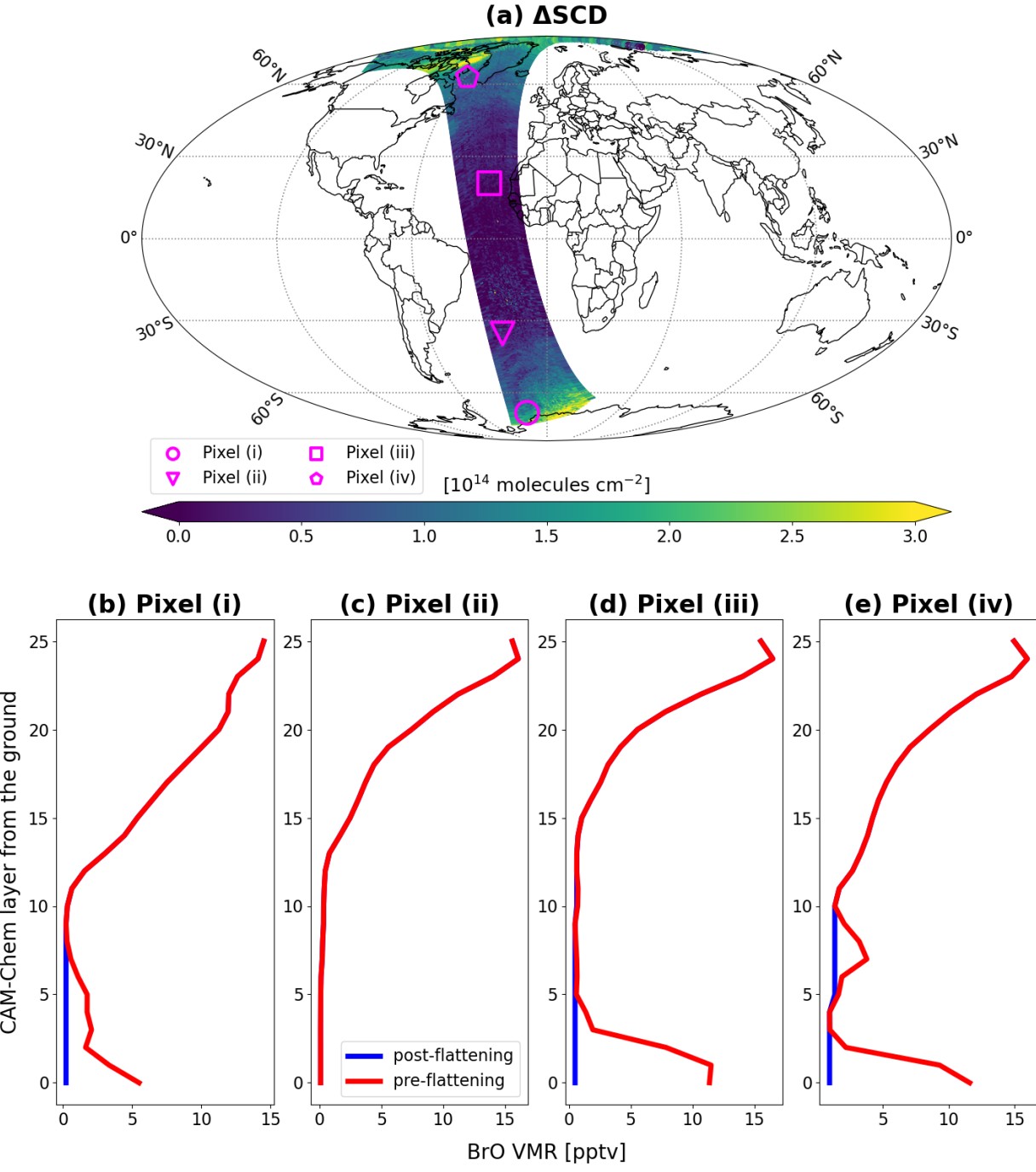

**Figure B3.** Profile flattening. Four pixels are selected from o7594 (15 April 2013). Panel (a) presents the ΔSCD field with the selected pixels indicated. Panels (b–e) display the pre- and post-flattening BrO profiles for each selected pixel.

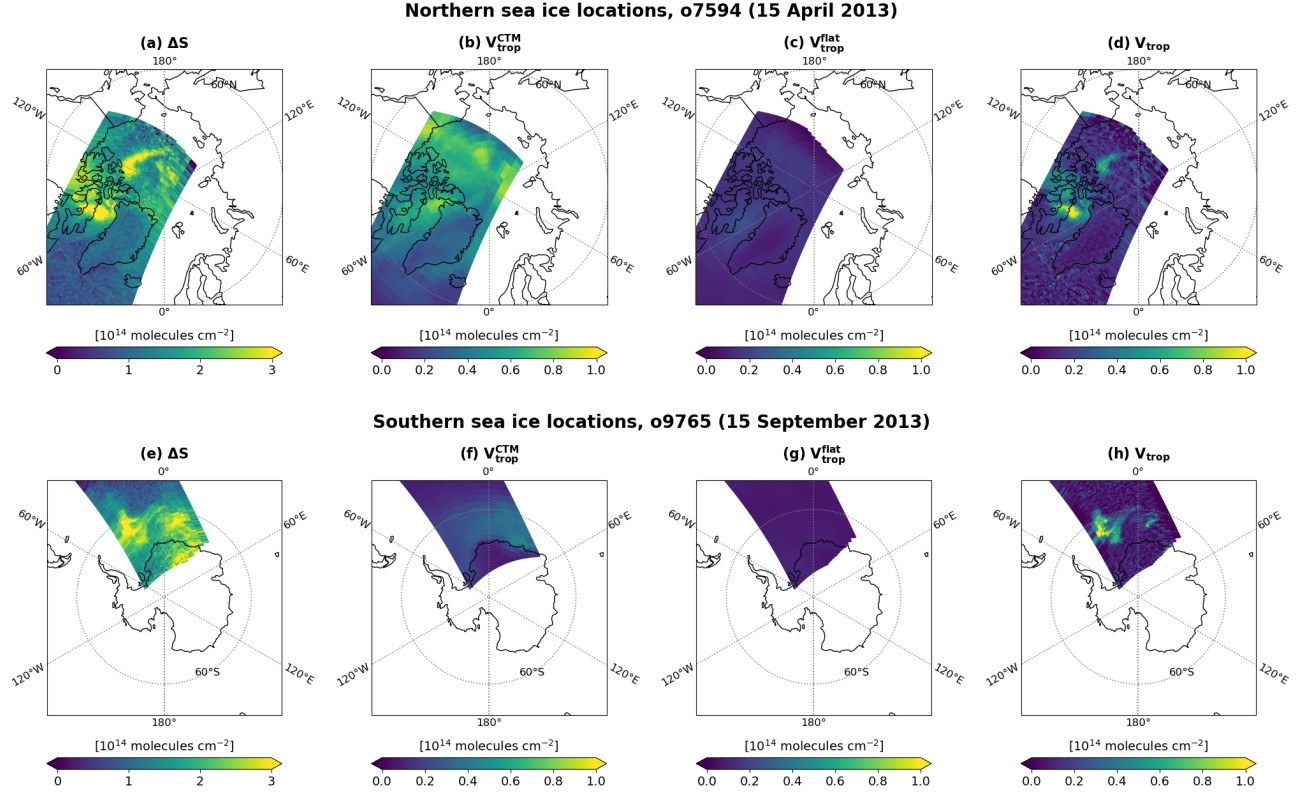

**Figure B4.** Comparisons between modeled and retrieved tropospheric BrO VCDs. Two orbits are selected to represent sea ice locations in the Northern Hemisphere (o7594, 15 April 2013) and Southern Hemisphere (o9765, 15 September 2013). Panels (a) and (e) show the retrieved $\Delta$SCDs ($\Delta S$). Panels (b) and (f) present the modeled tropospheric VCDs before flattening ($V_{\text{trop}}^{\text{CTM}}$), while (c) and (g) show the post-flattening tropospheric VCDs ($V_{\text{trop}}^{\text{flat}}$). The retrieved tropospheric VCDs ($V_{\text{trop}}$), i.e., the results of the stratosphere-troposphere separation (STS), are displayed in panels (d) and (h).

## Appendix C: Random uncertainties in surface reflectances

The uncertainties in surface reflectances over land from the MCD43C1 product differ depending on the surface type (Wang et al., 2018; Wu et al., 2018). Therefore, we first determine the surface type globally at $0.05° \times 0.05°$ resolution, using the variable "Majority_Land_Cover_Type_1" from the MODIS Land Cover Climate Modeling Grid (CMG) product (MCD12C1 Version 6) (Friedl and Sulla-Menashe, 2019). Then, we assign an uncertainty for each surface type, using the values estimated by Wang et al. (2018) and Wu et al. (2018) (Table C1). For the surface types with no uncertainty estimates, we assign the largest value among similar surface types.

To quantify the random uncertainties in ice surface reflectances, we first estimate the uncertainties in the ice BRDF climatology and convert them to 340-nm GLER uncertainties. As described in Sect. 2.3, the ice BRDF climatology is derived by calculating a global median for each kernel (i.e., isotropic, geometric, and volumetric) from the four shortest wavelength bands of MODIS. Therefore, we define the ice GLER uncertainty as the change in the 340-nm GLER that occurs when each kernel value increases by the global MAD. At this stage, the kernels other than the one of interest are fixed at their median values. The results for the isotropic, geometric, and volumetric kernels are referred to as $\varepsilon_{r,\mathrm{iso}}$, $\varepsilon_{r,\mathrm{geo}}$, and $\varepsilon_{r,\mathrm{vol}}$, respectively. Since these values vary with surface pressure and observation geometries, we construct LUTs for each of them. Table C2 shows the nodes and intervals of the LUTs. These LUTs are applied to every OMPS-NM pixel to derive the second term of Eq. (15) by

$$\left(\frac{\partial A_z}{\partial r}\right)^2 \varepsilon_r^2 = \left(\frac{\partial A_z}{\partial r}\right)^2 \varepsilon_{r_{\mathrm{iso}}}^2 + \left(\frac{\partial A_z}{\partial r}\right)^2 \varepsilon_{r_{\mathrm{geo}}}^2 + \left(\frac{\partial A_z}{\partial r}\right)^2 \varepsilon_{r_{\mathrm{vol}}}^2. \tag{C1}$$

**Table C1.** Random uncertainties in surface reflectances.

| Surface type | Detail | Uncertainty | Reference |
|---|---|---|---|
| Sea ice | sea ice fraction > 0% | look-up table | this study |
| Water bodies | Permanent water bodies > 60% | 0.0180 | Fasnacht et al. (2019) |
| Evergreen needleleaf forests | Tree cover > 60%; canopy > 2 m | 0.0237 | Wang et al. (2018) |
| Evergreen broadleaf forests | Tree cover > 60%; canopy > 2 m | 0.0196 | Wang et al. (2018) |
| Deciduous needleleaf forests | Tree cover > 60%; canopy > 2 m | 0.0237 | This study (maximum) |
| Deciduous broadleaf forests | Tree cover > 60%; canopy > 2 m | 0.0196 | Wang et al. (2018) |
| Mixed forests | Dominated by neither deciduous nor evergreen tree type; tree cover > 60%; canopy > 2 m | 0.0201 | Wang et al. (2018) |
| Closed shrublands | Tree cover > 60%; canopy of 1–2 m | 0.0125 | Wang et al. (2018) |
| Open shrublands (tundra) | Tree cover of 10–60%; canopy of 1–2 m | 0.0318 | Wang et al. (2018) |
| Woody savannas | Tree cover of 30–60%; canopy > 2 m | 0.0318 | This study (maximum) |
| Savannas | Tree cover of 10–30%; canopy > 2 m | 0.0125 | Wang et al. (2018) |
| Grasslands | Herbaceous annual cover > 60%; height < 2 m | 0.0318 | Wang et al. (2018) |
| Permanent wetlands | Water cover of 30–60%; Vegetated cover > 10% | 0.0318 | This study (maximum) |
| Croplands | Cultivated cropland cover > 60% | 0.0318 | Wang et al. (2018) |
| Urban and built-up lands | Impervious surface cover > 30% | 0.0318 | This study (maximum) |
| Cropland/natural vegetation mosaics | Small-scale cultivation of 40–60% | 0.0130 | Wu et al. (2018) |
| Permanent snow and ice | Snow and ice cover > 60% at least 10 months of the year | 0.0505 | Wang et al. (2018) (maximum) |
| Barren | Non-vegetated barren cover > 60%; vegetated cover < 10% | 0.0111 | Wang et al. (2018) |

**Table C2.** Nodes and intervals of the look-up tables for $\varepsilon_{r,\mathrm{iso}}$, $\varepsilon_{r,\mathrm{geo}}$, and $\varepsilon_{r,\mathrm{vol}}$.

| Parameter | Number of nodes | Nodes |
|---|---|---|
| Surface pressure | 11 | 100–1100 hPa with 100 hPa interval |
| Solar zenith angle | 10 | 0–90° with 10° interval |
| Viewing zenith angle | 10 | 0–90° with 10° interval |
| Relative azimuth angle | 9 | −180–180° with 45° interval |

## Appendix D: Sea ice and snow fractions

Sea ice and snow surfaces at high latitudes often host heterogeneous reactions that lead to large tropospheric BrO VCDs (Simpson et al., 2015). Figure D1 shows the mean sea ice fractions for April and September of 8 years (2013–2020) for the northern and southern high latitudes, respectively. Figure D2 shows snow fractions. These figures assist in the interpretation of Fig. 11.

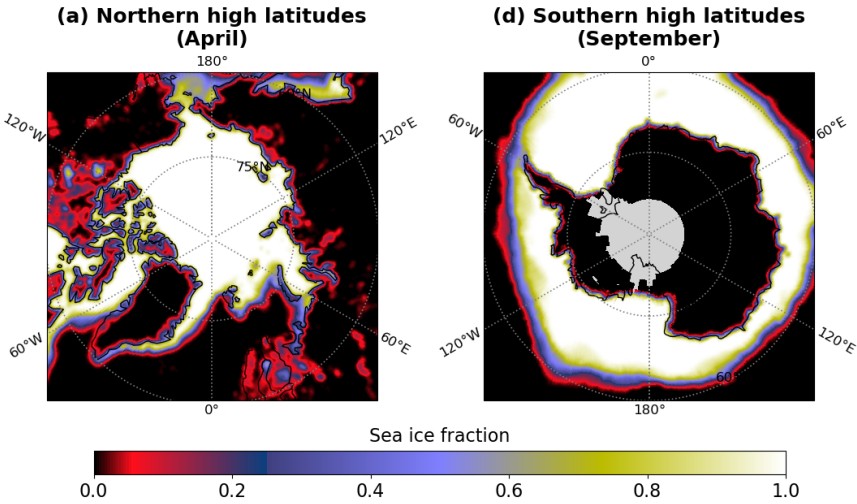

**Figure D1.** Eight-year (2013–2020) sea ice fractions for (a) the northern high latitudes in April and (b) the southern high latitudes in September.

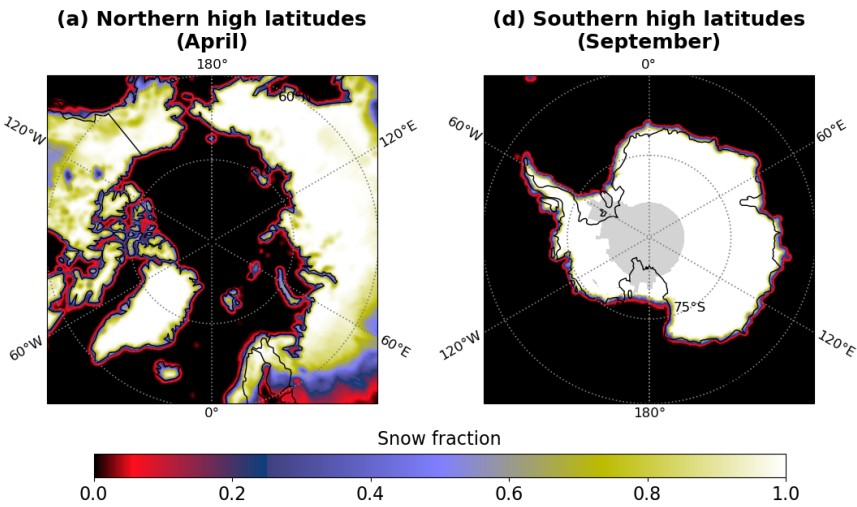

**Figure D2.** Eight-year (2013–2020) snow fractions for (a) the northern high latitudes in April and (b) the southern high latitudes in September.

*Author contributions.* HC performed the BrO retrieval and wrote the manuscript. HC and GGA designed the research. HC, GGA, CRN, CCM, HAK, ZA, HW, AHS, EOS, and RMS provided analysis, investigation, and software. ASL and RPF provided the CAM-Chem model data. XL and KC provided supervision. GGA, XL, KC, JK, JHK, GJ, and CS provided resources. WRS, FH, and RQ provided ground-based data. All authors provided interpretations, feedback, and suggestions, and edited the manuscript.

5  *Competing interests.* Some authors are members of the editorial board of journal Atmospheric Measurement Techniques. The peer-review process was guided by an independent editor, and the authors have also no other competing interests to declare.

*Acknowledgements.* This study was supported by NASA's The Science of Terra, Aqua and SuomiNPP (80NSSC18K0691), NASA's Making Earth System Data Records for Use in Research Environments (80NSSC18M0091), and NASA's Aura Project Core Data Analysis (80NSSC21K0177) grants. Computations in this paper were conducted on the Smithsonian High Performance Cluster (SI/HPC), Smith-
10  sonian Institution (https://doi.org/10.25572/SIHPC). The BrO measurements at Lauder are supported by the NZ Government's Strategic Science Investment Fund (SSIF) through the CAAC research programme at NIWA. Hyeong-Ahn Kwon was supported by the National Research Foundation of Korea (NRF) grant funded by the Korea government (MSIT) (RS-2023-00253460).

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
