# Peer review of "Global retrieval of stratospheric and tropospheric BrO columns from OMPS-NM onboard the Suomi-NPP satellite"

_EGUsphere, 2023_

## Author Comment (AC1)

We greatly appreciate the comments and queries from the reviewer. We were able to enhance the scientific quality of our manuscript significantly by incorporating the reviewer's suggestions during the revision. We tried our best to accommodate all the comments in the revised manuscript. Our answers to the comments and questions are written below in blue. Specific revisions made in the manuscript are underlined.

**RC1**

Review of "Global retrieval of stratospheric and tropospheric BrO columns from OMPS-NM onboard the Suomi-NPP satellite"

In this manuscript, the authors describe a new BrO data product using OMPS-NM measurements. The product includes stratospheric and tropospheric columns, uses a complex stratosphere – troposphere separation algorithm, applies pixel specific airmass factors and provides detailed uncertainty estimates. The topic of the manuscript fits well into the AMT scope, the product described is of interest to the atmospheric chemistry community and the paper is well written and includes detailed descriptions of the algorithms used. I therefore recommend it for publication in AMT.

I have however several questions, comments and suggestions to the manuscript and the algorithm, which the authors should address before the manuscript is accepted:

**1) Reference sector correction**

What the authors call reference sector correction in fact includes two corrections: a) the addition of the modelled BrO offset necessary when using a radiance background spectrum and b) a latitudinal correction of the slant columns based on the model. The latter correction is critical as the way I understand it, it forces the baseline of the measurements on the model values. Therefore, I think the stratospheric BrO product is lo a large extent just reproducing the model values. This is in contrast to the statements in the manuscript claiming that both stratospheric and tropospheric column are retrieved from the measurements.

Please a) discuss this point and b) include an example of the correction for one full orbit, for example one of those shown in Fig. 3.

→ As the reviewer described, the reference sector correction includes two parts. We refer to the latter part as bias correction, in which the bias correction terms ($S_B$) are defined as the differences between two third-degree polynomials fitted to (a) the modeled total SCDs and (b) the background-corrected SCDs ($\Delta S + S_R$) in the along-track dimension for each cross-track position. Since the third-degree polynomials are smooth enough, this approach is capable of correcting smoothly varying biases in the SCD retrievals in the along-track direction without introducing detailed spatial structures from the model into the retrievals. Therefore, this process is rather a correction of offsets than just a reproduction of the model values.

We have added the figure below to the revised manuscript to include an example of the correction for o7594, the orbit shown in Fig. 3a from the initially submitted manuscript (Fig. 4a

in the revised manuscript). Panel (e) displays all four quantities associated with the reference sector correction ($\Delta S$, $S_R$, $S_B$, and $S_{total}$) for the 15th cross-track position (0-based), with $S_B$ values plotted in green. The along-track variability in the retrieved $\Delta$SCDs ($\Delta S$, blue) is well preserved in the total SCDs ($S_{total}$, red). This example demonstrates the capability of the reference sector correction.

A description of the new Fig. 3 has been added to the final part of Sect. 2.2.3 (which became Sect. 2.4 in the revised manuscript).

[Figure]

**Figure 3.** Description of the reference sector correction. Intermediate quantities are presented for o7594 from 15 April 2013. Panels (a–d) show the $\Delta$SCD ($\Delta S$), background SCD ($S_R$), bias correction term ($S_B$), and total SCD ($S_{total}$), respectively. Panel (e) depicts the along-track variabilities of the four quantities for the 15th cross-track position (0-based).

**2) CAM-chem climatology**

The algorithm described heavily relies on the CAM-chem climatology for the stratospheric columns (see above), the separation between stratospheric and tropospheric signals and for the airmass factors. However, a) it is not clear how well the climatology represents the real atmospheric BrO field and b) tropospheric BrO enhancements are very dynamic events, and their magnitude and location cannot be reflected by a static monthly climatology. This has important implications for the airmass factors which probably are often not correct as the monthly mean profiles are neither a good representation of BrO events, nor of background conditions. The algorithm foresees the use of "flattened" profiles in case the measurements with low BrO columns, but the opposite case (high BrO in a region where the climatology does not expect a BrO event) is not treated separately.

Please a) discuss the impacts of using a climatology as input for the airmass factors and b) include a figure comparing the modelled climatological tropospheric columns in comparison to the measurements, for example for the orbits shown in Fig. 3.

→ Using climatology for AMF calculations can result in both random and systematic uncertainties in the retrievals. The contribution to the random uncertainties has already been taken into account in the AMF uncertainty estimates; it is represented by the standard deviation of AMFs for each profile cluster, as described in Sect. 2.3.2 (Sect. 2.6.2 in the revised manuscript).

Regarding systematic uncertainties, there are two main quantities in the retrieval algorithm that can be impacted by using climatology: (a) the initial estimates of tropospheric VCDs and (b) the tropospheric AMFs. The initial tropospheric VCD estimates are supposed to represent the background conditions, and a flattening technique is proposed in this study to achieve it. Still, if there are biases in the flattened profiles (columns), they are propagated to the initial estimates and, ultimately, to the final tropospheric retrievals. This impact is universal. On the other hand, the impacts on the AMF calculations appear differently for non-hotspots and hotspots. The flattened profiles are employed to calculate tropospheric AMFs for non-hotspots, while climatology is applied for hotspots. This approach assumes that climatology can represent profile shapes for tropospheric enhancement events. Systematic errors can occur when this assumption fails.

We have added discussions about the systematic uncertainties due to climatology to the final part of Sect. 3.

To address the reviewer's bullet point (b), we have added Appendix B to the revised manuscript and included the figure below in it. (Appendices B and C in the initially submitted paper have become C and D in the revised manuscript.) This figure compares the modeled and measured tropospheric columns for the orbits shown in Fig. 3 (Fig. 4 in the revised manuscript). Without flattening, the modeled tropospheric columns ($V_{\text{trop}}^{\text{CTM}}$) show homogeneously elevated values, while the final retrievals ($V_{\text{trop}}$) capture dynamic tropospheric enhancement events individually.

[Figure]

**Figure B3.** Comparisons between modeled and retrieved tropospheric BrO columns. Two orbits are selected to represent sea ice locations in the Northern Hemisphere (o7594, 15 April 2013) and Southern Hemisphere (o9765, 15 September 2013). Panels (a) and (e) show the retrieved $\Delta$SCDs ($\Delta$S). Panels (b) and (f) present the modeled tropospheric columns before flattening ($V_{trop}^{CTM}$), while (c) and (g) show the flattened tropospheric columns ($V_{trop}^{flat}$). The retrieved tropospheric columns ($V_{trop}$), i.e., the results of the stratosphere-troposphere separation (STS), are displayed in panels (d) and (h).

**3) Polar vortex**

The authors discuss the well known problem of using O3 columns as proxy for the stratospheric BrO columns and state, that their method "still preserves the overall spatial pattern of the stratospheric field". I have not understood why that should be the case. If we have ozone depletion (and possibly stratospheric BrO enhancement) within the vortex, the relationship between O3 and BrO will be different for vortex and non-vortex air masses, leading to large scatter in the O3 – BrO plot. If all the in vortex values are removed and filled with surrounding (out of vortex) values, the stratospheric BrO column will not be correct.

Please explain why your method is less affected by this problem than that of previous studies.

→ It is true that the stratospheric BrO columns cannot be correct in the polar vortex if all the in-vortex values are removed. However, the removed portion within the vortex is not necessarily 100% because only hotspots detected by the STS scheme are removed within the vortex.

Previous studies directly estimated the stratospheric BrO columns using $O_3$ columns, whereas we use $O_3$ columns only for detecting and reconstructing hotspots. That's why we mentioned that our method preserves the overall spatial pattern of the stratospheric field.

However, the reviewer's comment is correct. Hotspots detected within the vortex have more possibilities to be stratospheric than those detected outside of the vortex. To address this issue, we have added a new quality flag for STS, which is a three-digit binary variable. The first digit represents whether a hotspot is detected, and the second and third represent whether the potential vorticity is larger than a threshold at 475 K and 550 K potential temperature, respectively. The thresholds are 38 PVU (475 K) and 80 PVU (550 K) in the Northern Hemisphere, while those are –55 PVU (475 K) and –90 PVU (550 K) in the Southern Hemisphere. For this purpose, we used potential vorticity data from MERRA-2.

Figure 4 (Figure 5 in the revised manuscript) has been revised as below. Polar-vortex pixels are marked with gray colors in panels (f) and (g). Hotspots within the polar vortex are marked with black (see legend for details). In the revised text, we now recommend that users decide whether to utilize the hotspot data within the polar vortex based on their specific analyses and requirements. If they choose not to use them, it is recommended to filter out only black pixels in panels (f) and (g). For our analyses in this manuscript, we didn't use this quality flag to filter out data points to give an idea of the general retrieval performance.

[Figure]

**4) Uncertainties**

Although a detailed discussion of uncertainties is given, I'm somewhat confused by what to expect from the data product. Are the uncertainties given for individual pixels? Has each pixel in the product an uncertainty value?

→ We apologize for the confusion. Yes, the uncertainties are given for individual pixels, and each pixel in the product has an uncertainty value. To clarify it, we have added a sentence to the beginning part of Sect. 2.3 (Sect. 2.6 in the revised manuscript).

Will uncertainties be smaller in monthly averages? If so, why are the DSCD uncertainties shown in Figs. 12 and 13 for monthly averages comparable to the median uncertainty quoted for an individual pixel?

→ It is correct that the uncertainty in a monthly average value is smaller than that for an individual pixel. However, what we presented in Figs. 12 and 13 (Figs. 13 and 14 in the revised manuscript) are "averages of individual pixel uncertainties," not "uncertainties of averages." That's why the presented values are comparable to the median of individual uncertainties. We chose to present these average uncertainties because the purpose was to investigate if individual uncertainties are low enough to ensure that individually detected $\Delta$SCDs are above the noise level.

To prevent readers' confusion, we have added a sentence to the caption that these are "averages of individual pixel uncertainties."

How is the uncertainty of having a high surface BrO event in the data at a location where the CAM-chem climatology has background conditions taken into account?

→ This situation can occur; however, we consider these systematic errors in retrievals. Since we estimate only random uncertainties in this study, these mismatches are not taken into account.

This comment aligns with the reviewer's comment 2 (CAM-chem climatology). As described in our response to that comment, we have added discussions about the systematic uncertainties due to climatology to the final part of Sect. 3.

Can the product be used on a daily basis (Figs. 3 and 4 suggest that this is the case) or should it better be used on a monthly basis?

→ Daily retrievals are capable of capturing tropospheric enhancements, as presented in the figures. However, as shown in Fig. 8 (Fig. 9 in the revised manuscript), daily retrievals have larger errors than monthly averages. The decision to utilize either daily or monthly data hinges on the specific analyses or requirements of the users. The random uncertainty estimates provided for individual pixels can assist the decision.

Please add error bars to the satellite data in Fig. 7 and a paragraph on data usage.

→ Error bars have been added to the figure, as shown below. However, the lower error bars, which are supposed to have the same lengths as the upper ones, are not plotted for display purposes. This description of the lower error bars has been added to the caption.

Also, we have added a paragraph on data usage to Sect. 5 (Discussion and conclusions).

[Figure]

**5) High values over the ocean**

In Fig. 11, large BrO columns are shown over much of the NH oceans, and as far as I can see, the largest BrO columns in that month are not found in the Arctic but somewhere over the Pacific. Do you think this is realistic? How do these findings compare to other satellite products and independent measurements? Are these high columns already visible in the slant columns or are the introduced by the CAMS-chem based airmass factors?

→ Relatively high values in the final retrievals ($V_{\text{trop}}$) can be either from the flattened columns ($V_{\text{trop}}^{\text{flat}}$), predominantly contributed by the free troposphere, or the detected tropospheric enhancements. Separating those two impacts can be achieved by calculating the difference between $V_{\text{trop}}$ and $V_{\text{trop}}^{\text{flat}}$, i.e., the contribution of enhancement $V_{\text{trop}}^{\text{enh}} = V_{\text{trop}} - V_{\text{trop}}^{\text{flat}}$. If the elevated values in the $V_{\text{trop}}$ field are due to tropospheric enhancements, those pixels should appear in the $V_{\text{trop}}^{\text{enh}}$ field.

The spatial distribution of $V_{\text{trop}}^{\text{enh}}$ for March 2018 has been added to Fig. 11 (Fig. 12 in the revised manuscript), as presented below. The new panel (b) shows the $V_{\text{trop}}^{\text{enh}}$ values, demonstrating where the enhancements were detected from the OMPS-NM instrument.

Regarding the comparison with other data, unfortunately, we were not able to find independent measurements that can be used to validate the tropospheric columns over the Pacific.

[Figure]

**6) Comparison to other satellite data**

Why is there no comparison to other satellite data? The data set is advertised as extension of the OMI afternoon BrO time series, and I think it would be good to include some kind of comparison between BrO observations from the two platforms, even if it is just a visual side-by-side comparison of a monthly average.

→ This is a valid point. We have added the two figures below to the revised manuscript (Figs. 15 and 16). The first figure shows the comparison between OMI and OMPS-NM BrO total columns over the Northern high latitudes (60–90°N). The monthly variations agree well, demonstrating that the OMPS-NM BrO retrievals are capable of extending the afternoon BrO time series. The OMPS-NM BrO retrievals are typically lower than OMI, and that's likely due to differences in the retrieval algorithms. For example, the OMI retrieval algorithm uses different settings for the source spectrum (solar irradiance) and the fitting window (319.0–347.5 nm). The second figure presents the comparison of total BrO columns from OMPS-NM and OMI over Rann of Kutch for March 2018, demonstrating the consistency between the two data sets.

[Figure]

Northern high latitudes

[Figure]

Rann of Kutch, March 2018

(a) OMPS-NM     (b) OMI

$[10^{13}\ \text{molecules cm}^{-2}]$

**7) Profile flattening**

This procedure seems arbitrary to me. I do not see why such a profile should give reasonable tropospheric BrO columns or realistic airmass factors. Please justify your approach.

→ For the justification, we have added the two figures below to Appendix B of the revised manuscript (Figs. B1 and B2). The upper figure shows the variations in the tropospheric profile shapes depending on tropospheric VCD values over land. The data shown here were sampled from the CAM-Chem climatology to represent 13:30 local solar time. For January, April, July, and October, we calculated the mean tropopause pressure ($P_{\text{trop}}$) values for each 45° latitude band. Then, for each month and each latitude band, we sampled BrO profiles from the model grid cells having $P_{\text{trop}}$ values within $\pm 20$ hPa from the mean. The sampled profiles are presented in the figure below for each month/latitude band. Every profile curve is color-coded using the tropospheric VCD value. The gray horizontal lines represent the tropopause. The lower figure shows the same as the upper one but for the ocean and sea ice regions.

As stated, the purpose of the flattening technique is to simulate vertical profiles that represent the background conditions. As shown in the figures below, higher tropospheric VCDs tend to have more complex vertical structures. On the other hand, the lowest (background) VCDs typically have flat profiles with a decreasing pattern of BrO volume mixing ratios (VMRs) from the tropopause toward the ground. Based on this characteristic of the modeled tropospheric BrO VMRs, we employ the flattening approach to generate vertical profiles exhibiting gradually decreasing (or constant) BrO VMRs from the tropopause toward the ground.

**Land**

[Figure]

[Figure]

**Ocean and sea ice**

**8) Coastal artefacts**

In Fig. 10, there are many localised spots of suspiciously high tropospheric BrO along the Antarctic coast but also at the sea ice edge. As this is a longterm average, it is clear that the high values are from too small airmass factors, and this is confirmed by figure (e). In my opinion, this should not be part of the product as it clearly is an artefact. I expect similar artefacts in the daily images also in the NH, for example close to Spitsbergen. My suggestion is to at least include a flag for those retrievals having very small airmass factors, and to increase the uncertainty estimates for such pixels.

→ Thank you for your comment. While addressing this issue, we were able to find the root cause and fix the problem. The cause was the improper treatment of 'ice shelves' around Antarctica in the retrieval algorithm. Specifically, the land mask data employed in this study didn't treat the ice shelves as land. Since they were not sea ice either, our retrieval algorithm treated them as if they

were water bodies. In reality, however, the surface of the ice shelves can be very bright, primarily associated with snow. Therefore, applying water-body surface albedo leads to significant underestimations of tropospheric AMFs.

During the revision, we found out that the ice shelves are treated as land in the NSIDC sea-ice data, and MODIS BRDF values are also given there. Therefore, we applied NSIDC's definition of land when estimating surface reflectivity for latitudes $< -39°$, which led to the use of MODIS BRDF values for the ice shelves. As a result, the localized spots of suspiciously high tropospheric BrO disappeared. The updated figure is presented below.

We haven't found such an issue for the Northern Hemisphere, but just in case, we changed the algorithm configuration to use MODIS BRDF if the IMS data reports the presence of snow over water pixels.

Descriptions have been added briefly to Sect. 2.2.2 (Sect. 2.3 in the revised manuscript).

[Figure]

**9) Large AMF above ocean**

I'm surprised by the large values of the airmass factor over much of the oceans. Values above two indicate the presence of a significant fraction of the BrO in the free troposphere. Do you think this is realistic?

➔ Unfortunately, we were not able to find independent measurement data sets that could be used for verifying BrO columns or profiles in the free troposphere.

This comment aligns with the reviewer's comment 5 (High values over the ocean). As described in our response to that comment, we have revised Fig. 11 (Fig. 12 in the revised manuscript) to address this issue.

**10) Data availability**

I understand that the product is not yet released, but to my knowledge, the data has to be available in some form for the manuscript to be published in AMT. Maybe just add that it is available on request?

→ The product has been accepted for online data distribution through GES DISC (NASA). We have recently received a DOI (10.5067/PSPSYHVDNSJE), but uploading the files has yet to be done. After the product is uploaded to the GES DISC server, it will be publicly available.

We will revise the 'Data availability' paragraph in accordance with the status of the data upload. If the uploading process is not finished by the final stage of the entire revision process, we will add that the product is available upon request before the official release.

---

## Author Comment (AC2)

We greatly appreciate the comments and queries from the reviewer. We were able to enhance the scientific quality of our manuscript significantly by incorporating the reviewer's suggestions during the revision. We tried our best to accommodate all the comments in the revised manuscript. Our answers to the comments and questions are written below in blue. Specific revisions made in the manuscript are underlined.

**RC2**

In the paper titled "Global Retrieval of Stratospheric and Tropospheric BrO Columns from OMPS-NM on the Suomi-NPP Satellite," Chong and colleagues introduce a new BrO column product derived from the OMPS-NM satellite instrument. The authors focus on presenting a long-term time series of tropospheric columns while effectively distinguishing between stratospheric and tropospheric contributions. The paper's strength lies in its innovative approach, combining the strengths of two state-of-the-art methods to separate stratospheric and tropospheric columns. Furthermore, the long operational lifespan of the OMPS-NM instruments lends a distinct advantage to this product, promising continuous data acquisition over an extended period which allows long-term time series of this chemically important tracer. This BrO column product makes a valuable addition to existing satellite-derived BrO products, with potential benefits for bromine chemistry research and is therefore in good alignment with the scope of AMT. I strongly recommend considering this paper for publication after addressing the noted corrections and points.

**General comments:**

**GC1)** The manuscript appears to primarily target polar applications, with retrieval considerations and examples predominantly centered around polar regions. For instance, the effective application of the flattening technique to polar hot-spots raises questions about its performance in non-polar tropospheric enhancements, confer also GC2. Moreover, the wavelength criterion selection (Page 39, line 6-7) implies a concentration on polar applications, aiming to achieve optimal results under high latitude and SZA conditions. While this focus is justified, I suggest to explicitly mention in the abstract, introduction, and potentially the title. Consider either highlighting the emphasis on polar BrO retrieval or substantiating why polar regions pose the greatest challenge.

→ It is a valid point that a significant portion of the retrieval considerations and examples targeted polar regions. In the revised manuscript, we have added sentences highlighting the emphasis on polar BrO retrieval to the abstract and introduction. However, we didn't change the title as the retrievals have global coverage. To incorporate the reviewer's comment, the fitting window optimization has been updated to cover non-polar regions (which will be discussed in the later part of this document).

**GC2)** Even though it is maybe a bit overcomplicated, I generally like the authors approach for the separation of tropospheric and stratospheric columns as it combines the two pathways taken in previous studies. However, there are several steps where I can see potential issues:

1. How does step (i), flattening the tropospheric model profile (page 17, Figure 1), perform with non-polar tropospheric enhancements? Is it specifically tailored for this scenario? For instance, if an extensive area of BrO tropospheric enhancements emerges in the equatorial Pacific due to an unusual climate change event or a significant volcanic plume, how would step (i) handle this situation? It appears that step (i) might overlook such occurrences, possibly leading to their inclusion in V_strat_0 (which may not be problematic by itself). However, a concern arises regarding the effectiveness of the O3-BrO relation-based separation (Figure 4c) under such circumstances.

→ First of all, we'd like to clarify that the flattening technique aims at non-hotspots rather than hotspots. We use flattened profiles (without tropospheric enhancement) for non-hotspots while using modeled profiles (with potential tropospheric enhancement) for hotspots.

We apologize if the description was not clear in the initially submitted manuscript. In the revised manuscript, we have added more sentences to the part where we describe the flattening technique in Sect. 2.2.4 (Sect. 2.5 in the revised manuscript). Also, we have added Appendix B to the revised manuscript to present more details about the flattening. (Appendices B and C in the initially submitted manuscript are C and D in the revised manuscript.)

Regarding the reviewer's questions, the flattening technique was designed for global applications. As described in the manuscript, the retrieval algorithm stores both pre- and post-flattening profiles for every pixel. Then, the algorithm constructs $V_{strat}^0$ by subtracting the post-flattening tropospheric column ($V_{trop}^{flat}$) from the total column. If a tropospheric enhancement emerges at a certain pixel, $V_{trop}^{flat}$ is supposed to be smaller than the actual tropospheric column. Therefore, this pixel must appear in the $V_{strat}^0$ field as a hotspot, just as intended. Then, this pixel is removed from the $V_{strat}^0$ field, using the $O_3$-BrO relation. This process doesn't discriminate between polar and non-polar tropospheric enhancements. For example, the hotspots in Rann of Kutch are well detected and masked, as shown in the figure below (for March 13, 2017).

Although this figure has not been added to the revised manuscript, we have added a description to clarify that the flattening step applies globally.

[Figure]

2. Page 19: In step (v) the authors suggest to use the model profile as input for AMF calculation under no hotspot conditions and in difference the "flattened" model profile as input for the AMF calculation under hotspot conditions. The validity of this approach does not occur to me as I have the following concerns/points.

→ As we described above, it's the opposite. We use the model profile under hotspot conditions and the flattened profile under no hotspot conditions.

3. How confident are you, that this model profile is representative for the tropospheric profile under "no-hotspot" conditions? Please shortly address this in the manuscript.

→ We don't use the model profiles under "non-hotspot" condition. We use flattened profiles in that case to avoid the potential mismatch between the model and the observation. During the revision, we added Appendix B to present more details about the flattening scheme.

4. For hotspot cases, the authors opt for the flattened tropospheric profile. While not explicitly stated, it seems this choice stems from the model profile being decidedly unsuitable for such scenarios. However, employing a flattened profile could potentially worsen the situation. This selection could yield a lower and possibly excessively low tropospheric AMF (A_trop_select, denominator in equation 12). Considering the high S_trop (numerator in equation 12), this might lead to an excessively high V_trop. This concern is particularly relevant when contrasting with non-hotspot pixels where a non-flattened profile is used for tropospheric AMF calculation. Consequently, I have three queries/suggestions regarding the treatment of hotspot cases:

→ As discussed above, the usage of pre- and post-flattening profiles is the opposite of the description in this comment.

4.1. Is there a justification for favoring a flattened profile over the model profile as an assumption?

→ For the justification, we have added the two figures below to Appendix B of the revised manuscript (Figs. B1 and B2). The upper figure shows the variations in the tropospheric profile shapes depending on tropospheric VCD values over land. The data shown here were sampled from the CAM-Chem climatology to represent 13:30 local solar time. For January, April, July, and October, we calculated the mean tropopause pressure ($P_{trop}$) values for each 45° latitude band. Then, for each month and each latitude band, we sampled BrO profiles from the model grid cells having $P_{trop}$ values within ±20 hPa from the mean. The sampled profiles are presented in the figure below for each month/latitude band. Every profile curve is color-coded using the tropospheric VCD value. The gray horizontal lines represent the tropopause. The lower figure shows the same as the upper one but for the ocean and sea ice regions.

As stated, the purpose of the flattening technique is to simulate vertical profiles that represent the background conditions. As shown in the figures below, higher tropospheric VCDs tend to have more complex vertical structures. On the other hand, the lowest (background) VCDs typically have flat profiles with a decreasing pattern of BrO volume mixing ratios (VMRs) from the

tropopause toward the ground. Based on this characteristic of the modeled tropospheric BrO VMRs, we employ the flattening approach to generate vertical profiles exhibiting gradually decreasing (or constant) BrO VMRs from the tropopause toward the ground.

[Figure]

**Ocean and sea ice**

[Figure]

4.2. Given that AMF carries a larger share of random uncertainties, could it be more accurate and practical to assume a distinct "hot-spot" profile shape (e.g., employing a ground-level profile shape as in Fig. 3e for polar hot spots, and a different assumption for tropical cases)?

→ This comment seems to describe an approach similar to what we have been employing. We use modeled profiles (with potential tropospheric enhancement) for hotspots and flattened profiles for non-hotspots. However, we don't discriminate between polar and non-polar (or tropical) pixels when applying this approach. As described above, we prepare flattened profiles all across the globe and selectively use either a modeled or flattened profile on a pixel-by-pixel basis, depending on whether a tropospheric enhancement is detected or not. As shown above, in the case of Rann of Kutch, this approach works even for non-polar regions, as well as the polar regions.

4.3. Is this factor considered in the error propagation of the AMF? Given its potential significance, it would be beneficial to explicitly acknowledge this and its impact.

→ Yes, as a way of considering the impact, we perform the AMF uncertainty estimation separately for the flattened and non-flattened profiles. As described in the manuscript, we calculate partial derivatives of AMF after binning AMFs from 2015 according to six parameters. At this point, we perform the binning separately for hotspots and non-hotspots. This way, we calculate the partial derivatives independently for hotspots (with non-flattened profiles) and non-hotspots (with flattened profiles). Therefore, ultimately, their AMF uncertainties are independent of each other.

To clarify it in the manuscript, we have added a corresponding description to the beginning part of Sect. 2.3.2 (Sect. 2.6.2 in the revised manuscript).

5. Page 19, line 13: Like Sihler et al. (2012), step (iii) employs the assumption of a constant O3/BrO relation to quantify tropospheric enhancements. However, the authors note that this assumption becomes problematic in polar vortex scenarios, potentially introducing bias to your data. Is this data utilized in the final product? If yes, provide rationale for its inclusion and discuss implications for the data quality flag.

→ To be precise, the issues are with hotspots detected in the polar vortex rather than all pixels within the polar vortex. That's because we perform only hotspot detection/reconstruction using the $O_3$ fields rather than constructing the entire stratospheric BrO field.

Nonetheless, we still use all data within the polar vortex in the analyses shown in the manuscript, including the hotspots, to give an idea of the general retrieval performance.

During the revision, however, we added a quality flag variable to the product for users who would like to filter out hotspots within the polar vortex. This variable is a three-digit binary. The first digit represents whether a hotspot is detected, and the second and third represent whether the potential vorticity is larger than a threshold at 475 K and 550 K potential temperature, respectively. The thresholds are 38 PVU (475 K) and 80 PVU (550 K) in the Northern Hemisphere, while those are –55 PVU (475 K) and –90 PVU (550 K) in the Southern Hemisphere. For this purpose, we used potential vorticity data from MERRA-2.

Figure 4 (Figure 5 in the revised manuscript) has been revised as below. Polar-vortex pixels are marked with gray colors in panels (f) and (g). Hotspots within the polar vortex are marked with black (see legend for details). In the revised text, we now recommend that users decide whether to utilize the hotspot data within the polar vortex based on their specific analyses and requirements. If they choose not to use them, it is recommended to filter out only black pixels in panels (f) and (g).

[Figure]

**GC3)** The paper would benefit from a comparison with other satellite studies. This does not need to be thorough, but differences and agreements should be adressed both with regard to the polar observations as well as the BrO from Rann of Kutch.

→ We have added the two figures below to the revised manuscript (Figs. 15 and 16). The first figure shows the comparison between OMI and OMPS-NM BrO total columns over the Northern high latitudes (60–90°N). The monthly variations agree well, demonstrating that OMPS-NM BrO retrievals are capable of extending the afternoon BrO time series. OMPS-NM BrO retrievals are typically lower than OMI, and that's likely due to differences in the retrieval algorithms. For example, the OMI retrieval algorithm uses different settings for the source spectrum (solar irradiance) and the fitting window (319.0–347.5 nm). The second figure presents the comparison of total BrO columns from OMPS-NM and OMI over Rann of Kutch for March 2018, demonstrating the consistency between the two data sets.

[Figure]

[Figure]

**Specific Comments**

**Abstract:**

The abstract could benefit from a clearer articulation of the paper's primary goal and focus. Consider adding a succinct sentence at the beginning of line 4 to outline the central theme of the

study. This could lead into the subsequent statement, "To address this concern and improve upon the current methods, our study introduces..." This adjustment would help provide a smoother transition into the specific achievements and advancements discussed.

→ We have revised the abstract following the reviewer's comment.

**Introduction:**

**Page 3, lines 10-29**: The provided overview of the broad variety of separation schemes used in the literature is commendably thorough and informative. Nonetheless, its level of detail seems too thorough for an introduction. In the introduction, the focus should be on the paper's new method, and a concise acknowledgment of various approaches would suffice to put the paper's method into perspective

Given the absence of a designated "methods" section to accommodate such content as a subsection, I understand the authors' predicament in determining its placement. To address this, a practical solution could be integrating it as a subsubsubsection within Page 14, subsubsection 2.2.4.

→ Following the reviewer's comment, we have shortened the part where we presented the overview of various separation schemes in the introduction. Then, we added Sects. 2.2.4.1 and 2.2.4.2 (Sects. 2.5.2 and 2.5.3 in the revised manuscript). The former provides the overview, moved from the introduction, and the latter delivers the texts originally written in Sect. 2.2.4.

**Page 3 line 35-page 4 line 5**: The paragraph's primary emphasis, as underscored by the authors first lines, lies in the substantial potential of OMPS to provide an extensive and enduring time-series well into the 2030s. This aspect should take precedence, and should be highlighted in the paragraphs first sentence, shifting the focus imediately to this critical attribute. Accordingly, I suggest starting with the assertion about OMPS's long time-series capability, and subsequently incorporating the initial sentence, "OMPS-NM instruments ... decommissioning of TROPOMI (Nowlan et al., 2023)," at the paragraph's conclusion.

→ We have reorganized the paragraph as suggested by the reviewer.

**Page 4, line 16-18 and fig. 1**: There are two confusing aspects:

1. The use of the numeration (1) – (4) in reference to fig. 1, lets the reader look for the numbers 1-4 in fig. 1. However, the corresponding fields are noted with (A)-(D). Please use the same symbols in text and figure

   → We have changed the symbols to (i)–(iv) both in Fig. 1 and the text.

2. When reading "highlighted in blue in fig. 1", the first look in figure 1 will be to the fields which have a blue background "OMPS-NM L1B product" etc. As Figure 1 contains a lot of information it is difficult to find the highlighted fields. Either specify in the text that they are "encircled/framed in blue" or reconsider the coloring within fig. 1 to avoid this

confusion. Also consider to increase the size of the border line to highlight the 4 fields. Furthermore, consider to place the A-D always at the same location w.r.t. the fields they refer to (either all on the top left or top right).

→ We have changed the wording to "framed in blue" and increased the size of the borderline. Also, we have placed the bullets (i)–(iv) at the same position (at the upper left corners).

The resultant figure is shown below.

[Figure]

**Page 4, lines 16-18 and Fig. 1**: This section presents two points of confusion:

1. The utilization of the numerals (1) – (4) in reference to Fig. 1 prompts readers to search for numbers 1-4 within the figure. However, the corresponding elements are actually labeled as (A)-(D). To enhance clarity, it's advisable to employ consistent symbols both in the text and the figure.

    → This appears to be a duplicate comment mirroring the one above.

2. When the text mentions "highlighted in blue in Fig. 1," readers instinctively turn their attention to fields with a blue background, such as "OMPS-NM L1B product," within Figure 1. Since the figure contains extensive information, locating the highlighted elements becomes challenging. To address this, you could specify in the text that the relevant fields are "encircled/framed in blue." Alternatively, reconsider the color scheme

within Fig. 1 to alleviate this confusion. Additionally, consider enhancing the border line's size to accentuate the four designated fields. Furthermore, for consistency, contemplate consistently placing the labels (A)-(D) at the same relative position with respect to the corresponding fields (either all at the top left or top right).

→ This appears to be a duplicate comment mirroring the one above.

By harmonizing symbols and refining visual cues, these adjustments can substantially improve the reader's comprehension.

→ We agree. Thank you for the comment.

**Page 4 and 6**: The mention of 2 times "retrieval" in headline 2 and 2.2 is redundant, I suggest to move all the subsubsections in 2.2 up by one rank in hierarchy (e.g. 2.2.1-> 2.1, etc.).

→ We have revised the hierarchy following the reviewer's comment.

**Page 6, line 6**: The phrasing currently implies that Beirle et al., 2017; Nowlan et al., 2023 originated the super-gauss concept. I recommend revising it to "super Gaussian and adopt the approach outlined in Beirle et al., 2017; Nowlan et al., 2023."

→ We have revised the sentence as recommended.

**Page 6, line 23**: The authors specify their utilization of a 20° latitude portion within a single orbit for the computation of the earthshine reference spectrum. With an assumed along-track pixel footprint of 50 km, this approach implies that each across-track reference spectrum would be derived from 40-50 individual spectra. Notably, other investigations involving 2D CCD satellites like OMI and TROPOMI adopt larger sectors (e.g., Seo et al., 2018: 150°E – 240°W, 30°S-30°N for BrO; Theys et al., 2017: 120-160°W, 10°N-10°S for SO2). Please Justify the rationale behind employing a relatively compact reference sector and argue why such a low statistic is deemed satisfactory for your study.

→ The reason why we use a narrower latitude band is that we found spatial and temporal variabilities of BrO concentrations in the modeled data (CAM-Chem) over the Pacific. The wider the latitude band, the less representative the background SCDs are. Our objective here was to avoid having large spatial and temporal variabilities of BrO within the reference sector.

As mentioned by the reviewer, there are previous examples of using wider latitude bands (e.g., Gonzalez Abad et al., 2015, 2016; Nowlan et al., 2023; Seo et al., 2019; Theys et al., 2017). Except for Seo et al. (2019), who assumed a constant BrO VCD within the reference sector, those retrieval examples mostly targeted a species primarily residing in the troposphere (e.g., HCHO and SO$_2$) with minimized variations in total columns over the Pacific. For BrO, which resides in the stratosphere with significant and varying amounts, a narrow reference sector can be safe as we derive only a single background SCD to represent the entire sector.

After deciding to use a narrower reference sector, we considered the resulting SNRs of the averaged radiance reference spectra to determine the exact width of the sector. Given that there are retrievals employing solar irradiance data, it would be enough if we could generate a radiance reference spectrum with a comparable SNR. The SNR tends to be proportional to the square root of the input signal. Given that the intensity ratios between radiance and irradiance values are typically ~0.05 in the BrO fitting window, the SNR of an individual radiance spectrum would be smaller than that of irradiance by a factor of $\sim\sqrt{20}$. Considering that the noise (error) of an average value is inversely proportional to the square root of the number of samples, we need ~20 radiance spectra for averaging to achieve our goal. In the case of the OMPS-NM instrument, which has 50 km footprints, it corresponds to a ~10°-wide latitude band.

We have added a brief description of why we chose this reference sector to Sect. 2.2.1 (Sect. 2.2 in the revised manuscript).

**Page 8, Table 1**: Regarding the SCD retrieval: All (to the reviewers knowledge) recent other BrO DOAS and "DOAS-like" spectral retrievals include OClO in their spectral retrieval (e.g. Suleiman et al., 2019; Herrmanns et al., 2022; They et al., 2011; Sihler et al., 2012; Seo et al., 2019). Please justify your choice not to include it especially with respect to the potential spectral interferences (see overlapping absorption peak at 344nm).

A similar argument can be made for SO2 although it was only implemented in the most recent publications (Suleiman et al., 2019 (Proposed to be implemented); Herrmanns et al., 2022; Sihler et al., 2012; Seo et al., 2019). Please explain why you have not chosed to include it and how strong you estimate for spectra affected by SO2 (e.g. strong pollution emitter or volcanoes) as well as how substantial this impact is on the global data-set.

→ We have excluded OClO and $SO_2$ from the spectral fitting for the following same reasons: (a) the spatial distribution of the ΔSCDs of the species from the fitting didn't look reasonable (or physical), and (b) the inclusion of the species led to increases in fitting RMS and uncertainty in polar regions, where the BrO chemistry is of particular significance.

The figure just below shows the monthly averages of (a) OClO ΔSCDs, (b) changes in BrO SCDs, (c) changes in fitting uncertainties, and (d) changes in fitting RMS values when we included OClO for December 2018. For the ΔSCD panel, only pixels with cloud fractions < 0.3 were used. Apparently, the spatial distribution of OClO ΔSCDs doesn't reflect well the physical distribution of OClO in the atmosphere. This finding aligns with Pukite et al. (2021), who found that the inclusion of BrO led to significant biases in OClO retrievals from TROPOMI. Pukite et al. (2021) chose to exclude BrO from the fitting and, as an alternative, employ BrO columns determined from an independent fitting for a correction. However, this type of correction was not necessary for the OMPS-NM BrO retrieval because the difference in BrO ΔSCD between with and without OClO was very small, as shown in the upper right panel. Still, since the inclusion of OClO increased the fitting uncertainty and RMS values over polar regions (the lower panels), we simply excluded OClO from the fitting.

[Figure]

The same type of figure for SO₂ is presented below (again for December 2018). The spatial distribution of $SO_2$ ΔSCDs doesn't reflect well the physical distribution of $SO_2$ in the atmosphere. Notably, strong negative values are found over the ocean. It's worth mentioning that the fitting window we employ (331.5–358 nm) doesn't cover the strongest $SO_2$ absorption features. The inclusion of $SO_2$ in the fitting even leads to non-negligible changes in BrO ΔSCDs as well as increases in fitting uncertainties all across the globe (the upper right and lower left panels). Therefore, we decided to exclude $SO_2$ from the fitting.

However, it is a valid point that excluding $SO_2$ might lead to biases in BrO retrievals over volcanoes. Although we haven't added these OClO and $SO_2$ figures to the manuscript, we have added a brief description to Sect. 2.2.1 (Sect. 2.2 in the revised manuscript).

[Figure]

**Page 8, Table 1**: I suggest to highlight the trace gas absorption spectra in the list. For example by horizontal lines. Also add "the parameter are listed in their order of appearance in eq. 2".

→ We have added two horizontal lines and also added the phrase suggested by the reviewer.

**Page 9, Figure 2**: It would be beneficial to also include the residual spectrum in this plot as it gives information on potential residual structures originating from absorbers which are not accounted for.

→ We apologize for the confusion. The residual structures are already presented, as the blue curves in Fig. 2, referred to as "measured optical depths" in the caption, represent the sum of the modeled optical depths and the residuals.

We have added the following sentence to the caption to clarify it in the manuscript: "The measured optical depths are defined as the sum of modeled optical depths and residuals."

**Page 13**: In other studies (such as Seo et al. 2019) a uniform background of 3.5x1013 moleculesc cm-1 is used (based on Richter et al., 2002). Include how your background correction S_R typically is with respect to this value.

Richter et al., 2002: Richter, A., Wittrock, F., Ladstatter-Weissenmayer, A., and Burrows, J. P.: GOME measurements of stratospheric and tropospheric BrO, in: Remote Sensing of Trace Constituents in the Lower Stratosphere, Troposphere and the Earth's Surface: Global

Observations, Air Pollution and the Atmospheric Correction, edited by: Burrows, J. P. and Takeucki, N., Adv. Space Res., 11, 1667–1672, 2002.

→ The total VCD values from CAM-Chem for the reference sector are typically ~2.0–2.2×10$^{13}$ molecules cm$^{-2}$, which is smaller than the value from Seo et al. (2019) and Richter et al. (2002) (3.5×10$^{13}$ molecules cm$^{-2}$). However, there's a difference in the reference-sector latitudes between Seo et al. (2019) and this study. The CAM-Chem total columns vary spatially and temporally. For example, the VCD range within 30°S–30°N is 2.0–5.9×10$^{13}$ molecules cm$^{-2}$ for February, with the mean of 2.9×10$^{13}$ molecules cm$^{-2}$, which is closer to that from Seo et al. (2019).

We have added a brief description to Sect. 2.2.3 (Sect. 2.4 in the revised manuscript).

**Page 13 line 17-31**: It would improve the readability to include the names "S_R" and "S_B", when talking about these quantities in the text.

→ We have added $S_R$ and $S_B$ in the text.

**Page 37 line 20**: Please remove "including volcanic plumes". Volcanic application is not mentioned at all in the result section and as the major volcanic constituent "SO2" is not accounted for in the spectral fitting, this statement is questionable.

→ This is a valid point. We have removed that phrase.

**Page 38, line 13-18**, concerning the "modeled stratospheric BrO DeltaSCD": I assume the "modeled Delta SCDs" in line 17 is the same as the "modeled stratospheric BrO Delta SCD". Name both the same, and consider to mention this in the first sentence of its explanation (line 13).

→ The reviewer's assumption is correct. We have changed "modeled ΔSCDs" in line 17 to "modeled stratospheric BrO ΔSCDs." Also, in Line 13, we have changed the wording "stratospheric BrO columns from CAM-Chem" to "stratospheric BrO ΔSCDs from CAM-Chem." Then, we have added "stratospheric" throughout the paragraph to prevent readers' confusion.

**Page 38, Line 15-17**: From your explanation, it looks like the "modeled stratospheric BrO Delta SCD", which is subtracted by is defined in a way that its mean is zero at 0-10°N and non-zero elsewhere. Thus the "Delta SCD bias" will then be the complete retrieved SCD subtracted by zero at the equator and non-zero

→ The reviewer's comment is accurate, although it appears that a part of it might be missing.

**Page 38-39 and figure A1**, regarding the correlation with O3:

1. How have you combined the different O3 SCDs to one O3 SCD? Did you follow the formula proposed by Pukite and Wagner (2016) eq. 16? If so, please add a reference.

→ In this appendix, we didn't combine the two O$_3$ SCDs. That's because we didn't use the SCD values directly when calculating the correlation coefficients (R). As stated in the manuscript, the R values presented in this appendix are between the Jacobians derived within the retrieval algorithm, not between the SCDs.

However, we did combine the two O$_3$ SCDs for the calculation of the O$_3$ optical depth in Fig. 2. For that calculation, we used the formula proposed by Pukite et al. (2010) rather than Pukite and Wagner (2016). During the revision, we added the 2010 reference to the part where we describe Fig. 2 instead. (This reference was already cited several times elsewhere in the manuscript because we use the Taylor-series parameters for the retrieval.)

2. I do not see the benefit of looking at two O3 absorptions at 243 and 273K, as there is only O3 absorption. If you do not gain any benefit from using the two, then I suggest to skip this.

→ The reason why we don't combine the two O$_3$ temperatures here is that, during the retrieval process, the BrO cross section technically interferes with each of the two O$_3$ cross sections individually rather than interfering with a single combined O$_3$ absorption spectrum. Figure A1 demonstrates that there are fitting windows where we can avoid interference with one of the O$_3$ temperatures but not with the other. Our objective was to find a fitting window where we could avoid strong interference from both O$_3$ temperatures.

3. Should you chose to keep the distinction between 243 and 273K, how did you avoid a cross correlation between the two spectra (which are very similar)? Have you orthogonalized the O3 absorption spectrum at 273K w.r.t the one at 243K? Please state this in the text.

→ We haven't orthogonalized the O$_3$ cross section. An additional step is not required because what we aim to assess here is the correlation between the Jacobians of BrO and the others, which are all equally treated in the retrieval algorithm (or program) regardless of the species. The algorithm provides a covariance matrix as one of the outputs, and we simply chose the proper elements in this matrix to assess the correlation between the BrO Jacobian and another. We have added a brief description to this appendix.

**Page 39 line 8**: The choice of a percentile seems suboptimal to me and introduced strong data-selection biases and I would urge to change this. For instance, the pixel at high VZA will have a higher SCD compared to the nadir looking pixel and they will be selected thus more frequently. Additionally, tropospheric enhancements will be more dominant. If you want to select for high latitude and high SZA, then why not use latitude and SZA as a selection criterium?

Additionally, the paper is about a global product of BrO. Please justify why you have not also looked if the fit is also performing well at other regions (cf. Seo et al., 2019, who performed a retrieval interval mapping for several cases and also for an equatorial region).

→ We have made several changes in this analysis following the reviewer's comment. First, we have changed the selection criterion. Considering correlations between latitudes and SZAs, we used only latitudes. Second, we have included all latitude ranges. We have done the same analysis for (a) 90°S–60°S and 60°N–90°N combined (high latitudes), (b) 60°S–30°S and 30°N–60°N combined (middle latitudes), and (c) 30°S–30°N (low latitudes). In addition, to constrain the conditions for stratospheric bias analysis, we have excluded pixels with (a) cloud fractions > 0.2 and (b) snow or ice. Also, we have excluded pixels with SZAs > 80°.

The three figures just below show the results for high latitudes, middle latitudes, and low latitudes, respectively, in the order of presence. The results for the same month (April 2018) is presented as in the previous analysis.

[Figure]

[Figure]

[Figure]

Since the fitting window originally selected shows an excellent performance here, we didn't change the window. Furthermore, the example below for January 2018 for high latitudes demonstrates that biases could steeply increase if we choose a longer wavelength for the lower limit of the fitting window.

We have updated Appendix A based on these new results.

[Figure]

**There is one paper about long-term BrO time-series:**

Bougoudis, I., Blechschmidt, A.-M., Richter, A., Seo, S., Burrows, J. P., Theys, N., and Rinke, A.: Long-term time series of Arctic tropospheric BrO derived from UV–VIS satellite remote sensing and its relation to first-year sea ice, Atmos. Chem. Phys., 20, 11869–11892, https://doi.org/10.5194/acp-20-11869-2020, 2020.

I suggest to include this paper in your introduction. You can frame this as an advantage of the OMPS data-set who in difference to Bougoudis et al., 2020 does not require a complicated inter-calibration of the time-series of the different sensors.

→ The reference has been added following the reviewer's comment.

**Technical comments:**

**Page 2, Line 30**: As the authors are very thorough in giving a complete list of relevant references in the introduction, I would here also strive for completeness and include the other two studies of BrO from GOME-2: Hörmann et al., 2013 and Sihler et al., 2012 (already included in the references)

→ The references have been added.

**Page 3, Line 14**: also here I would complete the list of references who used an area as an estimate for the stratospheric correction and include Hörmann et al., 2013 to the list of Wagner et al., 2001; Hörmann et al., 2016).

→ The reference has been added.

**Page 4 line 22**: I believe you mean "local solar time"?

→ It has been revised.

**Page 13 line 17**: The comm in "(i.e., […])" is not needed.

→ The comma has been removed.

**Page 13 line 20**: add "separately" after "across-track position"

→ It has been added.

**Page 38 footnote**: somewhere there is a bracket missing or one too many.

→ The bracket has been removed.